# *AutoDavis*: Automatic and Dynamic Evaluation Protocol of Large Vision-Language Models on Visual Question-Answering

## Abstract

Large Vision-Language Models (LVLMs) have become essential for advancing the integration of visual and linguistic information. While existing benchmarks have laid a solid foundation for evaluation, they are often static, resource-intensive to build, and limited in adaptability. In comparison, automatic evaluation has shown promise in the textual domain, but the visual modality remains far less explored. To advance this frontier, in this work, we introduce AUTODAVIS, a first-of-its-kind automatic and dynamic evaluation protocol that enables on-demand benchmarking of LVLMs across specific capability dimensions. AUTODAVIS leverages text-to-image models to generate relevant image samples and then utilizes LVLMs to orchestrate visual question-answering (VQA) tasks, completing the evaluation process efficiently and flexibly. To ensure data diversity, our framework employs a hierarchical aspect-driven generation process enhanced with semantic graph-based constraints. To safeguard reliability, the framework incorporates a self-validation mechanism to detect and correct errors, along with an error-driven adjustment module to mitigate potential bias. Through an extensive evaluation of 11 popular LVLMs across five demanded user inputs (*i.e.*, evaluation capabilities), the framework shows effectiveness and reliability, offering a new paradigm for dynamic benchmarking of multimodal intelligence.

## 1 Introduction

Large Vision–Language Models (LVLMs) have rapidly advanced multimodal perception and reasoning (Liu et al., 2024a), enabling unified image–text understanding across tasks such as VQA, grounding, and GUI manipulation (Antol et al., 2015; Zou et al., 2023; Ghandi et al., 2023; Chen et al., 2024a). As LVLMs are deployed more broadly, *trustworthy evaluation* becomes a first-order concern: practitioners need to know *which* capabilities a model possesses, *under what* conditions it fails, and *how* those conclusions evolve as models and data change.

**Why static benchmarks are insufficient.** Recent benchmarks cover important axes of LVLM competence (Xu et al., 2024a; Liu et al., 2025; Li et al., 2023b;a; Yin et al., 2024; Ying et al., 2024; Fan et al., 2025), yet static, once-published test sets face three practical limitations: (i) *On-demand capability slicing.* Real users and domains require targeted probes (e.g., "spatial reasoning with occlusion at medium difficulty") that existing suites cannot flexibly instantiate at evaluation time. (ii) *Staleness and leakage.* Once public, a test distribution can be memorized or overfitted, inflating scores without genuine generalization. (iii) *Difficulty calibration.* As models improve, fixed items cease to be discriminative, and curating fresh, well-calibrated items is costly and slow. These limitations motivate *dynamic* benchmarks that can be generated, validated, and refreshed *on demand*.

**Our question.** Motivated by progress in using generative models for automated assessment in the text-only setting (Zhu et al., 2023b; Li et al., 2024; Huang et al., 2025), we ask: ***Can LVLMs themselves be used for automatic and dynamic evaluation in the visual domain?*** In other words, is it feasible to leverage LVLMs as *both* examiners and testset generators—under minimal human intervention—to produce flexible, reliable, and leakage-resistant multimodal tests? Doing so requires addressing (a) *capability targeting* from user intents to fine-grained skills, (b) *visual grounding and controllability* so that items truly test the intended skill rather than textual shortcuts (Chen et al.,

2024b), and (c) *leakage and self-enhancement bias* when the same family of models writes and takes the exam (Zheng et al., 2023; Ye et al., 2024).

**Our answer: AUTODAVIS, a dynamic evaluation protocol.** We propose AUTODAVIS, a protocol that turns user-specified aspects (e.g., *spatial understanding*) into validated VQA-style test items through a controlled, automated pipeline. Crucially, AUTODAVIS is organized around *three testable commitments* that operationalize the advantages of dynamic benchmarks:

- **C1—Flexibility at evaluation time.** Given an aspect and difficulty, the system can rapidly generate diverse, de-duplicated items that align with the requested capability while keeping cost/latency low.
- **C2—Leakage resistance by regeneration.** For a fixed capability definition, re-sampling the test distribution (*dynamic regeneration*) neutralizes gains attributable to exposure or overfitting, revealing true generalization.
- **C3—Visual grounding over textual shortcuts.** An explicit *no-image control* and an *error-driven option adjustment* strategy suppress text-only guessing, compelling models to use the visual signal (Chen et al., 2024b).

**Snapshot of AUTODAVIS.** Given a user query, an *examiner LVLM* decomposes the aspect into fine-grained components and proposes image descriptions at controllable difficulty. Text-to-image synthesis renders candidate scenes; a VQA-based *self-validation* checks image–question–answer alignment and rejects ill-posed items. An *error control* step fixes detected mismatches. To avoid conflict of interest and self-enhancement bias, we adopt a *multi-examiner* setting (distinct examiners vs. evaluatee). Finally, curated items are administered to the target LVLM, and responses are scored against references through an error-driven option adjustment. The protocol also defines a *no-image* ablation and *option-position balancing* as default controls.

**Research questions.** We instantiate the above as three empirical RQs that tie directly to C1–C3:

- **RQ1 (Flexibility).** Can AUTODAVIS produce high-alignment, diverse test items for user-specified aspects and difficulties with practical time/cost?
- **RQ2 (Leakage resistance).** Does dynamic regeneration of items for the *same* capability definition remove gains from distribution exposure or SFT on similar items?
- **RQ3 (Visual grounding).** Do the no-image control and error-driven option adjustment substantially reduce text-only guessing, indicating that items truly require vision?

**Our findings.** Using AUTODAVIS, we evaluate 11 widely used LVLMs across five core capability dimensions (see Figure 6). We observe (i) consistent performance drops with increased difficulty, (ii) substantial variance across models and aspects, with many failures in fine-grained visual reasoning, and (iii) strong suppression of text-only guessing under our controls. Dynamic regeneration neutralizes gains attributable to prior exposure, and human studies indicate high alignment between intended capability targets and realized items. These results suggest that dynamic benchmarks are a *necessary complement* to static suites: compatible with prior leaderboards yet uniquely suited for on-demand assessment and leakage stress-testing.

**Contributions.** (1) We introduce AUTODAVIS, a *dynamic, automatic* LVLM evaluation protocol that operationalizes flexibility, leakage resistance, and visual grounding as testable commitments, with concrete controls and procedures. (2) We provide comprehensive experiments—main evaluations, ablations of each module, option-position bias analyses, and human studies—demonstrating that the protocol yields targeted, discriminative, and reliable assessments, and that its results are highly correlated with established human-curated benchmarks. (3) We report an in-depth analysis of capability profiles across models and difficulties, highlighting systematic gaps in fine-grained visual reasoning. *As a secondary finding*, we show that AUTODAVIS-generated items can be repurposed for supervised fine-tuning to improve targeted capabilities; we treat this as complementary to (rather than a substitute for) the evaluation focus.

## 2 METHOD: AUTODAVIS

In this section we present AUTODAVIS, a framework for *dynamic, automatic* evaluation of LVLMs (notations are summarized in Appendix Q). The system uses an LVLM examiner $\mathcal{M}_v$ and a text-to-image model $\mathcal{M}_d$ to support robust and scalable assessments. As shown in Figure 1, the pipeline comprises four modules: *user-oriented aspect generation*, *guided description generation*, *image*

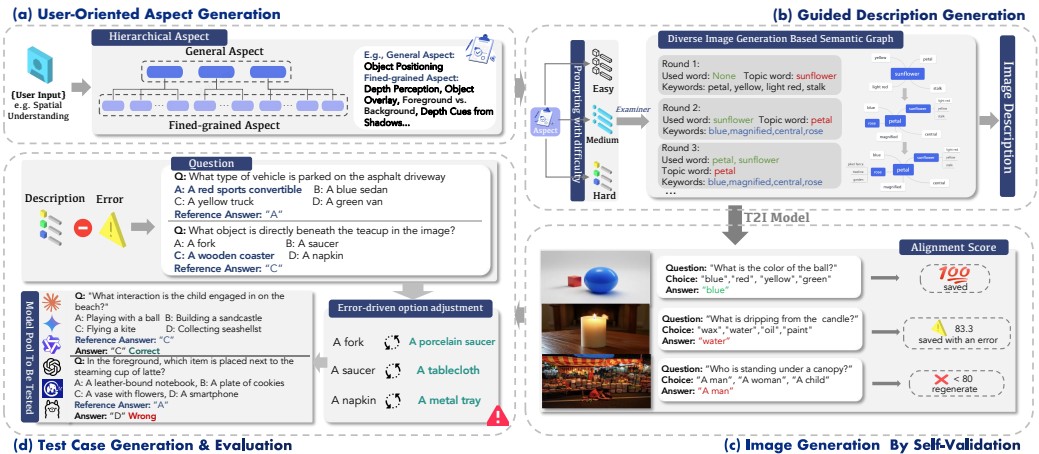

Figure 1: An overview of AUTODAVIS. The pipeline flows from (a) user-oriented aspect generation, which expands the user's requested evaluation scope into hierarchical aspects; (b) guided description generation via a semantic graph, creating varied textual prompts for the T2I model; (c) T2I image generation followed by self-validation to check image–description alignment; and (d) final test-case creation and evaluation with error-driven option adjustment. The arrows show how each module's output feeds into the next.

*generation with self-validation*, and *test-case generation and evaluation*. Each module is designed to address one or more of the following challenges.

- **Capability Coverage & Difficulty Control.** AUTODAVIS targets comprehensive capability coverage with explicit difficulty grading. The user-oriented aspect generation expands a user-specified aspect into fine-grained components, and the guided description generation instantiates them using a hierarchical aspect-guided strategy with semantic graph–based prompt generation. This yields diverse, de-duplicated test intents that align with requested capability dimensions and difficulties.
- **Validated & Error-Controlled Test Cases.** To ensure item quality and alignment, AUTODAVIS includes a self-validation step that checks image–question–answer consistency and filters ill-posed cases. An error-control mechanism then revises detected issues, reinforcing factual correctness and improving reliability before items enter evaluation.
- **Bias Minimization via Multi-Examiner & Option Adjustment.** To reduce bias and limit self-enhancement effects, AUTODAVIS maintains a *set of LVLM examiners* rather than relying on a single model, avoiding conflicts when generating and answering questions (Ye et al., 2024). In addition, *error-driven option adjustment* is applied during test-case generation to suppress text-only guessing strategies and increase the effective difficulty of the items for models that do not leverage visual information.

## 2.1 USER-ORIENTED ASPECT GENERATION

**User input.** The user input can specify an evaluation target focused on certain aspects of LVLMs' capability. AUTODAVIS covers the following key evaluation aspects, which are the most crucial for assessing the capabilities of LVLMs: *Basic Understanding*, *Spatial Understanding* (Li et al., 2023b), *Semantic Understanding* (Meng et al., 2024), *Reasoning Capacity* (Liu et al., 2025), and *Atmospheric Understanding* (Geetha et al., 2024). Notably, the user input is not limited to the above kinds, and can be customized as needed.

**Hierarchical aspect generation.** For each user input, we derive a set of aspects representing specific capability items. For example, as shown in Figure 6, *contextual comprehension* is an aspect under *Basic Understanding*. However, directly generating aspects from user input can lead to excessive repetition, reducing both diversity and reliability by overlapping in semantics and repeatedly evaluating the same capability. To mitigate this, we propose hierarchical aspect generation inspired by the previous study (Qin et al., 2023) to constrain the aspect generation process. Formally, given the user input $q$, we first generate $n$ general aspects $\{A_1^{(g)}, A_2^{(g)}, \ldots, A_n^{(g)}\}$ by $\mathcal{M}_v$, which can be formulated as: $\{A_1^{(g)}, A_2^{(g)}, \ldots, A_n^{(g)}\} = \mathcal{M}_v(q)$. These general aspects represent high-level evaluation dimensions based on $q$. Next, for each general aspect $A_i^{(g)}$, we further generate $m$ fine-grained

aspects $\{A_{i1}^{(f)}, A_{i2}^{(f)}, \ldots, A_{im}^{(f)}\}$, where each fine-grained aspect provides more specific criteria related to the general aspect. The fine-grained aspects are also generated by $\mathcal{M}_v$ and depend on both the user input $q$ and the corresponding general aspect $A_i^{(g)}$. The fine-grained aspect of generation can be represented as $\{A_{i1}^{(f)}, A_{i2}^{(f)}, \ldots, A_{im}^{(f)}\} = \mathcal{M}_v(q, A_i^{(g)})$. Thus, the hierarchical aspect generation yields a structured set of evaluation aspects (*i.e.*, fine-frained aspect) $\mathcal{A} = \bigcup_{i=1}^{n} \left( \{A_i^{(g)}\} \cup \bigcup_{j=1}^{m} \{A_{ij}^{(f)}\} \right)$, where $|\mathcal{A}| = mn$.

**Summary:** This module generates evaluation aspects based on user input by first identifying general capability categories. It then refines them into fine-grained sub-aspects through a hierarchical process to ensure diversity and reduce redundancy.

## 2.2 DIVERSE DESCRIPTION GENERATION

**Image description with difficulty grading.** To enable a more comprehensive evaluation, we introduce a difficulty-grading mechanism for the image descriptions, which includes the evaluation cases from different difficulties. We define three difficulty levels: easy, medium, and hard. We show the examples across different difficulties in Figure 8. The difficulty level $d$ is determined by key factors such as background complexity, element relationships, and the intricacy of textures, which are carefully discussed and designed from the perspective of visual perception. The generation of $\omega$ image descriptions $\{\mathcal{T}_{ij1}^d, \mathcal{T}_{ij2}^d, \ldots, \mathcal{T}_{ij\omega}^d\}$ for $A_{ij}^{(f)}$ at a specific difficulty level $d$ can be defined as: $\bigcup_{k=1}^{\omega} \{\mathcal{T}_{ijk}^d\} = \mathcal{M}_v(q, A_{ij}^{(f)}, d)$, where $d \in \{\text{easy}, \text{medium}, \text{hard}\}$. More details about the difficulty grading are provided in Appendix P.

**Semantic graph-constraint description generation.** A key challenge when generating image descriptions at the same difficulty level is minimizing repetitive elements and backgrounds, which can reduce the diversity and generalization of the evaluation. For example, given a user input $q$ related to spatial understanding, the model $\mathcal{M}_v$ might tend to produce descriptions centered around urban landscapes, potentially compromising the variety of test cases. To address this, we introduce a description optimization strategy using a semantic graph (Quillian, 1966) to enhance the diversity of image prompts generated by $\mathcal{M}_v$, with significant results referred to Figure 7 in Appendix H. Moreover, we show a visualization of specific words in Figure 9 and Figure 10 in Appendix H. The semantic graph is designed to assist AUTODAVIS in avoiding the generation of duplicate descriptions by serving as an iteratively updated structure tailored to prompts within the specified aspect. To illustrate the construction process, without loss of generality. Considering $e'$th iteration of prompt generation, a LLM identified topic word $t_e$ and a set of $|c|$ related keywords $K_e = \{k_{e1}, k_{e2}, \ldots, k_{ec}\}$ are selected. These keywords are added as nodes to the semantic graph $G$, where nodes are connected by edges representing semantic relationships between them.

Formally, let $G_e = (V_e, E_e)$ be the semantic graph generated at iteration $e$, and let $S_e = (V_{e-1} \cup \{t_e\} \cup K_e)$. Then $V_e = S_e \setminus f(S_e)$ represents the node set of topic words and keywords, and $E_e$ is the set of edges capturing the relationships between them. After each round of prompt generation, we apply a degree-based exclusion algorithm, where the number of excluded nodes is determined by a function $f(S_e)$. This function defines the number of top-degree nodes to be excluded, allowing flexibility in adjusting how many frequently used words are removed as the iterations progress. Here, the function $f(S_e)$ can be defined by users' requirements, and in our experiment we define it as $f(S_e) = \arg\max_{V' \subseteq S_e, |V'|=e} \sum_{v \in V'} \deg(v)$, where $\deg(v)$ represents the degree of node $v \in V_i$, or it could take a more complex form based on specific conditions. We mitigate redundancy and promote diversity in the generated prompts by excluding these high-degree nodes, which correspond to the most commonly used words. The function $f(S_e)$ offers the flexibility to control how aggressively the exclusion process operates based on the round number $e$.

Overall, the generation of an image description $\mathcal{T}_{ij}^e$ can be formalized as follows:

$$\bigcup_{k=1}^{\omega} \{\mathcal{T}_{ijk}^{de}\} = M_v(q, A_{ij}^{(f)}, V_e, d), \tag{1}$$

where $V_e$ represents the refined and diverse set of topic words and keywords after the exclusion mechanism has been applied. We show the detailed procedure in Appendix Q and scalability analysis in Appendix E.

Table 1: Effectiveness of hierarchical aspect generation under various hyperparameter settings.

| m=3, n=5 | | m=3, n=6 | | m=3, n=7 | | m=4, n=5 | | m=4, n=6 | | m=4, n=7 | |
|---|---|---|---|---|---|---|---|---|---|---|---|
| Raw | +Hierarchy | Raw | +Hierarchy | Raw | +Hierarchy | Raw | +Hierarchy | Raw | +Hierarchy | Raw | +Hierarchy |
| 0.767 | 0.778 (1.4% ↑) | 0.773 | 0.780 (1.0% ↑) | 0.779 | 0.825 (5.9% ↑) | 0.780 | 0.790 (1.3% ↑) | **0.786** | **0.849 (10.2% ↑)** | 0.798 | 0.842 (5.5% ↑) |

> **Summary:** This module generates image descriptions with controlled difficulty by assigning each prompt a level—easy, medium, or hard—based on visual complexity. To ensure diversity and reduce redundancy, it employs a semantic graph that guides prompt generation by excluding frequently used keywords through a degree-based filtering strategy.

### 2.3 Image Generation By Self-Validation

**Self-validation.** The image descriptions $\mathcal{T}_{ij}^d$ and their corresponding aspects $A_{ij}^{(f)}$ are subsequently provided to the text-to-image model for image generation. At this stage, a potential issue is the possibility of generated images $\mathcal{I}_{ij}^d$ not aligning with the descriptions, due to hallucinations inherent in the text-to-image model (Lee et al., 2023). To tackle this issue, drawing inspiration from TIFA (Hu et al., 2023), we employed a self-validation process to evaluate the consistency of images with their descriptions via VQA.

In the self-validation process $\mathcal{F}$, for each image $\mathcal{I}_{ij}^d$, based on its image description, $\mathcal{M}_v$ is asked to generate a set of simple questions $\Phi_{ij}^d = \{\phi_{ij1}^d, \phi_{ij2}^d, \ldots, \phi_{ijp}^d\}$ (*e.g.,"Is there a wooden chair in the image?"*), where $p$ denotes the question number to evaluate alignment. The function $\mathcal{F}$ takes the image $\mathcal{I}_{ij}^d$, its description $\mathcal{T}_{ij}^d$, and the set of questions $\Phi_{ij}^d$ as inputs and outputs an alignment score $S_{ij}^d$, which is calculated as the ratio of correctly answered questions to the total number of questions, denoted as $S_{ij}^d = \mathcal{F}(\mathcal{I}_{ij}^d, \mathcal{T}_{ij}^d, \Phi_{ij}^d)$. We set a threshold $\zeta$, where: (i) If $S_{ij}^d < \zeta$, the image $\mathcal{I}_{ij}$ will be reworked in line with the description until it meets the required standard; (ii) If $\zeta \le S_{ij}^d < 1$, the image meets the basic criteria but contains an error $\mathcal{E}_{ij}^d$, which will be documented for further processing (described in subsection 2.4); and (iii) If $S_{ij}^d = 1$, the image is considered to fully align with the description and is deemed acceptable. Additionally, we explored the potential of a self-validation generation loop using image editing models (see Appendix F for details).

> **Summary:** This module uses self-validation to ensure image-text alignment by generating VQA-based questions from the description. Images are accepted, flagged, or regenerated based on their alignment score, reducing hallucinations in text-to-image generation.

### 2.4 Test Case Generation & Evaluation

**Q&A generation with error control.** To enhance question generation accuracy, especially for addressing potential image flaws, we propose an error control mechanism. While self-validation helps, not all images can be guaranteed flawless. To mitigate biases from relying on a single examiner LVLM (Zhang et al., 2024b), we incorporate a diverse set of examiners, inspired by prior work (Bai et al., 2024). This approach reduces self-enhancement bias (Ye et al., 2024) and promotes greater diversity in evaluation.

When generating questions, we will only include the image description $\mathcal{T}_{ij}^d$ and any identified defects $\mathcal{E}_{ij}^d$ in the input to the examiners $\mathcal{M}_v^k$, where $k$ represents the index of the selected examiner from the set of available examiners. Each time a question is generated, an examiner $\mathcal{M}_v^k$ will be randomly selected. The function $\mathcal{M}_v^k$ generates the question $Q_{ij}^d$ based on the image description and errors, denoted as $Q_{ij}^d = \mathcal{M}_v^k(\mathcal{T}_{ij}^d, \mathcal{E}_{ij}^d)$.

Moreover, the options for each question will also be generated by a randomly selected examiner from the set. For each image, we will provide a related question $Q_{ij}^d$ (*e.g.*, *multiple-choice or true/false*). These questions, along with the accompanying images, will be presented to the LVLMs under evaluation for their response.

**Error-driven option adjustment.** A key challenge in evaluating LVLMs is the generation of distractor options that are both plausible and challenging. Distractors generated by LVLMs are often trivial, leading to inflated performance, even under high-difficulty settings or purely text-based eval-

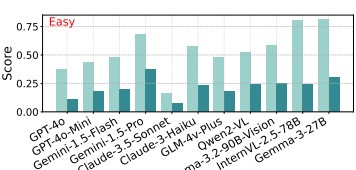 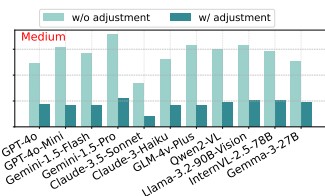 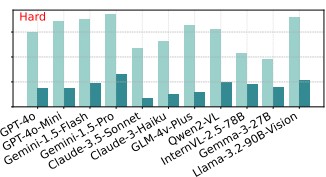

Figure 2: Models' accuracy (here we take "basic understanding" as an example) before and after the option adjustment, without image input (only guess the answer by textual information).

uations, as models can infer correct answers without relying on visual inputs (Chen et al., 2024b). Attempts to directly create harder options often result in low-quality, overly obscure distractors.

To address this, we propose *error-driven option adjustment*, a mechanism to improve distractor difficulty while preserving quality. For a given question $Q_{ij}^d$, with correct option $O_{ij}^c$ and distractors $O_{ij}^f = \{O_{ij,1}^f, \ldots, O_{ij,k}^f\}$. First, the randomly selected LVLM examiner is given $Q_{ij}^d$ and options $\{O_{ij}^c\} \cup O_{ij}^f$, but the correct option $O_{ij}^c$ is intentionally mislabeled as incorrect. The model generates a plausible "correct answer" $O_{ij}^{f*}$, given by $O_{ij}^{f*} = \mathcal{M}_v^k(Q_{ij}^d, O_{ij}^c)$. This generated $O_{ij}^{f*}$ replaces one or more original distractors in $O_{ij}^f$, resulting in an updated set $O_{ij}^{f'}$. This adjustment increases the difficulty of distinguishing the correct answer from distractors, as $O_{ij}^{f'}$ now contains plausible alternatives generated by the model itself under adversarial conditions. For detailed mathematical proofs and examples, see Appendix A and Appendix G, respectively.

**Evaluation.** Accuracy was measured by comparing the model response $\mathcal{P}_{ij}^d$ with the reference $A_{ij}^d$, computed as the proportion of correct answers.

**Summary:** This module generates questions and options with error control by randomly selecting examiner LVLMs to reduce bias. It enhances distractor difficulty through an error-driven adjustment mechanism, where models generate plausible alternatives by treating the correct answer as incorrect. Accuracy is computed by comparing model responses with reference answers.

## 2.5 THEORETICAL GUARANTEE FOR AUTODAVIS

We formally define accuracy and diversity for a set of generated test cases, and prove that AU-TODAVIS achieves strong guarantees on both. Specifically, as shown in Appendix B, with only $O(\log N)$ validation trials per case, the empirical accuracy and diversity of *any sufficiently large subset* of test cases concentrate near their true values with high probability.

*For any subset $\mathcal{S} \subseteq \mathcal{S}_{\text{full}}$ with $|\mathcal{S}| \geq k$, the empirical accuracy and diversity satisfy*

$$\text{Acc}(\mathcal{S}) \geq \tau_{\text{acc}} - O\left(\frac{1}{\sqrt{r|\mathcal{S}|}}\right), \quad \text{Div}(\mathcal{S}) \geq \tau_{\text{div}} - O\left(\frac{1}{\sqrt{r|\mathcal{S}|}}\right)$$

*with high probability.*

## 3 EXPERIMENTS

**Experiment Setup.** We evaluate 11 representative LVLMs, including proprietary models such as GPT-4o, Claude-3.5-Sonnet, and Gemini-1.5-Pro, as well as competitive open-source models like Qwen2-VL, InternVL-2.5-78B, and Llama-3.2-90B-Vision (see Table 9). We designate GPT-4o, Gemini-1.5-Pro, and Claude-3.5-Sonnet as *examiner models* due to their consistently strong performance. Image generation is handled by Flux-1.1-Pro (Labs, 2023), selected for its reliable output quality. Details on model selection and hyperparameter settings are provided in Appendix H.

## 3.1 MAIN RESULTS

*Performance drops significantly with increasing difficulty.* As shown in Table 2 and Table 3, across all models and user inputs, scores decrease as the difficulty progresses from easy to medium to hard. In the basic understanding, the average score across all models drops from approximately 75.01% at easy to 37.54% at hard. This trend is consistent across the Table 2, where the hard is noticeably shorter than the easy and medium.

Table 2: Performance (accuracy) of all models on five user inputs and three difficulty levels (red and blue represent the highest and lowest scores, respectively, in each row). Claude-3.5 is Claude-3.5-Sonnet and Llama-3.2-90B is Llama-3.2-90B-Vision.

| User Input | Difficulty | Model | | | | | | | | | | |
|---|---|---|---|---|---|---|---|---|---|---|---|---|
| | | GPT-4o | GPT-4o mini | Gemini-1.5-Flash | Gemini-1.5-Pro | Claude-3.5 | Claude-3-Haiku | GLM-4v-Plus | InternVL-2.5-78B | Gemma-3-27B | Qwen2-VL | Llama-3.2-90B |
| BASIC. | Easy | 77.03% | 72.73% | 78.47% | 78.47% | 84.69% | 70.81% | 74.64% | 80.01% | 81.20% | 76.56% | 50.72% |
| | Medium | 56.65% | 46.35% | 44.21% | 51.93% | 57.51% | 44.64% | 47.64% | 58.20% | 51.24% | 53.88% | 32.19% |
| | Hard | 41.15% | 30.53% | 34.07% | 39.82% | 45.58% | 31.42% | 34.07% | 43.21% | 38.15% | 45.58% | 29.78% |
| SPATIAL. | Easy | 59.22% | 44.69% | 60.00% | 60.56% | 65.00% | 42.78% | 57.22% | 63.00% | 64.05% | 62.78% | 36.31% |
| | Medium | 34.50% | 22.71% | 29.69% | 41.48% | 38.86% | 25.33% | 31.44% | 38.40% | 42.00% | 36.68% | 24.45% |
| | Hard | 21.46% | 15.14% | 23.32% | 27.35% | 26.01% | 15.84% | 17.49% | 28.33% | 29.12% | 25.91% | 14.55% |
| SEMAN. | Easy | 73.10% | 64.62% | 68.53% | 70.56% | 72.59% | 55.54% | 62.94% | 67.28% | 71.11% | 66.33% | 42.64% |
| | Medium | 61.90% | 46.75% | 54.11% | 58.44% | 61.90% | 46.75% | 51.95% | 64.20% | 60.00% | 61.04% | 39.57% |
| | Hard | 43.64% | 33.79% | 44.55% | 47.27% | 45.45% | 26.36% | 36.36% | 42.00% | 49.34% | 44.55% | 30.14% |
| REASON. | Easy | 57.06% | 46.20% | 56.40% | 57.56% | 57.00% | 36.63% | 46.51% | 62.05% | 57.40% | 53.49% | 36.26% |
| | Medium | 47.16% | 35.22% | 40.43% | 47.39% | 46.09% | 25.22% | 36.96% | 45.63% | 43.06% | 48.03% | 26.64% |
| | Hard | 38.21% | 30.66% | 21.43% | 35.71% | 36.59% | 14.29% | 24.53% | 41.08% | 39.02% | 39.15% | 25.00% |
| ATMOS. | Easy | 60.20% | 55.94% | 58.91% | 66.83% | 60.40% | 48.02% | 52.48% | 64.24% | 63.20% | 56.72% | 36.63% |
| | Medium | 39.73% | 30.67% | 39.21% | 39.65% | 44.93% | 27.75% | 33.48% | 43.02% | 40.00% | 39.21% | 28.63% |
| | Hard | 33.79% | 18.26% | 28.64% | 30.45% | 34.55% | 15.00% | 23.18% | 38.20% | 33.05% | 31.05% | 22.73% |
| AVERAGE | | 49.65% | 39.63% | 45.46% | 50.23% | 51.81% | 35.11% | 42.06% | 51.73% | 50.67& | 49.40% | 31.75% |

*Basic understanding is the simplest task, while spatial understanding represents the most challenging one:* Most models perform better in *basic understanding* compared to *spatial understanding*. For example, the average score for basic understanding at easy difficulty is 75.01%, while *spatial understanding* achieves lower averages of 55.96%. This indicates the imbalance of current LVLMs' capabilities and emphasizes the importance of holistic improvement.

Table 3: Average accuracy for various user inputs at different difficulty levels (red and blue represent the highest and lowest scores in each column).

| User Input | Easy | Medium | Hard | Average |
|---|---|---|---|---|
| BASIC. | 75.01% | 49.45% | 37.54% | 54.00% |
| SPATIAL. | 55.96% | 33.19% | 22.18% | 37.11% |
| SEMAN. | 65.02% | 55.13% | 40.28% | 53.48% |
| REASON. | 51.46% | 40.11% | 31.42% | 41.00% |
| ATMOS. | 56.64% | 36.93% | 28.06% | 40.54% |

*Some models achieve scores below 25% on tasks with medium and hard difficulty, which suggests that the model performs worse than random guessing.* As shown in Table 2, GPT-4o Mini, Gemini-1.5-Flash, Claude-3.5-Haiku, GLM-4v-Plus, and Llama-3.2-90B-Vision achieved scores below 25% on hard tasks involving spatial understanding and reasoning capacity. This indicates that AU-TODAVIS effectively challenges the evaluated models, mitigating the answer leakage issue where models rely solely on textual information to answer correctly (we will discuss this in detail in subsection 3.2).

## 3.2 EXPLORATORY EXPERIMENTS

We aim to develop a deeper understanding of the following questions at this section: **Q1:** What impact does error-driven option adjustment have on visual question answering? **Q2:** What are the advantages of multi-evaluator assessment compared to single-evaluator evaluation? **Q3:** Does the position of correct options affect model performance? **Q4:** Is the ranking of the same model consistent across different user inputs and different levels of difficulty?

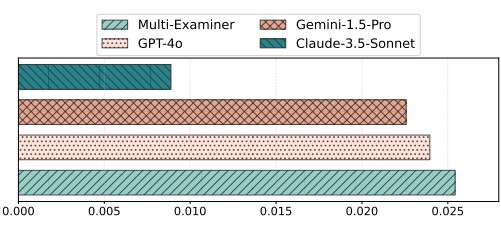
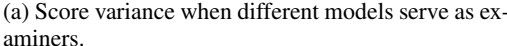

(a) Score variance when different models serve as examiners.

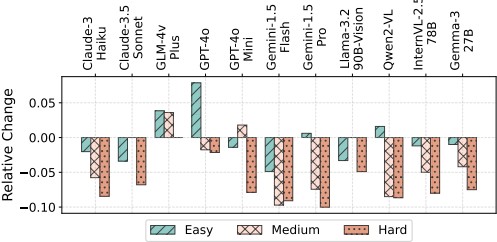

(b) Answer distribution under position-bias versus uniform settings.

Figure 3: Analysis of examiner-induced variance and position bias in AutoDavis.

*Unnecessary visual information was significantly reduced after the option adjustment.* As illustrated in Figure 2, in the absence of image input, the scores of various models before option adjustment were concentrated within the range of 40 to 60. Interestingly, without option adjustment, the "guess accuracy" significantly increases with the difficulty level. For instance, GPT-4o can guess less than 40% answers correctly on the easy level while achieving a guessing accuracy of 60% on the hard level. After option adjustment, the performance of all models declined significantly, demonstrating that error-driven option adjustment effectively mitigates answer leakage in the questions.

*Under the setting of multiple examiners, the performance differences among models become more pronounced.* As shown in Figure 3a, when multiple evaluators are employed, the variance in the performance of the three models is significantly larger. This suggests that using a single model for evaluation may reduce the difference gap between models, leading to an "evaluation bottleneck" while the multi-examiner can mitigate this to achieve a more diverse assessment and lead to a larger performance gap across models. We systematically explored the bias of multiple examiners in subsection 3.6.

*The position of the correct options can influence model performance.* To avoid potential position bias in the evaluation process, we ensured an even distribution of correct options. To further explore the possible effects of position bias on model performance, we conducted a case study where all correct answers were intentionally placed in option A and compared it to the evenly distributed case (25% per option) to assess the necessity of maintaining even distribution, as shown in Figure 3b. The deviation rate was calculated using the formula $R = \frac{S_A - S_U}{S_U}$, where $S_A$ is the score with correct answers concentrated in option A, and $S_U$ is the score under evenly distributed conditions. As illustrated in Figure 3b, most models exhibit decreased performance after the answer distribution is balanced, indicating a strong preference for selecting option A.

*Different models have distinct performance patterns under various difficulties.* As shown in Figure 4, Claude-3.5-Sonnet achieves the best performance, consistently ranking highest across most tasks, while Llama-3.2-90B-Vision performs the weakest. Interestingly, models like Qwen2-VL rank higher on hard tasks than on easy ones, whereas models like Gemini-1.5-Pro excel on easy tasks but decline slightly on hard ones. These findings highlight the importance of difficulty-based evaluation and demonstrate AUTO-DAVIS's effectiveness in revealing such performance imbalances.

Table 4: Average performance (Accuracy) of all models at different difficulty levels (red and blue represent the highest and lowest scores, respectively, in each column).

| Model | Easy | Medium | Hard |
|---|---|---|---|
| GPT-4o | 65.32% | 47.99% | 35.65% |
| GPT-4o mini | 56.83% | 36.34% | 25.68% |
| Gemini-1.5-Flash | 64.46% | 41.53% | 30.40% |
| Gemini-1.5-Pro | 66.79% | 47.78% | 36.12% |
| Claude-3.5-Sonnet | 67.93% | 49.86% | 37.63% |
| Claude-3-Haiku | 50.82% | 33.94% | 20.58% |
| GLM-4v-Plus | 58.76% | 40.29% | 27.13% |
| InternVL-2.5-78B | 67.22% | 49.60% | 38.44% |
| Gemma-3-27B | 67.20% | 47.25% | 37.60% |
| Qwen2-VL | 63.17% | 47.77% | 37.25% |
| Llama-3.2-90B-Vision | 40.51% | 30.30% | 24.44% |

### 3.3 HUMAN EVALUATION

We conducted human evaluations to assess the alignment between the generated questions and the reference answers. The evaluations were performed using a metric we refer to as the *alignment rate*, defined as the proportion of aligned samples out of the total. Detailed procedures and results for the human evaluation can be found in Appendix L. The results show that the model achieves high alignment across difficulty levels: 95.20% for Easy tasks, 88.13% for Medium, and 84.55% for Hard. This demonstrates the model's ability to generate well-aligned question-answer pairs, excelling in simpler tasks while maintaining strong performance under higher complexity, highlighting its effectiveness in producing accurate, contextually appropriate outputs for visual tasks.

### 3.4 CORRELATION WITH EXISTING BENCHMARK

As a dynamic and automated evaluation framework, AUTODAVIS is compared against MMMU (Yue et al., 2024), currently the largest and most comprehensive LVLM benchmark. Notably, top-performing models in MMMU—such as Claude-3.5-Sonnet and Gemini-1.5-Pro—also achieve high rankings in AUTODAVIS (see Table 4), while weaker models like Llama-3.2-90B-Vision consistently underperform across both. To quantify this alignment, we computed the Spearman rank correlation (Bauermeister et al., 2022) between the rankings of models that appear in both benchmarks. The resulting coefficient of **0.817** demonstrates strong consistency, supporting the reliability of AUTO-DAVIS as an evaluation framework.

Figure 4: The model performance ranking given five user inputs under different difficulty levels.

## 3.5 SCALABILITY & COST ANALYSIS

We evaluate the overhead of semantic graph–based prompt generation and the end-to-end cost of AUTODAVIS. Because graphs are constructed per aspect and discarded (Appendix E), scalability depends on aspect/prompt counts rather than a global structure: across 1–10 aspects and 2–10 prompts per aspect, the average time per prompt stays at ∼3–5 s; generating 100 prompts over 10 aspects finishes in 468.34 s (≈7.8 min) sequentially; peak/average memory is 32.84/26.49 MB with 0.26 MB per prompt (Table 7); aspects are trivially parallelizable. For cost, using GPT-4o, Gemini 1.5-Pro, Claude 3.5-Sonnet, and Flux-1.1-Pro (Table 17), the total is **$51.82**, dominated by image generation ($43.20, ∼83.4%); stage-wise details are in Table 18 (Appendix M).

## 3.6 AUTODAVIS AS A TRAINING DATA GENERATOR

It naturally raises the question of *whether **AUTODAVIS**'s generated data can also benefit LVLM training*. Prior work highlights both the promise and pitfalls of synthetic training data, including risks of overfitting or leakage and uncertain generalization to human benchmarks (Long et al., 2024; Liu et al., 2024b). To probe these issues, we conduct two targeted experiments evaluating the training utility and generalization of AUTODAVIS-generated data.

**Mitigating Data Leakage via Dynamic Regeneration.** Static benchmarks can suffer from data leakage, where models fine-tuned on a subset of the benchmark's data show inflated performance across the entire test set (Xu et al., 2024b). AUTODAVIS directly mitigates this risk by dynamically generating new, out-of-distribution test data on the fly. In Figure 5, we demonstrate this with a leakage simulation: a model fine-tuned on our data (SFT Model) achieved an artificially high score of **62.50%** on a test set from the same data distribution (Test-Set-1). However, when evaluated on a completely fresh, dynamically generated benchmark with a new distribution (Test-Set-2), its advantage vanished entirely, with the score dropping to

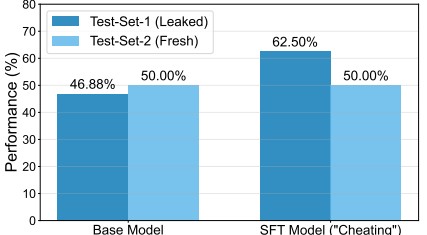

Figure 5: Demonstration of data leakage mitigation. The SFT Model's advantage on the "leaked" set is neutralized on the fresh set.

**50.00%**—matching the base model. This confirms that AUTODAVIS's on-the-fly regeneration is a crucial mechanism for ensuring fair evaluation by neutralizing leakage-based advantages.

**Validating Generalization on External Benchmarks.** To prove our generated data imparts generalizable skills, we fine-tuned the `Qwen2.5-VL-3B-Instruct` model on a diverse set of 2,400 samples from AUTODAVIS. We then evaluated it on 300 questions from the human-annotated **MM-Bench** (Liu et al., 2025) to break the "in-domain" cycle. The experiment confirmed a significant real-world improvement: the fine-tuned model's accuracy on MMBench rose from **79.0%** (base model) to **86.2%**, a 7.2-point gain. This result validates that AUTODAVIS can generate effective training data that enhances model abilities on external benchmarks.

In addition, we investigated how different image styles (Appendix N) and resolution Appendix I.1 affect model performance and conducted an analysis of model errors (Appendix R).

Table 5: Rank stability metrics when excluding each examiner (LOEO analysis). A high Spearman $\rho$ indicates robustness against single-examiner dominance.

| Excluded Examiner | Spearman $\rho$ | Kendall $\tau$ | Max Rank Shift |
|---|---|---|---|
| GPT-4o | 0.914 | 0.867 | 1 |
| Claude-4-Sonnet | 0.600 | 0.333 | 2 |
| Qwen3-VL | 0.943 | 0.867 | 1 |

### 3.7 EXAMINER ROBUSTNESS AND BIAS MITIGATION

To address the major concern regarding examiner bias and circularity risk, we conducted a comprehensive ablation study using three conditions: **Baseline (Multi-Examiner)**, **LOEO (Leave-One-Examiner-Out)**, and **Single-Examiner**. We utilize three examiners for question generation: the proprietary models GPT-4o and Claude-4-Sonnet, and the open-source model Qwen3-VL. Our goal was to quantify cross-family variance and prove the robustness of the aggregated ranking. The detailed experimental design, specific model scores, and ICC decomposition are presented in Appendix H.

*1. Rank Stability (Cross-Judge Robustness).* We assessed the stability of the final model ranking by comparing the correlation when excluding each examiner (LOEO condition). This directly quantifies the reliance on any single examiner.

The high rank correlations (Spearman $\rho = 0.914$–$0.943$ for GPT-4o and Qwen3-VL) demonstrate strong cross-judge robustness, showing that the evaluation ranking is not dominated by any single examiner and remains stable even when one examiner is excluded.

*2. Examiner Influence and Self-Bias Mitigation.* We quantified the influence of each examiner by measuring the average score change when they are excluded. This reveals the existence and direction of individual biases.

The data shows influence is moderate (3% to 6% score shifts) and varies directionally (e.g., GPT-4o and Qwen3-VL are more lenient, while Claude-4-Sonnet is more strict). Analysis of the Single-Examiner condition confirms that the multi-examiner baseline effectively neutralizes individual self-bias, further supporting the observed rank stability.

Table 6: Examiner influence scores (average score change when excluded).

| Examiner | Influence Score |
|---|---|
| GPT-4o | +0.053 (more lenient) |
| Claude-4-Sonnet | -0.032 (more strict) |
| Qwen3-VL | +0.061 (more lenient) |

*3. Cross-Family Agreement.* We also quantified cross-family agreement using the Inter-Class Correlation (ICC). The ICC(2,1) value of **0.297** quantifies the cross-family variance while confirming that the multi-examiner setup effectively aggregates these differing perspectives.

## 4 CONCLUSION

In this work, we introduce AUTODAVIS, an dynamic and automatic framework designed to benchmark LVLM. It integrates a series of innovative modules that ensure diversity and reliability in dataset generation and evaluation. Extensive experiments demonstrate its robustness and unbiased process, offering a solid foundation for future research.

## ETHICS STATEMENT

This research adheres to the ethical principles of academic integrity and responsible innovation. Our work does not involve human subjects, private user data, or sensitive personal information. The data generated and used for evaluation within our framework is synthetic and created programmatically.

We acknowledge that the large language and vision-language models (e.g., GPT-4o, Claude-3.5-Sonnet) that form the core components of our framework may inherit biases from their original

training data. While a detailed audit of these biases is outside the scope of this work, we recognize its importance for any downstream application. Our framework, AUTODAVIS, is presented as a tool for research and evaluation, intended to help the community better understand and assess the capabilities and limitations of LVLMs.

The primary goal of our research is to improve the robustness and transparency of AI model evaluation, which we believe is a positive contribution to the field's overall safety and reliability. We encourage the use of our framework for constructive purposes that advance the understanding and responsible development of artificial intelligence.

## REPRODUCIBILITY STATEMENT

We are committed to ensuring the reproducibility of our research. To this end, we provide the following resources and details.

**Source Code.** The complete source code for the AUTODAVIS framework, including all scripts used for data generation, model evaluation, and analysis presented in this paper, will be made publicly available on GitHub upon publication.

**Models and Dependencies.** Our framework relies on several API-based large models, including GPT-4o, Claude-3.5-Sonnet, Gemini-1.5-Pro for language tasks, and Flux-1.1-Pro for image generation. We have specified the models used for each component within the paper. While we acknowledge that results from API-based models may have minor variations over time, our framework's multi-examiner protocol is designed to enhance the stability of evaluation outcomes. All other key software dependencies will be detailed in the repository's requirements file.

**Experimental Details.** All hyperparameters, prompts, and configuration details for our experiments are described in the main body of the paper and further elaborated in the Appendix. Since our framework is a data generator, researchers can use our released code to regenerate the exact datasets used in our study and to create new benchmarks for their own work. We believe this provides a clear and direct path for the full replication of our results.

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

## A   PROOF OF ERROR-DRIVEN OPTION ADJUSTMENT (EOA)

### A.1   PROBLEM SETUP AND DEFINITION

Let $Q$ denote a VQA prompt consisting of an image and a question. The original option set is
$$\mathcal{O} = \{O_c, O_1, \ldots, O_{k-1}\},$$
where $O_c$ is the ground-truth answer. For any language model evaluated *without* the image, let
$$p_{\text{text}}(O \mid Q)$$
be the probability (or softmax score) it assigns to option $O$.

**Definition 1 (Error-Driven Option Adjustment (EOA))** *Given $(Q, \mathcal{O})$, ask the examiner model $\mathcal{M}_v$ to "fix" the claim that $O_c$ is wrong. Let*
$$\tilde{O} = \arg \max_{O \neq O_c} p_{\text{text}}(O \mid Q)$$
*be the most plausible distractor under the text-only distribution. The adjusted option set is*
$$\mathcal{O}' = \{O_c, \tilde{O}\} \cup (\mathcal{O} \setminus \{O_c, \tilde{O}\}),$$
*i.e. we replace all but the top two contenders with this new pair $\{O_c, \tilde{O}\}$ plus the remaining $k - 2$ distractors.*

**Assumption 1 (Sampling and Boundedness)** *We assume that the option set $\mathcal{O}$ is finite with $|\mathcal{O}| = k$; that we obtain $m$ independent and identically distributed samples $X_1, \ldots, X_m \sim p_{\text{text}}(\cdot \mid Q)$, each taking values in $\mathcal{O}$; and that each indicator $\mathbf{1}\{X_i = O\}$ is bounded in $[0, 1]$, so that Hoeffding's inequality applies.*

**Lemma 1 (Sup-norm PAC bound)** *Under Assumption 1, define the empirical distribution*
$$\hat{p}_{\text{text}}(O) = \frac{1}{m} \sum_{i=1}^{m} \mathbf{1}\{X_i = O\} \quad (O \in \mathcal{O}).$$
*Then for any $\varepsilon > 0$ and $\delta \in (0, 1)$, if*
$$m \geq \frac{1}{2\varepsilon^2} \ln \frac{2k}{\delta},$$
*we have with probability at least $1 - \delta$ (Shalev-Shwartz & Ben-David, 2014) that*
$$\|\hat{p}_{\text{text}} - p_{\text{text}}\|_\infty = \max_{O \in \mathcal{O}} |\hat{p}_{\text{text}}(O) - p_{\text{text}}(O)| \leq \varepsilon.$$

### A.2   DIFFICULTY GUARANTEE UNDER PAC

**Theorem 1 (EOA Difficulty Guarantee)** *Under the event of Lemma 1, the adjusted challenge in Definition 1 satisfies*
$$P_{\text{pre}} = \Pr\left[\arg \max_{O \in \mathcal{O}} \hat{p}_{\text{text}}(O \mid Q) = O_c\right], \quad P_{\text{post}} = \Pr\left[\arg \max_{O \in \mathcal{O}'} \hat{p}_{\text{text}}(O \mid Q) = O_c\right],$$
*and with probability at least $1 - \delta$,*
$$P_{\text{post}} \leq \frac{1 + \varepsilon}{2} < P_{\text{pre}},$$
*hence*
$$R^*_{\text{text,post}} \geq R^*_{\text{text,pre}} + \tfrac{1}{2}(1 - \varepsilon).$$

**Proof 1** *Condition on $\|\hat{p}_{\text{text}} - p_{\text{text}}\|_\infty \leq \varepsilon$. Since $O_c$ maximizes $p_{\text{text}}$ over $\mathcal{O}$, define*
$$\Delta = p_{\text{text}}(O_c) - p_{\text{text}}(\tilde{O}) \geq 0.$$
*By the sup-norm bound,*
$$\hat{p}_{\text{text}}(O_c) \in [p_{\text{text}}(O_c) \pm \varepsilon], \quad \hat{p}_{\text{text}}(\tilde{O}) \in [p_{\text{text}}(\tilde{O}) \pm \varepsilon].$$
*If $\Delta \leq 2\varepsilon$, these intervals overlap, so the best-guess accuracy on $\{O_c, \tilde{O}\}$ is at most $(1 + \varepsilon)/2$. Thus*
$$P_{\text{post}} \leq \tfrac{1}{2}(1 + \varepsilon) < P_{\text{pre}}.$$
*The Bayes-risk bound follows from $R^* = 1 - P$. All claims hold with probability $\geq 1 - \delta$.*

## B    PROOF OF ACC. & DIV. TRADEOFF

We analyse the full set of $N = n \times m$ test cases $\mathcal{S}_{\text{full}} = \{s_{ij}\}_{i=1,j=1}^{n,\,m}$ generated by AUTODAVIS for a query $q$. All notation ($Q_{ij}^d$, $O_{ij}^d$, $\mathcal{T}_{ij}^d$, $\mathcal{I}_{ij}^d$) is denoted as $s_{ij}$ and the self-validation function $r_{ij} = F\big(I_{ij}^d, T_{ij}^d, \Phi_{ij}^d\big) \in [0,1]$ follow §2.4 of the main paper.

**Diversity.** Embed query and test cases via $\varphi(\cdot)$ (uniform embedding model) and set

$$d(q,s) = \frac{1 - \cos\big(\varphi(q), \varphi(s)\big)}{2} \in [0,1], \qquad \text{Div}(\mathcal{S}) = \frac{1}{|\mathcal{S}|} \sum_{s \in \mathcal{S}} d(q,s).$$

**Accuracy.** $\text{Acc}(\mathcal{S}) = \dfrac{1}{|\mathcal{S}|} \sum_{s \in \mathcal{S}} r(s).$

**Assumption 2 (Bounded observations)** *Single-trial values of $d(q,s)$ and $r(s)$ lie in $[0,1]$.*

**Assumption 3 (Monte-Carlo estimation)** *Each $d(q,s)$ and $r(s)$ is estimated from $r$ i.i.d. trials; write the empirical means as $\widehat{d}(q,s)$ and $\widehat{r}(s)$.*

**Assumption 4 (Finite pool)** *The total number of cases is $N = |\mathcal{S}_{\text{full}}| < \infty$.*

**Lemma 2 (Hoeffding concentration (Hoeffding, 1963, Th. 2))** *Let $X_1, \ldots, X_r \sim [0,1]$ be i.i.d. with mean $\mu$. For $\bar{X} = \frac{1}{r} \sum_{t=1}^{r} X_t$ and any $\epsilon > 0$,*

$$\Pr\big[\,|\bar{X} - \mu| > \epsilon\,\big] \le 2 \exp(-2r\epsilon^2).$$

**Theorem 2 (Uniform sublinear guarantee over subsets)** *Fix accuracy/diversity thresholds $(\tau_{\text{acc}}, \tau_{\text{div}}) \in (0,1)^2$, failure probability $\delta \in (0,1)$, and minimum subset size $k_{\min} \le N$. If*

$$r \ge \max\left\{\frac{\ln(4N^2/\delta)}{2\tau_{\text{acc}}^2}, \frac{\ln(4N^2/\delta)}{2\tau_{\text{div}}^2}\right\}, \tag{1'}$$

*then with probability at least $1 - \delta$, for all subsets $\mathcal{S} \subseteq \mathcal{S}_{\text{full}}$ with $|\mathcal{S}| \ge k_{\min}$, it holds that*

$$\mathbf{Acc}(\mathcal{S}) \ge \tau_{\text{acc}} - \sqrt{\frac{\ln(2N^2/\delta)}{2r|\mathcal{S}|}}, \quad \mathbf{Div}(\mathcal{S}) \ge \tau_{\text{div}} - \sqrt{\frac{\ln(2N^2/\delta)}{2r|\mathcal{S}|}}.$$

*Hence for any subset $\mathcal{S}$ of size $\ge k_{\min}$, accuracy and diversity remain near the global thresholds, with sublinear validation trials $r = O(\log N)$.*

**Proof 2** *Fix any subset $\mathcal{S}$ of size $k \ge k_{\min}$. Each term in $Acc(\mathcal{S})$ is an average of $r$ i.i.d. bounded values $r(s) \in [0,1]$. By Hoeffding's inequality and union bound over $N$ elements:*

$$\Pr\big[\,|\widehat{r}(s) - \mathbb{E}r(s)| > \epsilon\,\big] \le 2 \exp(-2r\epsilon^2), \quad \forall s \in \mathcal{S}_{\text{full}}.$$

*Take $\epsilon = \sqrt{\frac{\ln(2N^2/\delta)}{2r}}$, and apply union bound over all $\binom{N}{k} \le N^k \le N^N$ subsets and $N$ individual elements. So with probability at least $1 - \delta$, for any $\mathcal{S} \subseteq \mathcal{S}_{\text{full}}$ with $|\mathcal{S}| \ge k$, we have:*

$$|Acc(\mathcal{S}) - \mathbb{E}[Acc(\mathcal{S})]| \le \sqrt{\frac{\ln(2N^2/\delta)}{2r|\mathcal{S}|}},$$

*and similarly for $Div(\mathcal{S})$. Rewriting gives the deviation bounds from thresholds $\tau_{\text{acc}}, \tau_{\text{div}}$.*

*This gives a uniform convergence bound over all subsets with $k \ge k_{\min}$.*

## C  RELATED WORKS

**Benchmark for LVLMs.** The emergence of LVLMs greatly promoted the development of multi-modal models, showcasing exceptional progress in their multimodal perception and reasoning capabilities. This makes previous benchmarks, which focused on isolated task performance (Karpathy & Fei-Fei, 2015; Antol et al., 2015) insufficient to provide a comprehensive evaluation. Subsequent studies have introduced various benchmarks to assess LVLMs across a range of multimodal tasks (Goyal et al., 2017; Lin et al., 2014; Russakovsky et al., 2015). However, these benchmarks often lack fine-grained assessments of capabilities and robust evaluation metrics. Hence, recent works (Liu et al., 2025; Ying et al., 2024; Fu et al., 2024; Yu et al., 2023; 2024; Zhou et al., 2024b), as well as more recent additions (Yao et al., 2025; Xie et al., 2025), underscore the critical need for developing advanced, comprehensive benchmarks to more accurately assess multimodal understanding and reasoning capabilities of LVLMs. **Specifically, LENS (Yao et al., 2025) addresses this need by providing a multi-level evaluation framework focusing on fine-grained reasoning, while MME-Unify (Xie et al., 2025) offers a comprehensive benchmark for unified multimodal understanding and generation models.** However, these benchmarks still have various limitations. For example, LVLM-eHub (Xu et al., 2024a) and LAMM (Yin et al., 2024) have utilized several classical datasets that are widely recognized but not sufficiently novel for current advancements, overlooking the possibility of data leakage during LVLM training. Hence, MMStar (Chen et al., 2024b) aims to solve the issue of unnecessary visual content and unintentional data leakage in LVLM training by constructing an elite vision-indispensable dataset.

**Dynamic benchmarks.** The rapid progress of LLMs has inspired benchmarks that automate evaluation processes. For example, LMExamQA (Bai et al., 2024) employs the concept of a Language-Model-as-an-Examiner to create a comprehensive and scalable evaluation framework. In addition, DYVAL (Zhu et al., 2023b) and DYVAL2 (Zhu et al., 2024) both highlight the importance of dynamic assessment, with DYVAL focusing on reasoning tasks and DYVAL2 adopting a broader psychometric approach. AutoBencher (Li et al., 2024) further expands automated benchmarking by generating novel and challenging datasets, while UNIGEN (Wu et al., 2024) and Task Me Anything (Zhang et al., 2024a) enable more customized and relevant evaluations. Recent attempts such as **Dynamic-ME** (Yang et al.) and **PuzzleBench** (Zhang et al., 2025) have extended dynamic evaluation into the multimodal domain, but they remain limited either in fine-grained controllability or task generality. By contrast, AUTODAVIS uniquely integrates an *aspect-driven hierarchical design*, a *self-validation mechanism*, and an *error-driven option adjustment module*, together with a *multi-examiner protocol*, thereby providing more reliable and flexible evaluation of LVLMs across dynamic and diverse capability dimensions.

## D  DISCUSSION

**Bias Mitigation.** Our study identifies two key sources of bias in multi-modal evaluation: **option position bias** and **self-enhancement bias** (Li et al., 2025). To address position bias, we enforce option order balancing across all samples, ensuring that correct answers are uniformly distributed among different positions. For self-enhancement bias, where a model might implicitly favor its own outputs when generating evaluation content, we introduce multi-examiner and error-driven option adjustment in the Q&A generation. This reduces undue self-preference and promotes a more objective evaluation process.

**Applicability to Domain-Specific Scenarios.** AUTODAVIS is primarily designed for general-purpose evaluation and dataset synthesis. While its core mechanisms—hierarchical aspect decomposition and error-driven refinement—are structurally adaptable to specialized domains such as medical imaging, its effectiveness depends critically on the domain alignment of underlying T2I models and LVLMs. To improve performance in expert scenarios, we outline two potential directions: (i) **Domain-Adapted Generation:** Incorporate medical-specific T2I models (e.g., GIST (McNulty et al., 2024), MINIM (Wang et al., 2025)) and LVLMs fine-tuned on clinical corpora to enhance semantic fidelity and visual realism. (ii) **Hybrid Data Sourcing:** Combine synthetic samples with limited real-world data (e.g., anonymized radiology cases) to reduce hallucinations and enhance the clinical plausibility of generated scenarios.

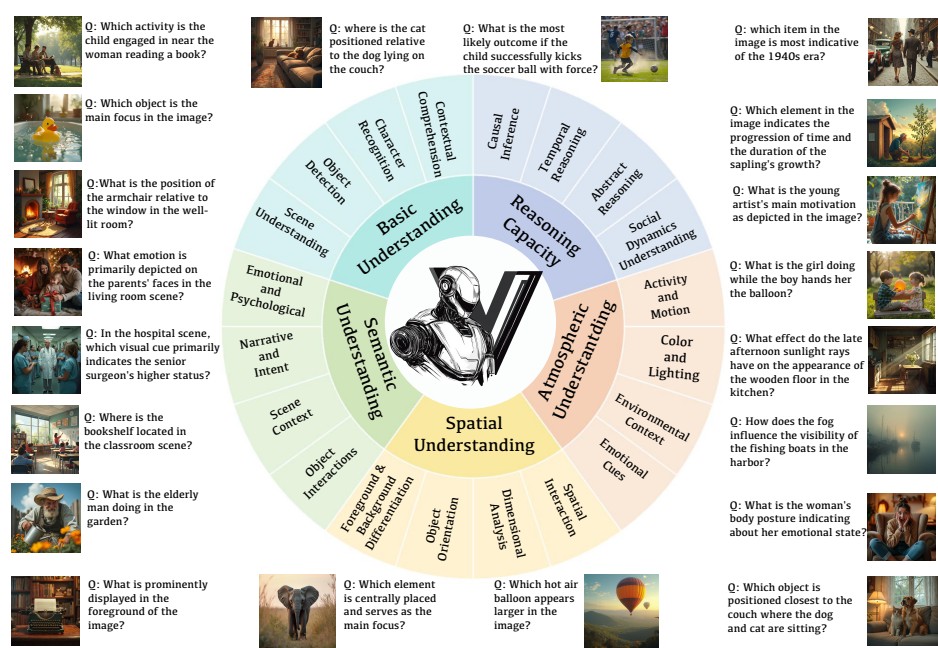

Figure 6: Five key evaluation dimensions in AUTODAVIS-Basic Understanding, Reasoning Capacity, Semantic Understanding, Spatial Understanding, and Atmospheric Understanding—and their sub-aspects, illustrated by the representative question–image pairs.

**Interpretation of Benchmark Correlation.** A key aspect of our validation is the high Spearman rank correlation (e.g., $\rho = 0.817$ with MMMU) between our protocol's rankings and those of established, human-curated benchmarks. It is crucial to interpret this result correctly: this high correlation is not an indicator of redundancy, but rather the core validation of our protocol. It demonstrates that our automated framework can reliably and accurately replicate human-curated results, proving its validity as a scalable and low-cost replacement for the expensive and time-consuming process of manual benchmark creation. The "difference" our protocol provides is not a contradictory overall ranking, but the ability to deliver these reliable results on-demand, with granular difficulty control, and immunity to static data leakage.

# E  SCALABILITY ANALYSIS OF SEMANTIC GRAPH CONSTRUCTION

A critical consideration for the practical application of AUTODAVIS is the scalability of its semantic graph construction, particularly as the number of evaluation aspects increases. This section provides a detailed analysis of the performance trade-offs to demonstrate that the framework remains computationally efficient at scale.

The framework's efficiency is rooted in its localized and stateless design. As described in Section 2.2, the semantic graph is not a persistent, global structure that grows over time. Instead, a new, temporary graph is constructed for each individual evaluation aspect and is discarded upon completion. This design ensures that the computational and memory footprint of any single task remains isolated and independent, preventing cumulative overhead as the evaluation scope expands.

To empirically validate this design, we conducted a large-scale scalability experiment. We systematically varied the number of evaluation aspects (from 1 to 10) and the number of prompts generated per aspect (from 2 to 10), measuring key performance metrics such as execution time and memory consumption. The results are summarized in Table 7.

The analysis of these results confirms the framework's high scalability across three key dimensions:

**Consistent Speed.** The average time per prompt remains remarkably stable regardless of the total number of aspects or prompts. Fluctuations are minor and show no upward trend, indicating the absence of computational bottlenecks as the workload increases.

Table 7: Scalability analysis of the semantic graph construction. We report the total time, average time per prompt, and peak/average memory increase (in MB) while varying the number of aspects and prompts per aspect. The results show consistent per-prompt performance and predictable resource usage, confirming the framework's scalability.

| Num. of Aspects | Prompts per Aspect | Total Prompts | Total Time (s) | Avg. Time per Prompt (s) | Memory Increase (MB) | | |
|---|---|---|---|---|---|---|---|
| | | | | | Peak | Average | Avg. per Prompt |
| 1 | 2 | 2 | 16.55 | 8.27 | 20.47 | 8.71 | 4.35 |
| 1 | 5 | 5 | 12.37 | 2.47 | 22.16 | 15.69 | 3.14 |
| 1 | 10 | 10 | 34.58 | 3.46 | 21.69 | 18.31 | 1.83 |
| 2 | 2 | 4 | 18.71 | 4.68 | 9.50 | 6.52 | 1.63 |
| 2 | 5 | 10 | 36.21 | 3.62 | 25.27 | 18.25 | 1.82 |
| 2 | 10 | 20 | 54.00 | 2.70 | 28.05 | 23.92 | 1.20 |
| 5 | 2 | 10 | 42.44 | 4.24 | 24.38 | 17.13 | 1.71 |
| 5 | 5 | 25 | 123.97 | 4.96 | 29.96 | 23.48 | 0.94 |
| 5 | 10 | 50 | 197.00 | 3.94 | 27.98 | 23.82 | 0.48 |
| 10 | 2 | 20 | 75.09 | 3.75 | 28.56 | 23.09 | 1.15 |
| 10 | 5 | 50 | 205.05 | 4.10 | 30.98 | 25.36 | 0.51 |
| 10 | 10 | 100 | 468.34 | 4.68 | 32.84 | 26.49 | 0.26 |

**Resource Efficiency.** The memory usage scales predictably with the total workload. While peak memory usage increases with the total number of prompts, it does not grow excessively with the number of aspects for a given task size. This underscores the efficiency of the localized, temporary graph lifecycle, which prevents undue memory overhead.

**High Parallelizability.** The independent, stateless design of each aspect's evaluation makes the entire process trivially parallelizable. While generating 100 prompts across 10 aspects takes approximately 8 minutes sequentially, this process can be significantly accelerated by running aspects in parallel, enabling efficient large-scale use of the framework.

# F  SELF-VALIDATION WITH GENERATIVE IMAGE EDITING

The rapid advancement of large vision-language models offers new possibilities beyond initial data generation. Inspired by the potential of highly-aligned models like GPT-4o, we explored whether its image editing capabilities could be used to programmatically correct defects in generated images, creating a "self-correcting loop" to enhance data quality.

## F.1  EXPERIMENTAL DESIGN

We conducted an experiment to test this hypothesis. We took a set of 144 images generated by our framework (48 from each difficulty level: Easy, Medium, and Hard) whose initial image-text alignment was imperfectly flagged by our self-validation module. Using their original text descriptions as prompts, we employed GPT-4o's image editing API to refine each image. We then recalculated their alignment scores to measure the improvement.

## F.2  RESULTS AND ANALYSIS

The results, summarized in Table 8, demonstrate the significant potential of this approach. The editing process substantially improved the average alignment scores across all difficulty levels, with the most notable gains seen in the more challenging "Hard" category (+5.2%). Critically, nearly all images in the lowest alignment bracket ($[0.0, 0.7)$) were corrected, and a large number of images were pushed into the highest quality tier ($[0.9, 1.0]$).

Table 8: Improvement in image-text alignment scores after programmatic editing with GPT-4o. The table shows the number of samples in each score bracket before and after refinement, demonstrating a clear shift toward higher quality.

| Initial Score Bracket | Easy (48 Samples) Before → After | Medium (48 Samples) Before → After | Hard (48 Samples) Before → After |
|---|---|---|---|
| $[0.0, 0.7)$ | 2 → **0** | 1 → **0** | 3 → **0** |
| $[0.7, 0.8)$ | 0 → 0 | 0 → 0 | 0 → 0 |
| $[0.8, 0.9)$ | 5 → 3 | 21 → 8 | 21 → 10 |
| $[0.9, 1.0]$ | 41 → 45 | 26 → 40 | 24 → 38 |
| **Mean Score (Before)** | 97.1% | 93.75% | 92.2% |
| **Mean Score (After)** | 99.2% | 97.9% | 97.4% |
| **Avg. Improvement** | **+2.1%** | **+4.15%** | **+5.2%** |

This experiment confirms that integrating advanced image editing models can create a powerful self-correcting loop, further enhancing the quality and reliability of a synthetically generated benchmark.

## F.3  LIMITATIONS ON SCALABILITY

Despite these promising results, we note a significant practical limitation. At the time of this research, the computational and financial cost associated with using state-of-the-art image editing models like GPT-4o is substantial. Therefore, while this approach validates the concept of a self-correcting loop, its direct application for large-scale data generation is currently cost-prohibitive. We leave the integration of more cost-effective image editing solutions as an important direction for future work.

# G CASE OF EOA

---

**Spatial Universtanding Task**

***Original Question (Before EOA):***
"Where is the black cat located relative to the table?"
*Options:*

    A) On the table

    B) Under the table

    C) Beside the table

    D) Behind the table

**Problem:** Models may exploit linguistic priors (e.g., "cats often sit under tables") without verifying the image.

**EOA Intervention Process**
*Step 1: Inject Adversarial Bias*
Feed the question and ground-truth answer ("B: Under the table") to an examiner LVLM, but deliberately mislabel it as incorrect. Generate plausible distractors that require visual verification.
*Step 2: Generate Visually Grounded Distractors*
The examiner produces revised options based on image semantics:

    A) Next to the green vase (requires checking table surface)

    C) Partially obscured by the floor lamp (requires spatial occlusion analysis)

    D) Between the table legs, facing leftward (requires pose/orientation verification)

*Final Options:*
A) Next to the green vase B) Under the table
C) Partially obscured by the floor lamp D) Between the table legs, facing leftward

---

**Basic Understanding Task**

***Original Question (Before EOA):***
"What color is the umbrella in the image?"
*Options:*

    A) Red

    B) Blue

    C) Green

    D) Yellow

**Problem:** Models might exploit dataset biases (e.g., "red umbrellas are common") without visual checking.

**After EOA Intervention**
*Step 1: Replace generic color options with visually nuanced distractors that require checking subtle differences:*

    B) Dark Red (similar to red but requires checking brightness)

    C) Orange-Red (a hue between red and orange, demanding color tone verification)

    D) Pink (light red variant, requiring saturation analysis)

*Step 2:*
*Final Adjusted Options:*
A) Red B) Dark Red
C) Orange-Red D) Pink

---

# H   DETAILS OF EXPERIMENT SETTING & RESULT

**Model Selection.** In addition to the models included in our benchmark, we initially tested popular open-source LVLMs such as LLaVA-1.6 (Liu et al., 2024a) and MiniGPT-4 (Zhu et al., 2023a). However, due to their unsatisfactory performance, they were excluded from final evaluations. Similarly, we experimented with multiple T2I models (Rombach et al., 2022; Podell et al., 2023; Betker et al., 2023), but ultimately selected FLUX-1.1-PRO for its superior generation quality. The details of models selected in our experiments are shown in Table 9.

**Alignment Evaluation.** Inspired by TIFA (Hu et al., 2023), we generated consistency tests for images across 12 aspects: *object, human, animal, food, activity, attribute, counting, color, material, spatial, location, shape, other*.

**Hyperparameter Settings.** We set the number of general aspects $n = 4$ and fine-grained aspects $m = 6$, which provided optimal diversity (Table 1). For each fine-grained aspect, $\omega = 10$ images were generated, totaling 720 images per user input across three difficulty levels. Self-validation thresholds were set to $\zeta_e = 1$ for easy, and $\zeta_m = \zeta_h = 0.8$ for medium and hard levels. This balance ensures quality control while maintaining generation efficiency.

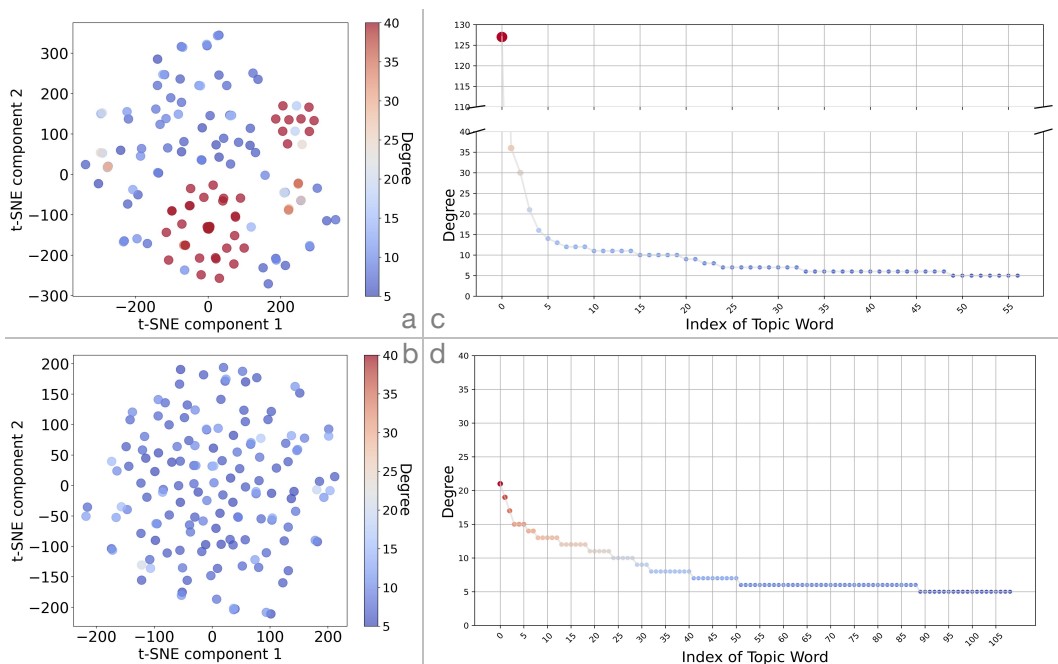

Figure 7: Visualization of image topic words. Topic words are converted into vectors using bge-large-en-v1.5 (Xiao et al., 2024), then perform dimensionality reduction via t-SNE (Van der Maaten & Hinton, 2008). Topic word distribution without semantic graph (a)(c) and with semantic graph (b)(d). It can be seen that with the semantic graph the diversity of topic words increases and the over-reliance on high-degree words is reduced.

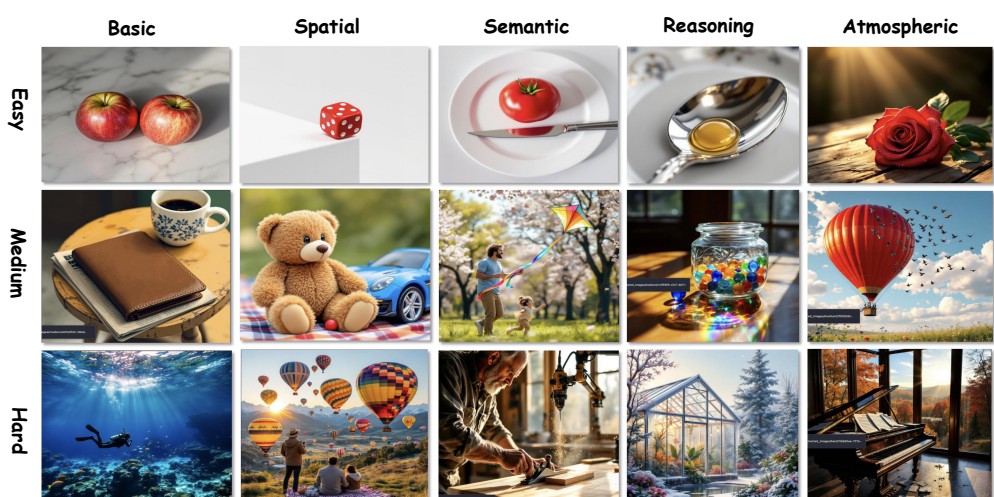

Figure 8: Images examples corresponding to different user inputs at varying difficulty levels.

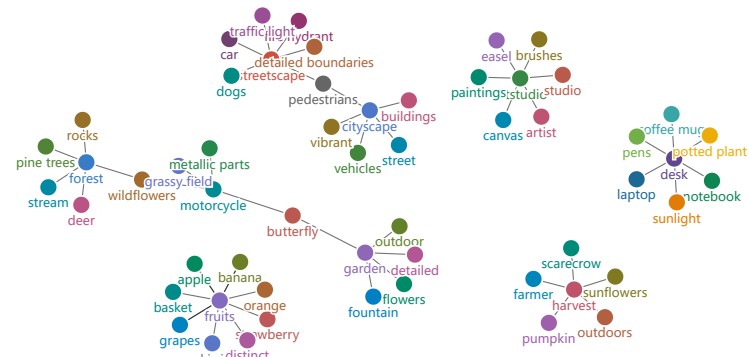

Figure 9: Topic words visualization using semantic graph under basic understanding.

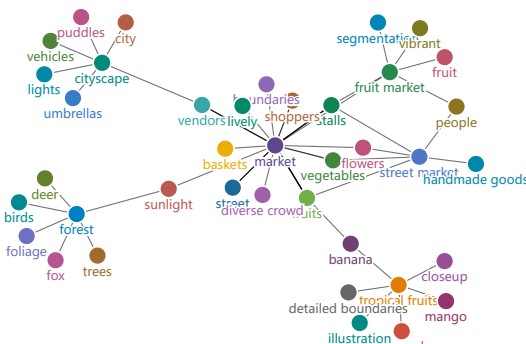

Figure 10: Topic words visualization without using semantic graph under basic understanding.

Table 9: Model names, Creators, whether it is open source, and their purpose.

| Model | Creator | Open-Source | Purpose |
|---|---|:---:|---|
| GPT-4o | OpenAI | ⊗ | Examiner&Candidate |
| GPT-4o mini | | ⊗ | Candidate |
| Gemini-1.5-Flash | Google | ⊗ | Candidate |
| Gemini-1.5-Pro | Google | ⊗ | Examiner&Candidate |
| Gemma-3-27B | Google | ⊘ | Candidate |
| Claude-3.5-sonnet | Anthropic | ⊗ | Examiner&Candidate |
| Claude-3-haiku | | ⊗ | Candidate |
| GLM-4v-Plus | Zhipu AI Inc. | ⊗ | Candidate |
| InternVL-2.5 | OpenGVLab | ⊘ | Candidate |
| Llama-3.2-90B-V | Meta AI Inc. | ⊘ | Candidate |
| Qwen2-VL-72B | Alibaba | ⊘ | Candidate |
| Flux-1.1-Pro | Black Forest Labs | ⊗ | Image generation |

Figure 11: Visualization of image descriptions in AUTODAVIS.

# I    DETAILED EXAMINER ROBUSTNESS ANALYSIS

This appendix provides the full methodology and data supporting the Examiner Robustness and Bias Mitigation analysis presented in Section 3.6.

## I.1    EXPERIMENTAL DESIGN AND METHODOLOGY

We employ three examiners comprising both proprietary models (GPT-4o, Claude-4-Sonnet) and an open-source model (Qwen3-VL-235B). These examiners generate questions together using round-robin distribution. We evaluate six models across three families: GPT family (GPT-4o, GPT-4o-mini), Claude family (Claude-4-Sonnet, Claude-3.5-Haiku), and Qwen family (Qwen3-VL-235B, Qwen2.5-VL-32B). The experiment utilizes three conditions:

- **Baseline (Multi-Examiner)**: All three examiners generate questions together; all examiners and evaluatees are scored on the same set.
- **LOEO (Leave-One-Examiner-Out)**: Excludes one examiner and generates questions using only the remaining two, isolating individual influence.
- **Single-Examiner**: Each examiner generates questions independently to measure individual examiner bias.

## I.2    FULL BASELINE AND SINGLE-EXAMINER SCORES

Table 10 presents the average scores for all evaluated models under the multi-examiner baseline condition. Table 11 provides the detailed comparison between single-examiner and multi-examiner scores, which forms the basis for neutralizing self-enhancement bias.

Table 10: Average scores under multi-examiner baseline condition.

| Model | Overall | Easy | Medium | Hard |
|---|---|---|---|---|
| GPT-4o | 0.584 | 0.86 | 0.51 | 0.38 |
| GPT-4o-mini | 0.553 | 0.78 | 0.52 | 0.36 |
| Claude-4-Sonnet | 0.620 | 0.82 | 0.58 | 0.46 |
| Claude-3.5-Haiku | 0.653 | 0.82 | 0.68 | 0.46 |
| Qwen3-VL | 0.587 | 0.86 | 0.52 | 0.38 |
| Qwen2.5-VL | 0.564 | 0.82 | 0.60 | 0.27 |

Table 11: Score comparison: single-examiner vs. multi-examiner baseline for key models.

| Model | Baseline | GPT-4o | Claude-4-Sonnet | Qwen3-VL |
|---|---|---|---|---|
| GPT-4o | 0.584 | 0.567 | 0.611 | 0.613 |
| Claude-4-Sonnet | 0.620 | 0.607 | 0.633 | 0.700 |
| Qwen3-VL | 0.587 | 0.593 | 0.597 | 0.640 |

## I.3    FULL ICC VARIANCE DECOMPOSITION

The Inter-Class Correlation (ICC) coefficient is **0.297**. This decomposition quantifies cross-family variance and agreement:

- Between-model variance ($SS_{model}$): 0.0204 (67% of total variance)

Table 12: Performance (Accuracy, %) of Qwen and InternVL models on AutoDavis.

| User Input | Difficulty | Model | | |
|---|---|---|---|---|
| | | Qwen2.5-VL | InternVL-3 | Qwen3-VL |
| BASIC | Easy | 78.0% | 82.0% | 83.0% |
| | Medium | 55.0% | 54.0% | 57.0% |
| | Hard | 41.0% | 45.0% | 35.0% |
| SPATIAL | Easy | 63.0% | 61.0% | 62.0% |
| | Medium | 36.0% | 36.0% | 27.0% |
| | Hard | 27.0% | 22.0% | 23.0% |
| SEMANTIC | Easy | 68.0% | 68.0% | 69.0% |
| | Medium | 58.0% | 60.0% | 57.0% |
| | Hard | 40.0% | 47.0% | 39.0% |
| REASONING | Easy | 53.0% | 64.0% | 58.0% |
| | Medium | 48.0% | 48.0% | 48.0% |
| | Hard | 40.0% | 40.0% | 40.0% |
| ATMOSPHERE | Easy | 59.0% | 68.0% | 66.0% |
| | Medium | 35.0% | 50.0% | 45.0% |
| | Hard | 33.0% | 42.0% | 38.0% |

- Between-examiner variance ($SS_{examiner}$): 0.0118 (39% of total variance)

- Error variance ($SS_{error}$): 0.0138 (45% of total variance)

The moderate ICC value confirms that while variance between examiners exists ($SS_{examiner}$), the majority of the variance is attributable to the true differences between the models being tested ($SS_{model}$), validating the multi-examiner approach.

## J  LATEST SOTA MODEL PERFORMANCE ON AUTODAVIS

In this section, we evaluate three new models proposed by Qwen and InternVL on AUTODAVIS, as shown in Table 12.

## K  RESOLUTION ROBUSTNESS EXPERIMENT

To investigate the robustness of LVLMs to different image resolutions, we conducted a controlled experiment using a sample of approximately 50 questions from our "Basic Understanding" dimension. We evaluated two representative models: GPT-4o and Qwen2.5-VL-32B, at three different resolutions: our baseline 1024px, 512px, and 256px.

To ensure consistency and avoid potential variability from model regeneration, we downsampled the existing 1024px baseline images to lower resolutions using bicubic interpolation (via PIL/Pillow's `Image.Resampling.BICUBIC` method) rather than regenerating images at different resolutions. This approach allows us to isolate the effect of resolution changes on model performance while maintaining identical image content across all resolution conditions.

Table 13 presents the average objective scores for both models across different resolutions and difficulty levels. The results reveal several key observations:

The experimental results demonstrate that resolution sensitivity varies significantly across models and difficulty levels:

Table 13: Model Performance Across Different Image Resolutions.

| Difficulty | Model | Resolution | | | $\Delta_{512-1024}$ | $\Delta_{256-1024}$ |
|---|---|---|---|---|---|---|
| | | 1024px | 512px | 256px | | |
| Easy | GPT-4o | 0.732 | 0.738 | 0.750 | +0.006 | +0.018 |
| | Qwen2.5-VL-32B | 0.780 | 0.727 | 0.727 | -0.053 | -0.053 |
| Medium | GPT-4o | 0.458 | 0.510 | 0.531 | +0.052 | +0.073 |
| | Qwen2.5-VL-32B | 0.479 | 0.469 | 0.469 | -0.010 | -0.010 |
| Hard | GPT-4o | 0.362 | 0.319 | 0.277 | -0.043 | -0.085 |
| | Qwen2.5-VL-32B | 0.234 | 0.234 | 0.234 | 0.000 | 0.000 |

- **GPT-4o** shows mixed sensitivity: performance improves slightly at lower resolutions for easy and medium tasks (+1.8% and +7.3% respectively when comparing 256px to 1024px), but degrades substantially for hard tasks (-8.5% at 256px compared to 1024px). This suggests that GPT-4o may benefit from reduced resolution for simpler tasks, possibly due to reduced noise or improved focus, while struggling with fine-grained details required for hard tasks.

- **Qwen2.5-VL-32B** exhibits more consistent behavior: performance remains relatively stable across resolutions for medium and hard tasks, with minimal changes ($<$1%). However, for easy tasks, there is a noticeable drop at lower resolutions (-5.3% at both 512px and 256px compared to 1024px), indicating potential information loss that affects even simpler visual understanding tasks.

- The impact of resolution reduction is most pronounced for **hard tasks**, where GPT-4o shows a clear performance degradation as resolution decreases, while Qwen2.5-VL-32B maintains consistent (though lower absolute) performance across all resolutions.

These findings highlight the importance of considering resolution robustness as a key capability metric for LVLMs, as different models exhibit varying sensitivities to image resolution depending on task complexity.

## L    HUMAN EVALUATION

### L.1    DETAILS OF HUMAN EVALUATION

The evaluation was carried out by a panel of five evaluators: three undergraduate students and two PhD students, all possessing professional English skills. Sample annotation screenshots from the human evaluation process are presented in Figure 14 and Figure 15. To ensure unbiased results, each evaluator independently assessed all samples. A sample was considered aligned if it received a majority vote (*i.e.*, more than half of the evaluators agreed on its alignment).

### L.2    HUMAN EVALUATION GUIDELINES

In this section, we outline the guidelines followed during human evaluations to ensure reliability and validity.

For **Description Generation Guideline**, the evaluators need to consider the following three points:

▷ **Alignment with Image:** The main criterion is how well the generated description reflects the visual content. Descriptions must accurately correspond to the image, avoiding vague or abstract expressions, as well as hallucination (Huang et al., 2024). Each description should provide clear, specific details that align with the image content and the defined fine-grained aspects.

▷ **Specificity and Clarity:** Ensure that descriptions are specific, directly related to the image, and free from ambiguous or overly generalized language.

▷ **Relevance to Aspects:** Assess whether the description aligns with the corresponding themes and expected content. Descriptions must clearly communicate the intended visual elements and avoid any misalignment between the image and the description.

For **Question-Answer Alignment**, there are two points that the evaluators should consider inspired by the previous study (Liu et al., 2023):

▷ **Clarity and Accuracy:** Each question must be clear, unambiguous, and directly derived from the image. The answers should correspond to observable details or logical inferences from the image, with only one correct answer for each question. There should be no irrelevant or misleading information in the questions or answers.

▷ **Consistency with Image:** Verify that both the question and answer are directly based on the image's content. The evaluation should ensure that there is a logical and clear relationship between the visual cues and the generated question-answer pair, particularly for tasks involving higher difficulty.

### L.3    RESULTS OF HUMAN EVALUATION.

We conducted comprehensive human evaluations across several key stages, including self-validation, the alignment between text and images generated by multiple text-to-image models, and manual answering of the dataset.

**Self-validation.** We randomly sampled 1,000 self-validation questions for each user input and manually annotated their accuracy. As shown in Table 14, the model achieved high accuracy all dimensions:

Table 14: Self-validation accuracy across different capability stages.

| Stage | Basic | Spatial | Semantic | Reasoning | Atmospheric |
|---|---|---|---|---|---|
| Self-validation | 97.5% | 97.8% | 98.2% | 98.0% | 97.5% |

These results confirm that the self-validation mechanism is highly reliable. Since this task involves answering a simple binary question of the image (*i.e., whether the image include this entity?*) (Hu et al., 2023).

**Text-Image Alignment Quality** To assess the alignment quality of text-to-image generation, we manually annotated 1440 images across all three models. For each model, images were sampled

across three difficulty levels, and annotators judged whether the generated image was consistent with the original description.

Table 15: Text-to-image model alignment rate across difficulty levels.

| T2I Model | Easy Level | Medium Level | Hard Level |
|---|---|---|---|
| Flux-1.1-Pro | 95.2% | 88.13% | 84.55% |
| Stable Diffusion 3.5 | 85.8% | 76.4% | 69.1% |
| DALL-E 3 | 87.5% | 73.1% | 65.5% |

As shown in Table 15, Flux-1.1-Pro consistently outperformed the other models, especially at higher difficulty levels. This superior alignment justifies our choice to use Flux-1.1-Pro as the primary image generator in our pipeline.

**Human Performance Upper Bound.** To establish a reference point for model evaluation, we conducted human answering experiments across five user input types and three difficulty levels. Each question was answered independently by two PhD students and three undergraduate students. The final scores were averaged to provide a human performance upper bound.

Table 16: Human performance across three difficulty levels.

| LEVEL | Basic | Spatial | Semantic | Reasoning | Atmospheric |
|---|---|---|---|---|---|
| Easy | 95.6% | 95.0% | 96.2% | 94.0% | 96.5% |
| Medium | 91.2% | 90.0% | 92.1% | 89.8% | 92.0% |
| Hard | 88.%5 | 87.5% | 89.5% | 87.2% | 90.0% |

The results in Table 16 demonstrate a consistent decline in accuracy as task difficulty increases. Nonetheless, human accuracy remains relatively high across all tasks, serving as a meaningful benchmark for evaluating model performance.

## M  COST ANALYSIS OF AUTODAVIS

In constructing the AUTODAVIS dataset, we employed four core models: GPT-4o, Gemini 1.5-Pro, Claude 3.5-Sonnet, and the text-to-image generator Flux-1.1-Pro. The cost of using each model is summarized below, including both input and output rates.

Table 17: Cost of different models for output and input.

| Model | Output | Input |
|---|---|---|
| **GPT-4o** | $10/1M tokens | $2.5/1M tokens |
| **Gemini 1.5-Pro** | $5/1M tokens | $1.25/1K tokens |
| **Claude 3.5-Sonnet** | $15/1M tokens | $3/1M tokens |
| **Flux-1.1-Pro** | / | $0.06/image |

Despite involving multiple high-capability models, the overall construction cost of AUTODAVIS remains extremely low—only **$51.82** in total—thanks to our automatic pipeline. This demonstrates a significant cost advantage compared to traditional manual dataset creation methods, which often require substantial human labor and time investment.

Table 18: Cost calculation summary for different stages. Claude-3.5 is Claude-3.5-Sonnet and Gemini-1.5 is Gemini-1.5-Pro.

| Stage | Models Involved | Total Cost |
|---|---|---|
| **Aspect Generation** | GPT-4o | $0.006 |
| **Image Description** | GPT-4o, Claude-3.5, Gemini-1.5 | $1.98 |
| **Alignment Validation** | GPT-4o | $3.60 |
| **Image Generation** | Flux-1.1-Pro | $43.20 |
| **Question Generation** | GPT-4o, Claude-3.5, Gemini-1.5 | $1.728 |
| **Error-Driven Option Adjust** | GPT-4o, Claude-3.5, Gemini-1.5 | $1.3068 |
| **Total Cost** | | $51.82 |

# N IMPACT OF IMAGE STYLE

Despite the absence of explicit style constraints during image generation, the model's inherent stochasticity resulted in images with diverse visual styles. To systematically examine the influence of image style on model performance, we selected two representative styles: oil painting and cartoon. We still used the Flux-1.1-Pro model to generate styled images. Specifically, we sampled **60 descriptions** for each difficulty level from the full image-description dataset, and appended a stylistic prompt (*e.g., "in the style of oil painting"*) to the end of each text.

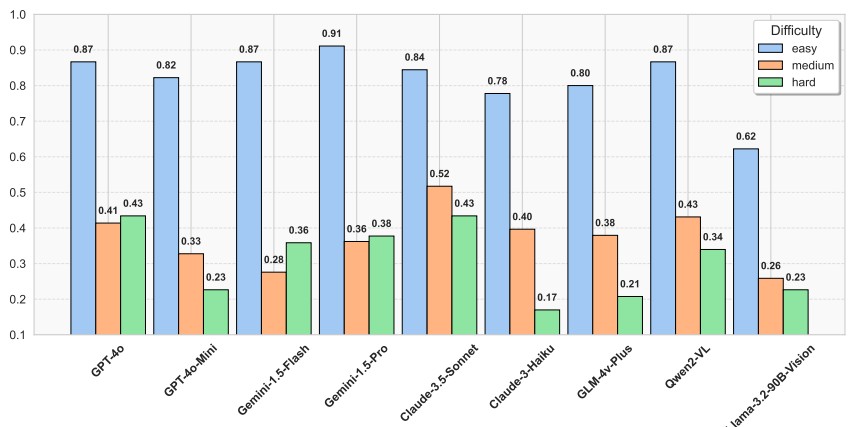

Figure 12: Model performance under cartoon style images in basic understanding.

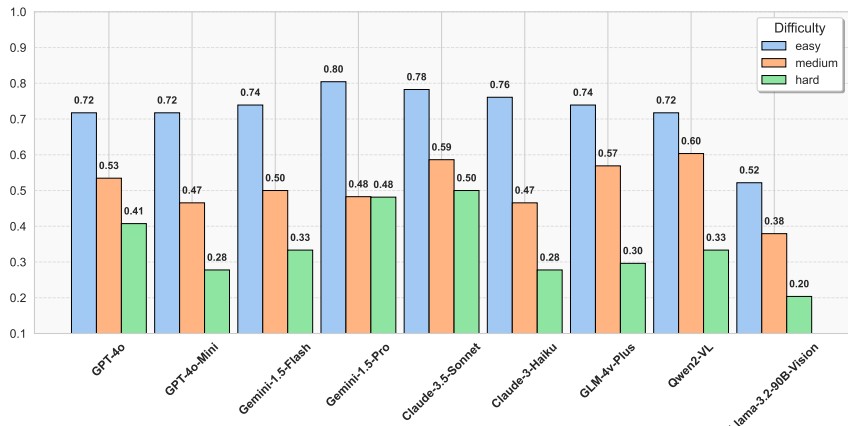

Figure 13: Model performance under oil painting style images in basic understanding.

The Figure 12 and Figure 13 show consistent trends across styles: while cartoon-style images yield slightly higher scores than oil paintings, all models exhibit similar performance declines with increasing difficulty and maintain stable relative rankings.

## N.1 DETAILS OF TRAINING DATA GENERATION

**Fine-tuning Details.** We performed full-parameter supervised fine-tuning (Ouyang et al., 2022) using the LLaMA Factory framework (Zheng et al., 2024) on a single NVIDIA GeForce RTX 4090 GPU with 24GB VRAM for less than 30 minutes. For training, we sampled **40** examples from each difficulty level in the full dataset, then selected only those that achieved perfect scores in self-validation, resulting in **82** high-quality training samples. For evaluation, we sampled another **20** examples from each difficulty level from the remaining data.

## O    DETAILS OF USER INPUT

In this section, we provide a comprehensive overview of the levels at which we categorize user inputs based on linguistic aspects. Our goal is to offer a comprehensive and broad representation of user requirements for LVLMs. However, as it is challenging to exhaustively cover every aspect, we base our categorization on aspects derived from the literature (Li et al., 2023b; Meng et al., 2024; Liu et al., 2025). These aspects are considered representative and comprehensive examples of the capabilities of LVLMs and other aspects like in (Chen et al., 2024b; Xu et al., 2024a) can be handled in a similar manner, without requiring additional fine-tuning or adjustments, as our framework is highly extensible, allowing users to propose their own aspects as needed.

### O.1    BASIC UNDERSTANDING

**Definition and Goal:** Basic Understanding refers to the recognition and identification of individual objects, characters, and scenes within an image. The goal is to accurately detect and label relevant elements, providing a foundation for more advanced tasks such as object tracking and scene interpretation (Wu et al., 2013; Xue et al., 2018).

**Requirement.** This task demands the ability to detect specific objects and differentiate between various types of objects. Additionally, it involves understanding the broader context of the scene and identifying real-life settings to enable accurate interpretation of the image's overall content.

### O.2    SPATIAL UNDERSTANDING

**Definition and Goal.** Spatial Understanding refers to the interpretation of the spatial arrangement and positioning of objects within an image (Cai et al., 2024; Guo et al., 2024). The goal is to comprehend both two-dimensional and three-dimensional relationships, determining which objects are in the foreground or background, assessing their relative sizes and orientations, and understanding how they are positioned within the scene.

**Requirement.** This task demands the ability to perceive depth, estimate distances between objects, and analyze how objects interact within the physical space of the image, providing a more accurate understanding of the spatial structure and context.

### O.3    SEMANTIC UNDERSTANDING

**Definition and Goal.** Semantic Understanding involves interpreting the higher-level meaning and relationships within an image (Meng et al., 2024). The goal is to move beyond simple object identification to understand the roles and interactions between objects, such as recognizing that a person is riding a bike or that two people are engaged in conversation. This level of understanding aims to capture the context and intent behind the scene, identifying how elements relate to each other to form a coherent narrative or message.

**Requirement.** This task requires discerning the interactions and relationships between objects, understanding their roles within the scene, and interpreting the overall context to accurately derive the narrative or intended message conveyed by the image.

### O.4    ATMOSPHERIC UNDERSTANDING

**Definition and Goal.** Atmospheric Understanding focuses on grasping the mood, tone, and emotional ambiance conveyed by an image. The goal is to interpret not just what is depicted or how elements are arranged, but also how the scene feels and the emotional resonance it conveys to the viewer. For instance, an image of children laughing under warm sunlight in a lush park combines their expressions, bright colors, and soft lighting to create a joyful and carefree atmosphere.

**Requirement.** This task requires the ability to capture and interpret subtle emotional cues and tonal qualities of the scene, distinguishing the overall mood and emotional impact of the image from more analytical aspects like semantic or spatial understanding.

## O.5 REASONING CAPACITY

**Definition and Goal.** Reasoning Capacity involves interpreting and analyzing the relationships and logical connections between different elements within an image (Zhou et al., 2024a; You et al., 2023). The goal is to infer potential outcomes, understand causal relationships, and make predictions about what might happen next based on visual cues. For example, if a person is holding an umbrella and the sky is dark, reasoning capacity would suggest that it might rain soon. This level also includes understanding abstract relationships, such as social dynamics or the intent behind actions, and making judgments about what is likely or possible given the visual information.

**Requirement.** This task requires the ability to analyze logical connections between elements, infer outcomes, and understand causal relationships, as well as to interpret abstract concepts and make predictions based on the visual context.

## P DETAILS OF DIFFICULTY GRADING

This section describes in detail the difficulty levels for the pictures (subsection P.1) and questions (subsection P.2) used in prompts respectively. The following is the instruction guiding the examiner model to generate image descriptions and questions of varying difficulty levels. For details, see Appendix T. Moreover, the difficulty levels are defined manually and are not fixed, allowing for flexible adjustments based on the user's specific requirements.

### P.1 IMAGE DESCRIPTION

**Easy difficulty.** Focus on a single subject but with minor additional complexity to test precision in details or context. Establish baseline capability with subtle challenges, such as fine texture, simple interactions, or slight variations in lighting. A single object or entity with some basic contextual details or additional features, placed against a minimally distracting background. Test finer details (e.g., texture, reflection) while still keeping the composition simple and clear. For examples: "A perfectly polished red apple with a small leaf attached, placed on a slightly reflective white surface."

**Medium difficulty.** Scenes involve multiple elements or interactive settings that require nuanced spatial arrangement and accurate relationships between objects. Evaluate the ability to generate cohesive, moderately dynamic scenes with layered realism and a stronger sense of depth. Description Requirements: Include multiple objects interacting naturally in a believable environment, with more intricate details and subtle light or shadow effects. Incorporate realistic environmental elements and ensure spatial coherence, emphasizing interactions and secondary details (e.g., shadows, water splashes). For examples: "A golden retriever running on a sandy beach, splashing water as it chases a bright orange ball, with distant waves and a partly cloudy sky."

**Hard difficulty.** Scenes incorporate high complexity with multiple interdependent elements, challenging perspectives, or dynamic and intricate lighting or textural effects. Push the limits of rendering capability to handle advanced relationships, environmental effects, and challenging compositions. The scene must include 3-4 main elements or a combination of dynamic features, such as motion, light interplay, or atmospheric conditions, while maintaining clarity and logic. Highlight challenges such as complex reflections, dynamic light, or multi-element interactions, ensuring visual harmony and detailed textures. For examples: "A raindrop-streaked window reflecting the interior of a cozy room, with a black cat sitting on the windowsill and a glowing city skyline visible through the glass, all under the warm hues of a sunset."

### P.2 QUESTION

**Easy difficulty.** Focus on questions that require identifying simple, prominent, and explicit details within the image. These questions should be straightforward, relying solely on basic observation without the need for inference or interpretation. For example, you might ask about the color of a specific object, the presence of a single item, or the shape of an easily recognizable feature. The key is to keep the questions direct and simple, ensuring that the answer is obvious and immediately visible in the image.

**Medium difficulty.** Design questions that necessitate a moderate level of observation and inference. These questions should involve understanding relationships between elements, recognizing interactions, or identifying less prominent features that are still clear but not immediately obvious. Examples could include questions about the relative position of objects, identifying an action taking place, or understanding the context of a scene. The goal is to require some level of thought beyond basic observation, challenging the model to understand the scene's composition or narrative without being overly complex.

**Hard difficulty.** Create questions that require the model to notice and interpret more detailed aspects of the image. These questions should involve recognizing multiple elements working together, understanding more complex interactions, or identifying details that are present but not immediately obvious. For example, you might ask about the positioning of objects relative to each other in a more crowded scene, subtle changes in lighting or color that affect the appearance of objects, or identifying an element that is not the main focus but is still visible in the background. The aim is to challenge the model to go beyond surface-level details, but without making the task too abstract or overly difficult.

# Q NOTATIONS & ALGORITHM

Table 19: Notation used in the methodology description.

| Notation | Description |
|---|---|
| $q$ | User input specifying the evaluation target or focus. |
| $n$ | Number of *general aspects*. |
| $\{A_1^{(g)}, A_2^{(g)}, \ldots, A_n^{(g)}\}$ | Set of $n$ general aspects, each representing a high-level capability dimension. |
| $A_i^{(g)}$ | The $i$-th general aspect. |
| $m$ | Number of *fine-grained aspects* under each general aspect. |
| $\{A_{i1}^{(f)}, A_{i2}^{(f)}, \ldots, A_{im}^{(f)}\}$ | Set of $m$ fine-grained aspects under the $i$-th general aspect. |
| $A_{ij}^{(f)}$ | The $j$-th fine-grained aspect under the $i$-th general aspect. |
| $\mathcal{M}_v$ | The LVLM used for aspect generation, description generation, and self-validation. |
| $\mathcal{A}$ | The complete set of generated aspects, i.e., $\bigcup_{i=1}^{n} \{A_i^{(g)}\} \cup \{A_{ij}^{(f)}\}$. |
| $|\mathcal{A}|$ | The cardinality of $\mathcal{A}$, with $|\mathcal{A}| = m \times n$. |
| $d \in \{\text{easy}, \text{medium}, \text{hard}\}$ | Difficulty level for image description generation. |
| $\omega$ | Number of image descriptions generated per aspect at each difficulty level. |
| $\{\mathcal{T}_{ij1}^d, \ldots, \mathcal{T}_{ij\omega}^d\}$ | Set of $\omega$ image descriptions for the fine-grained aspect $A_{ij}^{(f)}$ at difficulty $d$. |
| $\mathcal{T}_{ij}^d$ | A specific image description for aspect $A_{ij}^{(f)}$ at difficulty $d$. |
| $t_e$ | Topic word selected at iteration $e$. |
| $K_e = \{k_{e1}, k_{e2}, \ldots, k_{ec}\}$ | Set of $|c|$ keywords related to the topic word $t_e$ at iteration $e$. |
| $G_e = (V_e, E_e)$ | Semantic graph at iteration $e$, whose nodes are topic/keyword sets and edges are semantic relationships. |
| $S_e$ | Combination of $V_{e-1}$, $t_e$, and $K_e$ before applying the exclusion mechanism. |
| $V_e$ | Refined node set (topic words/keywords) after excluding high-degree nodes in iteration $e$. |
| $f(S_e)$ | A function that determines how many top-degree nodes to exclude from $S_e$. |
| $\{\mathcal{I}_{ij}^d\}$ | Generated images corresponding to the image description(s) $\mathcal{T}_{ij}^d$. |
| $\Phi_{ij}^d = \{\phi_{ij1}^d, \ldots, \phi_{ijp}^d\}$ | Set of $p$ simple VQA-style questions for self-validation of $\mathcal{I}_{ij}^d$. |
| $p$ | Number of questions for each image's self-validation. |
| $\mathcal{F}(\mathcal{I}_{ij}^d, \mathcal{T}_{ij}^d, \Phi_{ij}^d)$ | Self-validation function returning the alignment score $S_{ij}^d$. |
| $S_{ij}^d$ | Alignment score, i.e., the proportion of correctly answered questions among $\Phi_{ij}^d$. |
| $\zeta$ | Threshold for deciding whether to regenerate an image or keep it with recorded errors. |
| $\mathcal{E}_{ij}^d$ | Noted error(s) if the image partially misaligns with the description ($S_{ij}^d < 1$). |

---

**Algorithm 1** Diverse Description Generation Strategy

**Require:** User input $q$, model $\mathcal{M}_v$, initial set of topic words and keywords $V_0$, exclusion function $f(S_e)$, number of iterations $N$

**Ensure:** Set of diverse image descriptions $\{\mathcal{T}(1), \mathcal{T}(2), \ldots, \mathcal{T}(N)\}$

1: Initialize iteration counter $e \leftarrow 1$
2: Initialize the set of topic words and keywords $V_1 \leftarrow V_0$
3: **while** $e \leq N$ **do**
4:     Select a topic word $t_e$ and related keywords $K_e = \{k_{e1}, \ldots, k_{ec}\}$
5:     Form node set $S_e \leftarrow V_{e-1} \cup \{t_e\} \cup K_e$
6:     Construct edge set $E_e$ based on semantic relationships
7:     Compute exclusion set: $f(S_e) = \underset{V' \subseteq S_e, |V'| = e}{\arg\max} \sum_{v \in V'} \deg(v)$
8:     Update node set: $V_e \leftarrow S_e \setminus f(S_e)$
9:     Set difficulty level $d$ and number of prompts $\omega$
10:     Generate descriptions:

$$\mathcal{T}(e) \leftarrow \bigcup_{k=1}^{\omega} \left\{ \mathcal{T}_{ijk}^{de} \right\} = \mathcal{M}_v(q, A_{ij}^{(f)}, \mathcal{D}_{ij}, V_e, d)$$

11:     Increment iteration counter: $e \leftarrow e + 1$
12: **end while**
    **return** $\{\mathcal{T}(1), \mathcal{T}(2), \ldots, \mathcal{T}(N)\}$

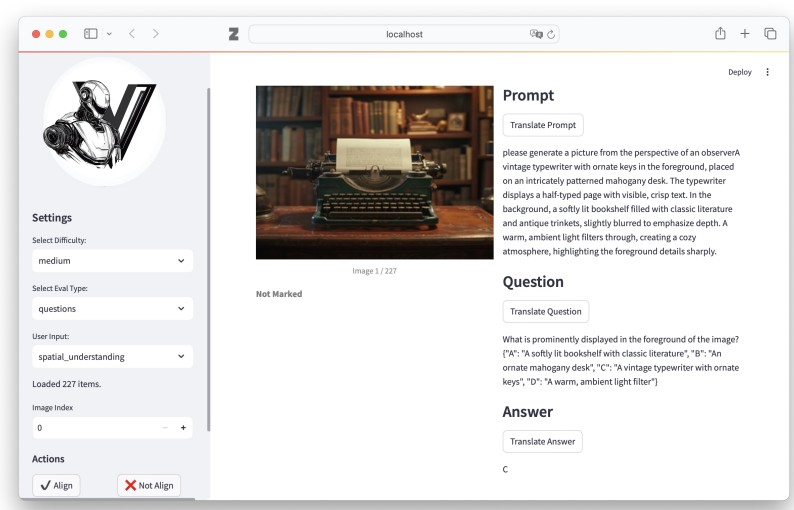

Figure 14: Screenshot of Human Evaluation (Example 1).

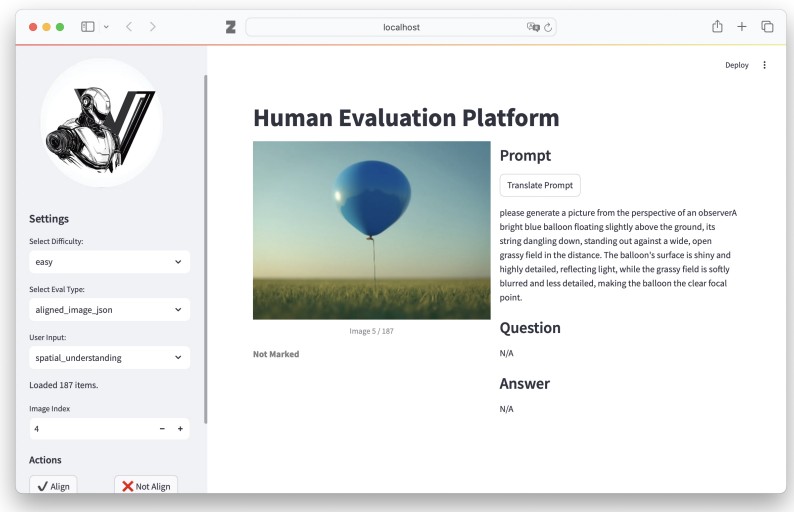

Figure 15: Screenshot of Human Evaluation (Example 2).

# R  ERROR STUDY

Through extensive experimental analysis, we have categorized the common problems encountered by LVLMs in VQA tasks into two main types: *image comprehension errors* and *image reasoning errors*. Regarding the first category, LVLMs often fail to truly understand the details in an image. For instance, in Figure 16, the model failed to accurately identify the *performer's attire*. In Figure 18, from the model's explanation, it is evident that the focus of its understanding was placed on the text rather than the image. Option B mentions *a swimming pool*, option C references *a helipad*, and option D refers to *a health center*—none of which are present in the image. In Figure 20, the model failed to correctly identify the *magician's facial expression* as a smile. Therefore, it cannot be associated with calmness or scheming; instead, it represents a confident smile. These errors demonstrate the model's inadequate comprehension of image details.

Image reasoning errors occur when models accurately perceive the image content but falter in their reasoning process, leading to incorrect answers. For instance, in Figure 17, the model correctly

identified the *blue whale* and *the small boat* in the image. However, based on its explanation, it relied more on its prior knowledge rather than the image itself, leading to an incorrect answer. In Figure 19, model accurately recognized the excitement and joy on the little girl's face but misinterpreted the intent. From the image, it is clear that the woman is holding a complete ice cream cone and is about to hand it to the girl, indicating that the girl desires the ice cream.

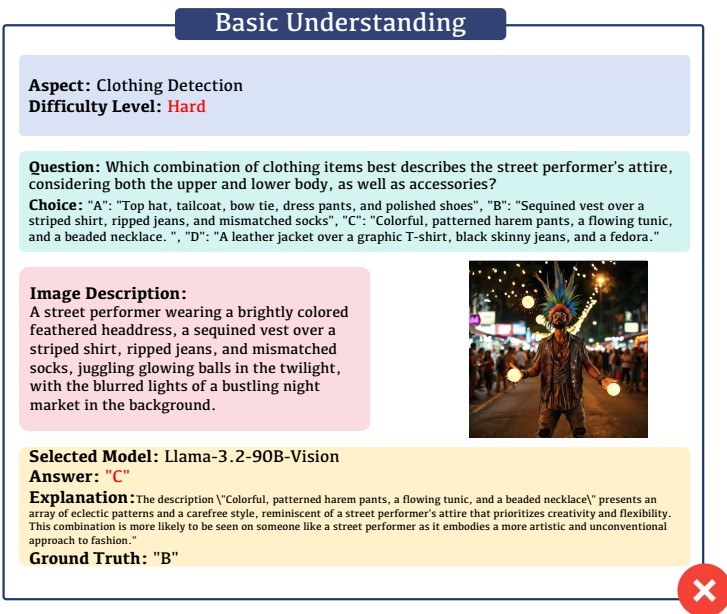

Figure 16: Error study of Llama-3.2-90B-Vision under basic understanding.

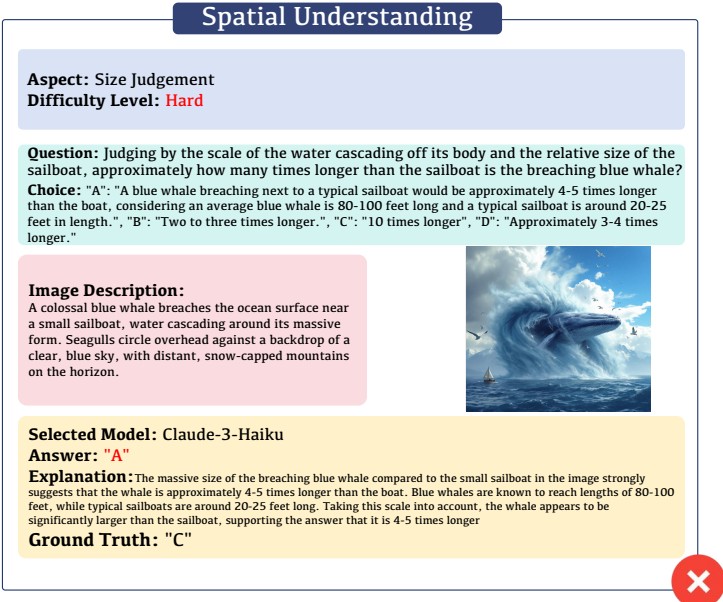

Figure 17: Error study of Claude-3-Haiku under spatial understanding.

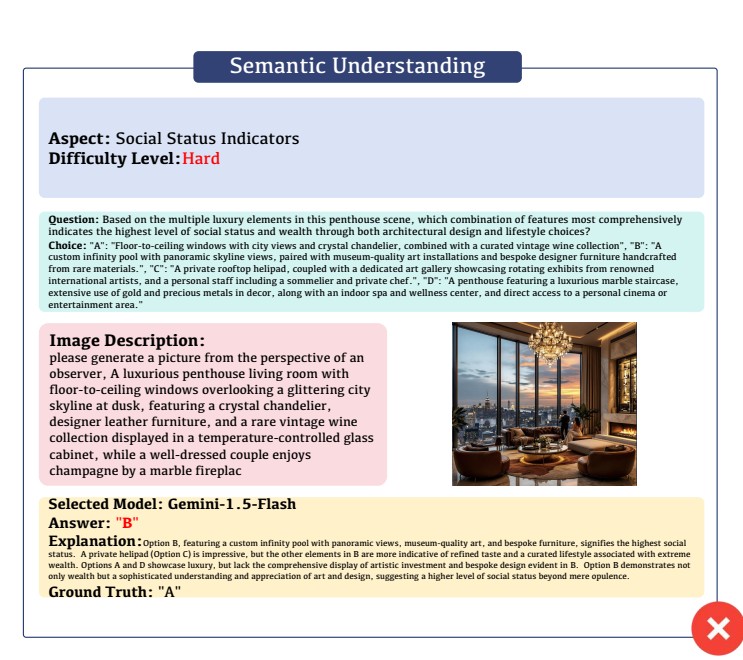

Figure 18: Error study of Gemini-1.5-Flash under semantic understanding.

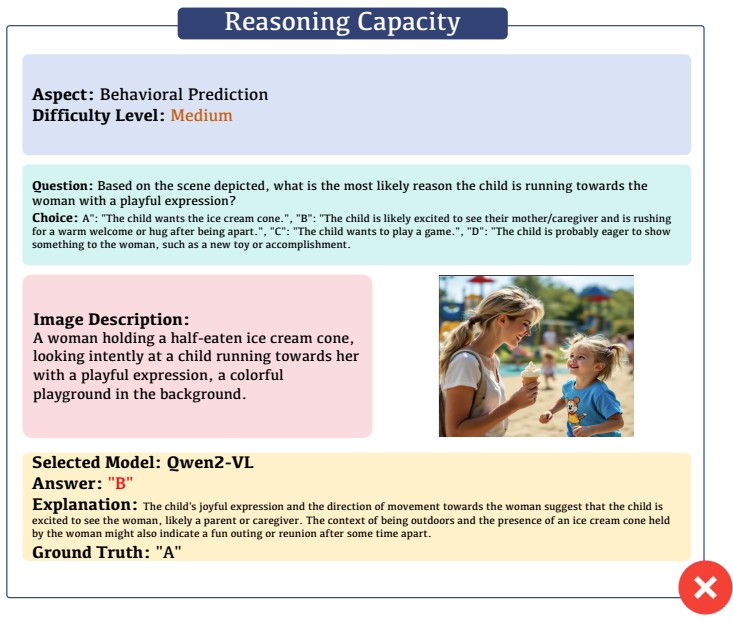

Figure 19: Error study of Qwen2-VL under reasoning capacity.

**Atmospheric Understanding**

**Aspect:** Facial Expression Analysis
**Difficulty Level:** Hard

**Question:** In this street performance scene, which combination of facial expressions most accurately captures the complex emotional dynamics between the magician and their audience, considering the interplay of lighting and atmosphere?
**Choice:** "A": "The magician displays confident concentration while the audience shows a mix of skepticism and wonder, with some faces betraying analytical observation and others showing childlike amazement", "B": "The magician exhibits a playful smirk while the audience members show genuine surprise and delight, with open-mouthed expressions and raised eyebrows illuminated by the warm street lighting.", "C": "The magician's intense focus contrasts with the audience's varied reactions: captivated awe, nervous anticipation, and curious amusement.", "D": "The magician maintains a calm and mysterious demeanor, while the audience displays a mix of curiosity and cautious intrigue, highlighted by the shifting street lighting."

**Image Description:**
A street magician, face lit by the warm glow of a flickering streetlamp, performs a card trick for a small, captivated crowd. Their faces, a mixture of awe, suspicion, and amusement, are illuminated by the dim light, reflecting in the magician's polished, silver-tipped cane leaning against his worn suitcase. The cobblestone street glistens from a recent rain.

**Selected Model:** GLM-4v-Plus
**Answer:** "D"
**Explanation:** The magician maintains a calm and mysterious demeanor, while the audience displays a mix of curiosity and cautious intrigue, highlighted by the shifting street lighting. The dim, moody lighting and the rainy atmosphere suggest a sense of mystery and intrigue. The magician's posture and the audience's engagement with the performance support this interpretation.
**Ground Truth:** "C"

Figure 20: Error study of GLM-4v-Plus under atmospheric understanding.

## S    CASE STUDY

Figure 21: Case Study 1.

Figure 22: Case Study 2.

## Basic Understanding

**Aspect:** Object Overlap Detection
**Difficulty Level:** Hard

**Question:** In the image of the Siamese cat partially obscured by a sheer, patterned curtain, what is the position of the cat relative to the curtain and the window?

**Choice:** "A": "The Siamese cat is likely sitting on the windowsill, partially visible through the sheer curtain. Its face and front paws are probably pressed against the window glass, with the curtain draping over part of its body.", "B": "The cat is behind the curtain, not in contact with the window.", "C": "The cat is in front of the curtain but not touching the window.", "D": "The cat is situated behind the curtain, with the window likely behind both.

**Image Description:**
A Siamese cat partially obscured by a sheer, patterned curtain, gazing out a window at a bustling street scene with colorful taxis and pedestrians. Rain streaks the window, distorting the view slightly.

**Selected Model:** Gemini-1.5-Pro
**Answer:** "A"
**Ground Truth:** "B"

Figure 23: Case Study 3.

## Spatial Understanding

**Aspect:** Object Layering
**Difficulty Level:** Easy

**Question:** What object is placed underneath the shiny green apple?
**Choice:** "A": "A circular wooden coaster", "B": "A plain white ceramic plate", "C": "A checkered tablecloth", "D": "A small black napkin."

**Image Description:**
A shiny green apple with a stem and a single leaf rests on a circular wooden coaster.

**Selected Model:** Claude-3-Haiku
**Answer:** "A"
**Ground Truth:** "A"

Figure 24: Case Study 4.

## Spatial Understanding

**Aspect:** Middle-ground Recognition
**Difficulty Level:** Medium

**Question:** What object is just positioned behind the woman in the red dress in the crosswalk?
**Choice:** "A": "A yellow taxi cab", "B": "A traffic light pole", "C": "A dark-colored SUV", "D": "A city bus

**Image Description:**
A woman in a red dress stands in the middle of a crosswalk, holding a half-eaten ice cream cone. A yellow taxi cab waits behind her, and a bus is approaching from the opposite direction. Buildings line the street, and a pedestrian walks on the sidewalk in the distance.

**Selected Model:** GPT-4o mini
**Answer:** "B"
**Ground Truth:** "A"

Figure 25: Case Study 5.

## Semantic Understanding

**Aspect:** Dynamic Action Identification
**Difficulty Level:** Easy

**Question:** What dynamic action is the ballerina performing in the image?
**Choice:** "A": "Executing a graceful grand jetu00e9 (split leap) through the air", "B": "Performing an arabesque.", "C": "Spinning in a twirl", "D": "The ballerina is leaping with one leg extended forward and the other backward, known as a saut de chat

**Image Description:**
A graceful ballerina performing a simple twirl on a polished wooden floor, her pink tutu creating a gentle circular motion, against a plain white wall.

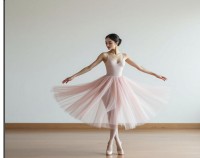

**Selected Model:** GPT-4o
**Answer:** "C"
**Ground Truth:** "C"

Figure 26: Case Study 6.

**Semantic Understanding**

**Aspect:** Personality Traits
**Difficulty Level:** Medium

**Question:** Based on the woman's expression and posture in the image, which personality trait is she most likely exhibiting?
**Choice:** "A": "Timid or shy", "B": "Timid or shy""Confident", "C": "Serene", "D": "Curious"

**Image Description:**
A woman in a sharp, tailored business suit stands with arms crossed, a briefcase at her feet, against a backdrop of a bustling city street with blurred yellow taxis and pedestrians. She wears a serious expression and her posture is rigid.

**Selected Model:** GLM-4v-Plus
**Answer:** "B"
**Ground Truth:** "B"

Figure 27: Case Study 7.

**Semantic Understanding**

**Aspect:** Cultural Context
**Difficulty Level:** Hard

**Question:** Based on the depiction of the dragon, costumes, and lanterns, which specific regional Chinese cultural influence is most prominently reflected in this Lantern Festival dragon dance celebration?
**Choice:** "A": "Southern Chinese traditions, particularly from Guangdong and Hong Kong regions, are evident through the ornate dragon design, vibrant colors, and elaborate paper lanterns typical of Cantonese festival celebrations. ", "B": "Northern Chinese folk traditions, with the dragon's long, serpentine body and the prevalence of red lanterns suggesting a focus on prosperity and good fortune. ", "C": "Hakka Chinese traditions, evident in the dragon's robust build and the use of specific Hakka colors and patterns in the costumes and lanterns. ", "D": "Eastern Chinese traditions, particularly from Jiangsu and Zhejiang provinces, are evident in the dragon's intricate design and the use of silk and embroidery in the costumes and lanterns

**Image Description:**
An intricately detailed Chinese dragon dance celebration during a vibrant Lantern Festival, with several performers in colorful traditional costumes maneuvering the dragon puppet, surrounded by glowing lanterns floating in the night sky, casting dynamic shadows and reflections on the ground.

**Selected Model:** Claude-3.5-Sonnet
**Answer:** "A"
**Ground Truth:** "B"

Figure 28: Case Study 8.

Figure 29: Case Study 9.

Figure 30: Case Study 10.

## Reasoning Capacity

**Aspect:** Irony and Contradictions
**Difficulty Level:** Hard

**Question:** What element in the scene highlights the irony between purity and decay in the image?
**Choice:** "A": "The pristine white wedding dress amidst the broken machinery.", "B": "The contrast between fresh white lilies blooming among rusted, deteriorating metal structures.", "C": "A delicate, untouched bridal veil draped over a decaying, graffiti-covered wall.", "D": "A clean, white dove perched on a corroded, neglected iron gate.

**Image Description:**
A pristine white wedding dress hanging elegantly in a dilapidated, abandoned factory, with broken machinery covered in rust and cobwebs surrounding it, while sunlight streams through shattered windows creating dramatic light patterns on the dress and dusty floor, and a single red rose lies wilted beneath it.

**Selected Model:** Llama-3.2-90B-Vision
**Answer:** "A"
**Ground Truth:** "A"

Figure 31: Case Study 11.

## Atmospheric Understanding

**Aspect:** Color Psychology
**Difficulty Level:** Easy

**Question:** Which color combination in this image creates a sense of contrast and visual interest through the use of complementary colors?
**Choice:** "A": "Purple and yellow, as these colors are opposite each other on the color wheel and create a strong contrast when used together.", "B": "Red and green, as these colors sit opposite each other on the color wheel and create strong visual contrast when paired together.", "C": "Orange and blue.", "D": "Yellow banana against blue background"

**Image Description:**
A single, ripe banana, primarily yellow with hints of green at the tip, lying on a vibrant blue background.

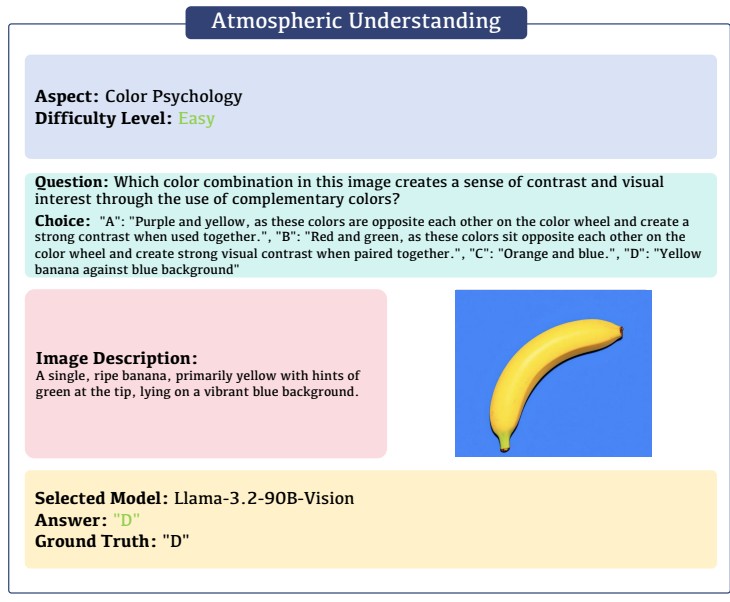

**Selected Model:** Llama-3.2-90B-Vision
**Answer:** "D"
**Ground Truth:** "D"

Figure 32: Case Study 12.

**Atmospheric Understanding**

**Aspect:** Body Language Interpretation
**Difficulty Level:** Hard

**Question:** Based on the body positioning and spatial relationship between the dancers, what complex emotional dynamic is most likely being conveyed through their choreography?
**Choice:** "A": "The dancers likely portray a dynamic of yearning and dependence, with the central figure reaching out for connection while the others remain grounded, perhaps representing unattainable desires or unreciprocated feelings.", "B": "The choreography appears to convey inner conflict and tension, with the contrasting movements between dancers suggesting emotional struggle and resistance. The spatial arrangement likely depicts power dynamics and psychological distance rather than unity.", "C": "A moment of collective support and elevation, where the supporting dancers' reaching gestures and the lead dancer's leap create a narrative of shared triumph and mutual trust", "D": "The choreography likely portrays harmonious partnership, with synchronized movements emphasizing connection and mutual understanding between the dancers.

**Image Description:**
Three ballet dancers performing in a grand theater, the lead dancer executing a perfect leap while two supporting dancers reach toward her with graceful, elongated arms, their shadows dancing across the polished wooden stage floor as stage lights create dramatic rim lighting on their flowing costumes

**Selected Model:** Gemini-1.5-Flash
**Answer:** "C"
**Ground Truth:** "C"

Figure 33: Case Study 13.

## T    PROMPT

---

**Aspect Generation**

You are an AI assistant specializing in designing prompts to test Large Vision-Language Models (LVLMs). Your task is to create meticulously aspect_count fined-grained aspects that evaluate LVLMs basic understanding of images.

Large Vision-Language Models are AI systems capable of understanding and analyzing images. Testing these models across various competencies is crucial for assessing their performance, limitations, and potential biases. The aspects you create will be used to challenge and evaluate LVMs.

1. Basic Understanding: This involves recognizing and identifying individual objects, characters, and scenes within an image. It includes tasks like detecting the presence of specific items (e.g., cars, trees, people), distinguishing between different types of objects, and understanding the general context of the scene (e.g., a park, a city street). The goal is to accurately label all relevant elements in the image, providing a foundation for more advanced analysis.

2. The aspects you generate must test the understanding of a single image, not multiple images, e.g. multiple perspectives. 3. Come up with 4 general aspects according to the basic understanding.

4. Then Create 6 fined-grained aspects within the basic understanding for each general aspect, do not go beyond. You can consider the definition of the basic understanding above.

5. List the aspects without using numbered lists.

6. Let's think step by step.

Please strictly respond in the following format:

General Aspect: [Aspect]

Fined-grained Aspect: [Aspect]

Introduction: [Introduction]

---

**Image Description Generation**

You are an AI assistant tasked with converting user inputs and their descriptions into suitable prompts for a text-to-image model. These prompts will generate images to test the capabilities of large vision language models (LVLMs).

Large Vision Language Models (LVLMs) are AI systems proficient in interpreting and analyzing images. Evaluating these models across different competencies is essential to understanding their performance, limitations, and potential biases. The prompts you create will be used to generate images through text-to-image models, which will then be used to challenge and evaluate LVLMs.

1. Carefully follow the given aspect: aspect, its introduction: introduction.

2. Generate a suitable prompt based on the provided aspect and introduction for the diffusion model to create an image. Ensure that the prompt is composed of simple phrases, avoiding overly complex descriptions, and is clear enough. If you deem the description irrelevant to the test content, do not generate a related prompt. 3. Consider including elements that might be particularly challenging for LVMs, such as unusual combinations, abstract concepts, or subtle details.

...

## Image Description Generation-2

... 4. Details about different difficulty levels:

- Easy Difficulty - Characteristics: Focus on a single subject but with minor additional complexity to test precision in details or context.
- Purpose: Establish baseline capability with subtle challenges, such as fine texture, simple interactions, or slight variations in lighting. - Description Requirements: A single object or entity with some basic contextual details or additional features, placed against a minimally distracting background. - Examples:- "A perfectly polished red apple with a small leaf attached, placed on a slightly reflective white surface." - "A blue balloon with a slightly wrinkled texture, tied to a thin white string, floating against a pale sky with soft clouds." - Key Points: Test finer details (e.g., texture, reflection) while still keeping the composition simple and clear.
- Medium Difficulty - Characteristics: Scenes involve multiple elements or interactive settings that require nuanced spatial arrangement and accurate relationships between objects. - Purpose: Evaluate the ability to generate cohesive, moderately dynamic scenes with layered realism and a stronger sense of depth. - Description Requirements: Include multiple objects interacting naturally in a believable environment, with more intricate details and subtle light or shadow effects. - Examples: - "A steaming cup of coffee on a wooden table, with a half-eaten croissant beside it, morning sunlight casting soft shadows through a lace curtain in the background." - "A golden retriever running on a sandy beach, splashing water as it chases a bright orange ball, with distant waves and a partly cloudy sky."
- Key Points: Incorporate realistic environmental elements and ensure spatial coherence, emphasizing interactions and secondary details (e.g., shadows, water splashes).
- Hard Difficulty - Characteristics: Scenes incorporate high complexity with multiple inter-dependent elements, challenging perspectives, or dynamic and intricate lighting or textural effects. - Purpose: Push the limits of rendering capability to handle advanced relationships, environmental effects, and challenging compositions. - Description Requirements: The scene must include 3-4 main elements or a combination of dynamic features, such as motion, light interplay, or atmospheric conditions, while maintaining clarity and logic. - Examples: - "A raindrop-streaked window reflecting the interior of a cozy room, with a black cat sitting on the windowsill and a glowing city skyline visible through the glass, all under the warm hues of a sunset." - "A knight in shining armor standing on a cliff edge, overlooking a stormy sea, with bolts of lightning illuminating the dark clouds and waves crashing against jagged rocks below." - Key Points: Highlight challenges such as complex reflections, dynamic light, or multi-element interactions, ensuring visual harmony and detailed textures.

5. Provide one overarching topic word that encapsulates the essence of your description.
6. List 4-6 key words that are closely related to your description and crucial for understanding the image.
7. Avoid using the following words in your new description: used_words_str
8. The required difficulty level is: level
9. Please use clear and accurate words, clear logic flow, do not use too abstract words. The length of the generated sentences needs to be consistent with the examples.
10. Just output the image description used to generate the image, don't mention anything else
Please strictly respond in the following format:
Aspect: aspect
Prompt: [Your detailed image description]
Topic word: [One word that captures the essence of the description]
Key word: [Word1, Word2, Word3,...]

---
**Align Question Generation**

Given the image descriptions:description, generate six questions (True/False or Multiple choice) with only one correct choice that verifies if the image description is correct.
Classify each concept into a type (object, human, animal, food, activity, attribute, counting, color, material, spatial, location, shape, other), and then generate a question for each type.
Here's some examples:
'''Description: A man posing for a selfie in a jacket and bow tie. Entities: man, selfie, jacket, bow tie Activities: posing Colors: Counting: Other attributes: Questions and answers are below: About man (human): Q: is this a man? Choices: yes, no A: yes Q: who is posing for a selfie? Choices: man, woman, boy, girl A: man About selfie (activity): Q: is the man taking a selfie? Choices: yes, no A: yes Q: what type of photo is the person taking? Choices: selfie, landscape, sports, portrait A: selfie About jacket (object): Q: is the man wearing a jacket? Choices: yes, no A: yes Q: what is the man wearing? Choices:jacket, t-shirt, tuxedo, swearter A: jacket About bow tie (object): Q: is the man wearing a bow tie? Choices: yes, no A: yes Q: is the man wearing a bow tie or a neck tie? Choices: bow tie, neck tie, cravat, bolo tie A: bow tie About posing (activity): Q: is the man posing for the selfie? Choices: yes, no A: yes Q: what is the man doing besides taking the selfie? Choices: posing, waving, nothing, shaking A: posing
Description: A horse and several cows feed on hay. Entities: horse, cows, hay Activities: feed on Colors: Counting: several Other attributes: Questions and answers are below: About horse (animal): Q: is there a horse? Choices: yes, no A: yes About cows (animal): Q: are there cows? Choices: yes, no A: yes About hay (object): Q: is there hay? Choices: yes, no A: yes Q: what is the horse and cows feeding on? Choices: hay, grass, leaves, twigs A: hay About feed on (activity): Q: are the horse and cows feeding on hay? Choices: yes, no A: yes About several (counting): Q: are there several cows? Choices: yes, no A: yes '''
And finally respond in the following format:
{{
"caption": "description",
"question": "Your question here",
"choices": [yes or no], "answer": "give your correct answer",
"element_type": "Type of the element",
"element": "Element name"
}},
{{
"caption": "description",
"question": "Your question here",
"choices": [yes or no],
"answer": "give your correct answer",
"element_type": "Type of the element",
"element": "Element name"
}},
...

---

---
**Align Answer Generation**

Given the image below, answer the questions: question from the choice: choices based on the image.
And directly give the answer.
{{
"answer":"yes or no"
}}

---

2754
2755
2756
2757
2758
2759
2760
2761
2762
2763
2764
2765
2766
2767
2768
2769
2770
2771
2772
2773
2774
2775
2776
2777
2778
2779
2780
2781
2782
2783
2784
2785
2786
2787
2788
2789
2790
2791
2792
2793
2794
2795
2796
2797
2798
2799
2800
2801
2802
2803
2804
2805
2806
2807

---

### Question Generation-1

You are an AI assistant tasked with converting user inputs and their descriptions into suitable questions to test the Large Vision Model's (LVM) abilities in given aspects.

Large Vision Models (LVMs) are AI systems proficient in interpreting and analyzing images. Evaluating these models across different competencies is essential to understanding their performance, limitations, and potential biases. We will provide you with a prompt to generate an image, which will create a specific image. You can then formulate questions about this image based on the prompt. The questions you create will be used to challenge and evaluate LVMs based on generated images.

1. Carefully analyze the given aspect and its Introduction: Aspect:aspect.

2. Generate a suitable question based on the provided image description to test the LVM's ability in the given aspect.

3. We categorize the difficulty of questions into easy, medium, and hard:

- Easy Difficulty:Focus on questions that require the identification of simple, prominent, and explicit details within the image. These questions should be straightforward, relying solely on basic observation without the need for inference or interpretation. For example, you might ask about the color of a specific object, the presence of a single item, or the shape of an easily recognizable feature. The key is to keep the questions direct and simple, ensuring that the answer is obvious and immediately visible in the image.

- Medium Difficulty:Design questions that necessitate a moderate level of observation and inference. These questions should involve understanding relationships between elements, recognizing interactions, or identifying less prominent features that are still clear but not immediately obvious. Examples could include questions about the relative position of objects, identifying an action taking place, or understanding the context of a scene. The goal is to require some level of thought beyond basic observation, challenging the model to understand the scene's composition or narrative without being overly complex.

- Hard Difficulty:Create questions that require the model to notice and interpret more detailed aspects of the image. These questions should involve recognizing multiple elements working together, understanding more complex interactions, or identifying details that are present but not immediately obvious. For example, you might ask about the positioning of objects relative to each other in a more crowded scene, subtle changes in lighting or color that affect the appearance of objects, or identifying an element that is not the main focus but still visible in the background. The aim is to challenge the model to go beyond surface-level details, but without making the task too abstract or overly difficult.

4. Avoid using overly complicated language or details unrelated to the image in the questions.

5. When generating problems of different difficulty, please combine the current specific aspect.

6. Due to potential discrepancies in image generation, we have detected the following errors: elements. Please avoid referencing these elements in your questions. If the prompt for generating the image does not describe in detail what the specific looks like, please do not ask related questions. For example, if the prompt mentions a forest with glowing plants but does not specify how many there are, please do not ask a question about counting the number of glowing plants.

7. The required difficulty level is: level

8. Please generate a multiple-choice question, which is four-option single-choice question.

9. The answers in the options need to be differentiated to a certain extent. There cannot be a situation where multiple options meet the requirements of the question. There can only be one answer that meets the question.

Image generation prompt: prompt
Aspect: aspect
Please directly output the generated question in the following JSON format: "question": "[your question]", "options": "A": "[Option A]", "B": "[Option B]", "C": "[Option C]", "D": "[Option D]" "reference_answer": "A or B or C or D" Without any other information and remember only one option in the reference answer.

**Option Adjustment**

You are an ai assistant tasked with answering questions based on the given picture description without given the image.
- Answer the questions based on your knowledge.
- Please note that some incorrect answers are provided below. You must not make the same mistakes
- Your answer needs to be semantically distinct from the given incorrect answer.
- Don't say you can't see the image, just answer based on your knowledge.
- Don't generate overly lengthy answers, keep them concise and to the point.
- The answer you generate needs to be factually different from the given incorrect answer.
- Try to use straightforward words instead of being too abstract or vague.

question:question
wrong answers:wrong_answer

Directly give you answer, don't add thinkings or other information.

**Answer Generation**

In order to test your ability with pictures, we have a question about aspect area. Please answer based on your knowledge in this area and your understanding of pictures. Given the image below, answer the questions: question based on the image. Please give the final answer strictly follow the format [[A]] (Strictly add [[ ]] to the choice, and the content in the brackets should be the choice such as A, B, C, D) and provide a brief explanation of your answer. Directly output your answer in this format and give a brief explanation. If you cannot read the picture, just answer based on your text ability.

## U    BROADER IMPACT

AUTODAVIS presents a dynamic and automatic framework for evaluating large vision-language models (LVLMs), offering significant implications for the broader multimodal AI community. By leveraging synthetic data generation and flexible evaluation pipelines, AUTODAVIS contributes to multiple dimensions of research and deployment, while also raising important ethical considerations.

On the positive side, AUTODAVIS enables more precise and scalable benchmarking of LVLMs, allowing researchers to systematically identify weaknesses in capabilities such as spatial reasoning, fine-grained inference, or visual comprehension. This transparency promotes targeted model improvement and inspires innovation in multimodal representation learning. Furthermore, by automating the generation of evaluation data through text-to-image models, AUTODAVIS reduces dependence on costly, manually curated datasets, thereby facilitating a more efficient and reproducible research paradigm. The dynamic and customizable nature of the framework allows domain-specific adaptation, making it suitable for application. In addition, the framework's automation and multi-examiner design help mitigate common sources of human bias, offering a step toward more fair and accountable model assessment.

However, the use of synthetic data generated by large language and image models introduces potential risks. If such models encode social, cultural, or distributional biases, these may be inadvertently amplified during evaluation or when the generated data is reused for model training, potentially reinforcing harmful patterns. In particular, feedback loops may emerge if models are trained on outputs of similar models, leading to homogenization or entrenchment of errors and stereotypes. This concern underscores the need for careful validation of generated content and continual monitoring of bias propagation when deploying synthetic data at scale.

In summary, while AUTODAVIS holds promise for accelerating the evaluation and development of LVLMs, it also highlights the importance of responsible data generation practices. Future work should explore bias mitigating mechanisms within synthetic pipelines and establish guidelines for safe reuse of generated content. Through such efforts, AUTODAVIS can serve not only as a tool for technical progress, but also as a catalyst for more transparent and socially responsible AI systems.

## V    THE USE OF LARGE LANGUAGE MODELS (LLMS)

LLMs were utilized in two primary capacities in this work: as core components of our proposed AUTODAVIS framework and as tools to assist in the preparation of this manuscript.

**As Core Technical Components.** Our framework fundamentally relies on the generative and reasoning capabilities of state-of-the-art LLMs. Various models, including GPT-4o, Claude-3.5-Sonnet, and Gemini-1.5-Pro, serve as essential functional modules throughout the pipeline. They are responsible for key stages of the automated benchmark generation process, such as hierarchical aspect generation, image description, question formulation, and error-driven option adjustment. The specific models used in each stage are detailed in their respective sections.

**As a Writing Assistance Tool.** In addition to their technical role within the framework, we used LLMs (primarily GPT-4) as an aid in the manuscript preparation process. This assistance was confined to language enhancement tasks, including improving grammar, refining sentence structure for clarity, and rephrasing text for better readability. All conceptual ideas, experimental designs, analyses, and conclusions presented in this paper are exclusively the work of the authors. The authors have reviewed, edited, and take full responsibility for the final content of this publication.

