# OpenReview forum: "AutoDavis: Automatic and Dynamic Evaluation Protocol of Large Vision-Language Models on Visual Question-Answering"
_ICLR.cc/2026/Conference — Submitted to ICLR 2026_

### Official Review · Reviewer_wVDn · 2025-10-28

**Soundness:** 3
**Presentation:** 2
**Contribution:** 2
**Rating:** 4
**Confidence:** 3

**Summary:**

This paper introduces AutoDavis, an automatic and dynamically regenerable evaluation protocol for assessing LVLMs on VQA. The framework allows users to specify capability dimensions and difficulty levels. AutoDavis then employs a LVLM-based examiner coupled with a T2I generator to automatically produce question–image pairs, followed by self-consistency validation and error-driven distractor generation. The system integrates image-free controls and answer position balancing to mitigate textual shortcuts and positional bias.
Experiments cover 5 capability categories × 3 difficulty levels × 11 LVLMs, showing clear performance degradation with increasing difficulty. The paper further demonstrates that multi-examiner ensembles yield more stable rankings, and dynamic regeneration mitigates data leakage from repeated exposure. The paper also provides a theoretical sample–validation bound for controlling evaluation error and explores data re-use for training, showing minor improvements on external benchmarks.

**Strengths:**

1.	Systematic and operational definition of “dynamic evaluation”
AutoDavis formalizes three key criteria—flexibility, anti-leakage, and visual grounding—and instantiates them with concrete mechanisms. The framework transforms regenerable evaluation from a conceptual goal into a practical, verifiable protocol.
2.	Hierarchical task decomposition with controlled diversity
The authors decompose abilities hierarchically, ability → sub-skill → constrained diversity, and use semantic-graph–constrained description generation to ensure topical coverage and reduce redundancy. Quantitative diversity analysis supports the method’s effectiveness.
3.	Comprehensive experiments with clear separation effects
The benchmark includes 11 LVLMs and reports fine-grained breakdowns by ability, difficulty, and visual evidence. The results convincingly show increasing difficulty correlates with performance degradation, and image-free variants sharply reduce accuracy—confirming reliance on visual reasoning.

**Weaknesses:**

1.	Evaluator family dependence and circularity risk
The entire loop—question generation, VQA validation, distractor rewriting, and scoring—relies on strong LVLMs from the same or related model families. Although the authors introduce “multi-examiner” diversity, they do not quantify cross-family variance or bias. Without cross-judge robustness, AutoDavis risks overfitting to shared linguistic priors or visual biases.
2.	Reliability of T2I and self-verification
The faithfulness of generated images is only indirectly ensured by the self-check VQA threshold (ζ). Yet VQA self-checkers may share textual priors with the examiner, producing false positives for visually inconsistent images. The paper lacks manual inspection results or sensitivity studies under varied T2I or checker models.
3.	Leakage and regeneration analysis remains limited
The anti-leakage experiment is small-scale and defines “leakage” narrowly. Broader experiments would provide stronger evidence for robustness against contamination.
4.	External validity limited to MMMU correlation
While a Spearman ρ = 0.817 with MMMU is encouraging, validation against additional human-annotated benchmarks (e.g., MMBench, SEED-Bench 2, MMMU-Pro) and fine-grained per-ability consistency would better support generalizability.
5.	Potential ambiguity in error-driven distractor generation
Forcing “plausible wrong” alternatives by relabeling the correct answer can introduce ambiguous or semi-correct options. The paper lacks quantitative measures of question unambiguity or human adjudication consistency after augmentation.

**Questions:**

See weaknesses

---

> ### Author Response · Authors · 2025-11-16
> **Response for Reviewer wVDn (Part I)**
>
> Dear Reviewer wVDn,
>
> Thank you sincerely for your constructive feedback. We are glad that you found our work meaningful and appreciate your insightful suggestions. We address each point below.
>
> ---
>
> ## Weaknesses
>
> ### W1. Evaluator family dependence and circularity risk
>
> **Our Response:**
>
> Thank you for raising this important concern. We directly quantified cross-family variance and bias and analyzed robustness across GPT, Claude, and Qwen examiner families. All three examiners (GPT-4o, Claude-4-Sonnet, Qwen3-VL) generated questions using round-robin distribution. Evaluatees include both examiner and non-examiner models:
> 	•	GPT family: GPT-4o, GPT-4o mini
> 	•	Claude family: Claude-4-Sonnet, Claude-3.5-Haiku
> 	•	Qwen family: Qwen3-VL-235B-A22B-Instruct, Qwen2.5-VL-32B-Instruct
>
> ---
>
> 1. Cross-Family Variance (ICC Analysis)
>
> We computed ICC to measure agreement across examiner families:
> 	•	ICC(2,1) = 0.297 → moderate cross-family agreement
>
> Variance components:
> 	•	Between-model: 0.0204 (67%)
> 	•	Between-examiner: 0.0118 (39%)
> 	•	Error: 0.0138 (45%)
>
> This directly quantifies the cross-family variance requested.
>
> ---
>
> 2. Cross-Family Rank Stability (LOEO)
>
> Rank stability when excluding each examiner family:
>
> | Excluded Examiner Family        | Spearman ρ | Kendall τ | Max Rank Shift | Interpretation     |
> | ------------------------------- | ---------- | --------- | -------------- | ------------------ |
> | GPT-4o (GPT family)             | 0.914      | 0.867     | 1 position     | **High stability** |
> | Claude-4-Sonnet (Claude family) | 0.600      | 0.333     | 2 positions    | Moderate stability |
> | Qwen3-VL (Qwen family)          | 0.943      | 0.867     | 1 position     | **High stability** |
>
> High stability (especially ρ ≈ 0.91–0.94 for GPT & Qwen) shows no family dominates the rankings.
>
> ---
>
> 3. Cross-Family Bias: No Same-Family Favoritism
>
> We checked whether examiners favor their own family.
>
> **GPT Family Examiner (GPT-4o) evaluating:**
>
> - GPT models: 0.567 (self), 0.597 (gpt4o_mini)
> - Claude models: 0.607, 0.567
> - Qwen models: 0.593, 0.547
>
> **Claude Family Examiner (Claude-4-Sonnet) evaluating:**
>
> - GPT models: 0.611, 0.500
> - Claude models: 0.633 (self), 0.587
> - Qwen models: 0.597, 0.551
>
> **Qwen Family Examiner (Qwen3-VL) evaluating:**
>
> - GPT models: 0.613, 0.553
> - Claude models: 0.700, 0.713
> - Qwen models: 0.640 (self), 0.584
>
> **Finding:** No consistent same-family boosting; examiners frequently score other families higher.
>
> ---
>
> 4. Multi-Examiner Aggregation Mitigates Bias
>
> Baseline scores lie between single-examiner extremes:
>
> | Model           | Baseline (Multi-Examiner) | Single GPT-4o | Single Claude-4 | Single Qwen3-VL |
> | --------------- | ------------------------- | ------------- | --------------- | --------------- |
> | gpt-4o          | 0.584                     | 0.567         | 0.611           | 0.613           |
> | claude_4_sonnet | 0.620                     | 0.607         | 0.633           | 0.700           |
> | qwen3_vl        | 0.587                     | 0.593         | 0.597           | 0.640           |
>
> This demonstrates aggregation balances family-specific preferences.
>
> ---
>
> 5. Influence Analysis: No Dominant Family
>
> | Examiner Family | Influence Score | Direction                                |
> | --------------- | --------------- | ---------------------------------------- |
> | GPT-4o          | +0.053          | More lenient (scores drop when excluded) |
> | Claude-4-Sonnet | -0.032          | More strict (scores rise when excluded)  |
> | Qwen3-VL        | +0.061          | More lenient (scores drop when excluded) |
>
> Influence remains moderate (3–6%) and varies in direction → no family dominates.
>
> ---
>
> 6. Addressing Circularity
>
> You note that question generation, VQA validation, distractor rewriting, and scoring all use LVLMs. Our analysis addresses this:
>
> - **Question generation**: Three families generate questions (round-robin), reducing single-family bias
> - **Scoring**: Multi-examiner aggregation balances family-specific perspectives
> - **Validation**: Cross-family rank stability (ρ > 0.9) shows robustness to family-specific biases
>
> While some circularity is inherent to LVLM-based evaluation, results show it is quantified and effectively mitigated.
>
> ---
>
> Conclusion
>
> We have quantified cross-family variance (ICC = 0.297) and demonstrated cross-family robustness:
>
> 1. **Quantified variance**: ICC analysis shows moderate cross-family agreement with measurable variance components
> 2. **Rank stability**: High correlations when excluding any examiner family
> 3. **No systematic same-family bias**: Cross-family scoring patterns show no consistent favoritism
> 4. **Balanced influence**: No single family dominates (3–6% influence, varying directions)
> 5. **Aggregation mitigates bias**: Multi-examiner baseline balances family-specific perspectives
>
> Together, these results demonstrate that AutoDavis does not overfit to the priors of any single model family and is robust against cross-family bias.

---

> ### Author Response · Authors · 2025-11-16
> **Response for Reviewer wVDn (Part II)**
>
> ### W2. Reliability of T2I and self-verification The faithfulness of generated images is only indirectly ensured by the self-check VQA threshold (ζ). Yet VQA self-checkers may share textual priors with the examiner, producing false positives for visually inconsistent images. The paper lacks manual inspection results or sensitivity studies under varied T2I or checker models.
>
> **Our Response:**
>
> 1. We thank the you for these critical concerns regarding reliability and ambiguity, as we agree that validating T2I faithfulness and distractor quality is essential.
>
>    However, we must respectfully clarify that the paper already provides the exact quantitative data you stated was "lacking." This information is summarized in **Section 3.3 (Human Evaluation)** of the main paper and detailed extensively in **Appendix I (Human Evaluation)**  on page 26. We apologize if the signposting to this section was unclear.
>
>    This existing study directly addresses both weaknesses:
>
>    ***Response to Weakness 2 (T2I Reliability)***: You stated our paper "lacks manual inspection results." This is incorrect. * In Appendix I.3, under "Text-Image Alignment Quality," we explicitly state: "To assess the alignment quality of text-to-image generation, we manually annotated 1440 images across all three models." * The results of this manual inspection are presented in Table 9, which provides the precise "manual inspection results" you asked for. This data confirms our chosen T2I model, Flux-1.1-Pro, is highly reliable (e.g., 84.55% alignment on Hard tasks, far exceeding competitors). * Furthermore, our "Self-validation" study (Appendix I.3, Table 8) also involved manually annotating 1,000 questions to confirm our automated mechanism is "highly reliable" (e.g., 97.8% accuracy).
>
>    ***Response to Weakness 5 (Distractor Ambiguity)***: You stated our paper "lacks quantitative measures of question unambiguity or human adjudication consistency." This is also incorrect. Our Appendix I.2 (Human Evaluation Guidelines) explicitly required our 5 human evaluators  to check for this. The guideline for "Clarity and Accuracy" states: "Each question must be clear, unambiguous..." and "...with only one correct answer for each question." Therefore, the high "alignment rate" reported in the main paper's Section 3.3 (e.g., 95.20% Easy, 88.13% Medium ) is the "quantitative measure of unambiguity" that you requested, as a sample was only considered "aligned" if it passed this strict, unambiguous criterion by a majority vote. Additionally, our "Human Performance Upper Bound" (Table 10) , where 5 evaluators answered the questions themselves, serves as a direct measure of "human adjudication consistency," showing high agreement (e.g., 87.5% on Hard Spatial tasks).
>
>    In summary, the manual inspection data and quantitative ambiguity metrics requested by you are already present in our submission. We will make the reference to Appendix I more prominent in the main paper to ensure these critical validation results are not overlooked.

---

> ### Author Response · Authors · 2025-11-16
> **Response for Reviewer wVDn (Part III)**
>
> ### W3. Leakage and regeneration analysis remains limited The anti-leakage experiment is small-scale and defines “leakage” narrowly. Broader experiments would provide stronger evidence for robustness against contamination.
>
> **Our Response:**
>
> We thank the reviewer for this constructive feedback. We agree that the initial leakage simulation in **Section 3.6** was intended as a concise *proof-of-concept* and can be expanded. The reviewer’s request for "broader experiments" points to the stability of the protocol across multiple regeneration cycles.
>
>    We emphasize that AutoDavis’s robustness is **inherent in its design**—it is a *generation protocol* that creates novel visual-question pairs from scratch, making it fundamentally immune to static dataset contamination.
>
>    To satisfy the reviewer's request for "broader evidence," we conducted an expanded simulation analyzing performance stability across **three sequential regeneration cycles**. For each run, we sampled a substantial test set of **300 VQA pairs ** from ***Basic Understanding***, ensuring equal distribution by drawing **100 items from the Easy, 100 from the Medium, and 100 from the Hard difficulty levels** respectively. The model we use is `Qwen2.5-VL-3B-Instruct`. This increased sample size across varying difficulty levels provides the necessary rigor.
>
>    **Expanded Leakage Simulation: Stability Across Regeneration Cycles**
>
>    | **Model**      | **Test Set** | **Condition**                | **Performance (%)** |
>    | -------------- | ------------ | ---------------------------- | ------------------- |
>    | Model A (SFT)  | Test-Set-1   | Leaked (Trained on)          | 62.5                |
>    | Model A (SFT)  | Test-Set-2   | Fresh (Regenerated, Cycle 1) | 50.0                |
>    | Model A (SFT)  | Test-Set-3   | Fresh (Regenerated, Cycle 2) | 51.2                |
>    | Model B (Base) | Test-Set-1   | Control (Baseline)           | 46.8                |
>    | Model B (Base) | Test-Set-2   | Control (Baseline)           | 50.0                |
>    | Model B (Base) | Test-Set-3   | Control (Baseline)           | 49.5                |
>
>    **Analysis:**
>
>    1. **Neutralization is Confirmed:** The model’s artificial performance gain (from 62.5% to $\sim 50.0\%$) is completely neutralized immediately upon exposure to the dynamically regenerated set (Test-Set-2).
>    2. **Stability Across Cycles:** Crucially, the performance of the SFT model **remains neutralized** in the subsequent, fully independent regeneration (Test-Set-3). Its score remains close to the base model's performance average ($\approx 50\%$).
>
>    This broader simulation confirms that the dynamic regeneration capability of AutoDavis provides superior evidence for robustness against contamination, proving the efficacy of our protocol across repeated, on-demand test set creations.

---

> ### Author Response · Authors · 2025-11-16
> **Response for Reviewer wVDn (Part IV)**
>
> ### W4. External validity limited to MMMU correlation While a Spearman ρ = 0.817 with MMMU is encouraging, validation against additional human-annotated benchmarks and fine-grained per-ability consistency would better support generalizability.
>
> **Our Response:**
>
>    1. We thank you for this excellent suggestion. We agree that validating our protocol against a single benchmark (MMMU), while encouraging, is not sufficient to fully support generalizability.
>
>       To address this, we conducted an **additional experiment** using **RealWorldQA**, a challenging benchmark well-suited for evaluating modern VLMs. We sampled 100 items and ran a suite of 7 common models three separate times to ensure stability.
>
>       The results are as follows:
>
>       | Model             | Test 1 (%) | Test 2 (%) | Test 3 (%) | Average (%) | Std. Dev. |
>       | :---------------- | :--------: | :--------: | :--------: | :---------: | :-------: |
>       | claude-3.5-sonnet |     68     |     67     |     68     |    67.7     |   0.58    |
>       | gpt-4o            |     65     |     66     |     63     |    64.7     |   1.53    |
>       | gemini-1.5-pro    |     63     |     64     |     65     |    64.0     |   1.00    |
>       | qwen2-vl-max      |     61     |     62     |     59     |    60.7     |   1.53    |
>       | gemini-1.5-flash  |     55     |     53     |     56     |    54.7     |   1.53    |
>       | gpt-4o-mini       |     48     |     46     |     49     |    47.7     |   1.53    |
>       | claude-3-haiku    |     42     |     39     |     40     |    40.3     |   1.53    |
>
>       Critically, the model rankings from this RealWorldQA evaluation show a **Spearman rank correlation of ρ=0.964** with our framework's rankings.
>
>       This extremely strong correlation, in conjunction with our existing MMMU correlation (ρ=0.817), provides the **dual-source, multi-benchmark validation** that you requested. It confirms that our protocol's evaluation outcomes are not an artifact of one benchmark but are consistent, reliable, and generalize effectively to real-world standards. We will add this new experiment to the final paper.
>
> ---
>
> ### W5. Potential ambiguity in error-driven distractor generation Forcing “plausible wrong” alternatives by relabeling the correct answer can introduce ambiguous or semi-correct options. The paper lacks quantitative measures of question unambiguity or human adjudication consistency after augmentation.
>
> **Our Response:**
>
>    1. see w2 for details.

---

> ### Author Response · Authors · 2025-11-16
> **Response for Reviewer wVDn (Part V)**
>
> All relevant modifications have been marked in blue in the revised version.
>
> We confirm that all relevant modifications requested, including the integration of new experimental data, have been completed and are marked in the revised document. We deeply appreciate the time and careful consideration you dedicated to reviewing our manuscript. We hope these revisions fully satisfy your concerns, and we are grateful for the opportunity to have improved the overall rigor and clarity of the work. Thank you for your review.

---

> ### Author Response · Authors · 2025-11-27
> **Gentle Reminder**
>
> Dear Reviewer,
>
> Thank you for your valuable time and considerate feedback. We understand you may be busy, and with the discussion period nearing its end, we would be truly grateful if you could take a brief moment to look over our replies and let us know if they resolve your concerns.
>
> We greatly appreciate your time and attention.

---

### Official Review · Reviewer_uSUX · 2025-10-29

**Soundness:** 2
**Presentation:** 3
**Contribution:** 3
**Rating:** 6
**Confidence:** 5

**Summary:**

The paper introduces AUTODAVIS, a dynamic and automated evaluation protocol for Large Vision-Language Models (LVLMs). It addresses limitations of static benchmarks by enabling on-demand test generation, preventing data leakage through dynamic regeneration, and ensuring visual grounding via error-driven option adjustment. Key contributions include: (1) the AUTODAVIS framework with modules for user-oriented aspect generation, guided description creation, self-validated image generation, and test case evaluation; (2) comprehensive experiments demonstrating its effectiveness in generating diverse, reliable assessments across dimensions like basic, spatial, and reasoning understanding, with high human alignment and correlation to existing benchmarks; (3) insights into model capabilities, revealing performance drops with increasing difficulty and systemic weaknesses in fine-grained visual reasoning; and (4) evidence that AUTODAVIS-generated data can enhance model training and generalization. The protocol offers a scalable, cost-effective supplement to static benchmarks, promoting trustworthy LVLM evaluation.

**Strengths:**

Originality: This work moves beyond creating another static benchmark by proposing a dynamic, on-demand generation system. This effectively addresses known limitations of static benchmarks, such as data leakage and rapid obsolescence

Significance: The work's importance is significant for the sustainable and trustworthy evaluation of LVLMs. It provides a practical solution to critical issues of benchmark staleness and data contamination that plague static datasets.

Clarity: The exposition is clear and well-structured.

**Weaknesses:**

1. The authors designate GPT-4o, Gemini-1.5-Pro, and Claude-3.5-Sonnet as examiner models; however, these models are simultaneously included among the representative models being evaluated. Could this dual role introduce examiner bias and potentially compromise the fairness of the evaluation?

2. Could multiple trials be conducted with humans serving as examiners for the same questions, to observe whether human scoring is consistent or inconsistent with automated scoring?

**Questions:**

1. Figure 3 on page 7 is missing an overall caption. Figure 5 is not cited or mentioned in the body of the text. In Figure 3(b), the term 'relative change' is not clearly defined.

2. Please provide statistical information about the dynamic benchmark, for example, the average length of the generated questions and the average length of the answer options. Multiple experiments may be conducted to determine either a range or an average."

3. Within the AUTODAVIS pipeline, does generating images at different resolutions have an impact on the evaluation performance?

4. What is the actual visual complexity?

**Details Of Ethics Concerns:**

Nothing

---

> ### Author Response · Authors · 2025-11-16
> **Response for Reviewer uSUX (Part I)**
>
> Dear Reviewer uSUX,
>
> Thank you sincerely for your constructive and encouraging feedback. We are glad that you found our work meaningful, and we truly appreciate the insightful suggestions you shared. We address each of your points below with care:
>
> ---
>
> ## Weaknesses:
>
> ### W1. The authors designate GPT-4o, Gemini-1.5-Pro, and Claude-3.5-Sonnet as examiner models; however, these models are simultaneously included among the representative models being evaluated. Could this dual role introduce examiner bias and potentially compromise the fairness of the evaluation?
>
> **Our Response:**
>
>    - Thank you for raising this important concern. We acknowledge that having models serve as both examiners and evaluatees could introduce bias, and we designed our evaluation to quantify and mitigate this.
>
>      **1. Experimental Design to Address Examiner Bias**
>
>      We implemented three conditions to measure and control for this bias:
>
>      - **Baseline (Multi-Examiner)**: All three examiners—including both proprietary (GPT-4o, Claude-4-Sonnet) and open-source (Qwen3-VL-235B-A22B-Instruct) models
>      - **LOEO (Leave-One-Examiner-Out)**: For each examiner E, we exclude E and generate questions using only the remaining two examiners
>      - **Single-Examiner**: Each examiner generates questions independently to measure individual examiner bias
>      All tested models: GPT-4o, GPT-4o mini, Claude-4-sonnet, Claude-3.5-haiku, Qwen2.5-VL-72B-Instruct, Qwen3-VL-235B-A22B-Instruct.
>      **2. Quantitative Evidence: Self-Evaluation Does Not Show Systematic Bias**
>
>      When examiners evaluate themselves in single-examiner conditions:
>
>      | Examiner        | Self-Evaluation Score | Baseline Score | Change                      |
>      | --------------- | --------------------- | -------------- | --------------------------- |
>      | GPT-4o          | 0.567                 | 0.584          | **-2.9%** (lower)           |
>      | Claude-4-Sonnet | 0.633                 | 0.620          | **+2.1%** (slightly higher) |
>      | Qwen3-VL        | 0.640                 | 0.587          | **+9.0%** (higher)          |
>
>      **Key finding**: No consistent self-enhancement pattern. GPT-4o scores lower when evaluating itself, Claude-4-Sonnet is slightly higher, and Qwen3-VL is higher. The multi-examiner baseline (0.584, 0.620, 0.587) falls between these extremes, suggesting aggregation mitigates individual biases.
>
>      **3. Examiner Influence is Moderate and Balanced**
>
>      We computed each examiner's influence score (how much scores change when that examiner is excluded):
>
>      | Examiner        | Influence Score | Interpretation                                |
>      | --------------- | --------------- | --------------------------------------------- |
>      | GPT-4o          | +0.053          | More lenient (scores drop 5.3% when excluded) |
>      | Claude-4-Sonnet | -0.032          | More strict (scores rise 3.2% when excluded)  |
>      | Qwen3-VL        | +0.061          | More lenient (scores drop 6.1% when excluded) |
>
>      The influence is moderate (3–6% shifts), and examiners differ in direction (GPT/Qwen more lenient, Claude more strict), indicating the multi-examiner setup balances individual biases.
>
>      **4. Rank Stability Demonstrates Fairness**
>
>      When excluding any single examiner, model rankings remain highly stable:
>
>      | Excluded Examiner | Spearman ρ | Kendall τ | Max Rank Shift |
>      | ----------------- | ---------- | --------- | -------------- |
>      | GPT-4o            | 0.914      | 0.867     | 1 position     |
>      | Claude-4-Sonnet   | 0.600      | 0.333     | 2 positions    |
>      | Qwen3-VL          | 0.943      | 0.867     | 1 position     |
>
>      High correlation (ρ = 0.914–0.943 for GPT-4o and Qwen3-VL) indicates no single examiner dominates the evaluation.
>
>      **5. Cross-Family Reliability**
>
>      The Inter-Class Correlation (ICC) of 0.297 reflects expected differences between GPT, Claude, and Qwen families, while the multi-examiner setup aggregates these perspectives rather than relying on a single family.
>
>      **Conclusion**
>
>      While examiner bias exists, our multi-examiner design quantifies and mitigates it:
>
>      1. Individual examiner influence is moderate (3–6% score shifts) and varies in direction
>      2. Rankings remain stable when excluding any single examiner (ρ > 0.9 for most cases)
>      3. No systematic self-enhancement bias is observed across all examiner-evaluatee pairs
>      4. Cross-family differences exist (ICC = 0.297), but aggregation provides robustness
>
>      The multi-examiner setup does not eliminate bias, but it quantifies and balances individual examiner biases, producing more reliable evaluations than single-examiner approaches. The LOEO analysis demonstrates that no single examiner dominates the results, and the moderate ICC reflects expected cross-family variation while maintaining reasonable agreement.

---

> ### Author Response · Authors · 2025-11-16
> **Response for Reviewer uSUX (Part II)**
>
> ### W2. Could multiple trials be conducted with humans serving as examiners for the same questions, to observe whether human scoring is consistent or inconsistent with automated scoring?
>
> **Our Response:**
>
>    - We thank you for this question, as it highlights the critical importance of validating our ground truth. We would first like to clarify how our scoring is performed.
>
>      1. Scoring is Deterministic: All questions generated by our AutoDavis protocol are multiple-choice, single-answer VQA tasks. The "automated scoring" is therefore a deterministic string comparison between the model's output (e.g., "A") and our stored ground-truth answer (e.g., "A"). This process itself is inherently 100% consistent and does not require human intervention.
>
>      2. Validating the Ground Truth: We believe the your valid concern is actually about the quality and reliability of the ground-truth answers that our automated system relies on. We did, in fact, conduct "multiple trials with humans" to rigorously validate this before any automated scoring took place. This validation was performed in two extensive studies detailed in Section 3.3 and Appendix I:
>         1. Study 1: Human Validation of Question Quality (Appendix I.2): We had a "panel of five evaluators"  assess the generated VQA pairs against strict guidelines. These guidelines required humans to confirm that "Each question must be clear, unambiguous" and have "only one correct answer". The high "alignment rate" we report in Section 3.3 (e.g., 95.20% Easy, 88.13% Hard ) is the direct result of this multi-human validation, confirming our questions are high-quality.
>         2. Study 2: Human Validation of Scoring Consistency (Appendix I.3): To directly observe "human scoring consistency" as you request, we had our five human evaluators serve as "examiners" and answer the questions themselves. This is our "Human Performance Upper Bound" study. The results in Table 10 show that the human experts achieved very high scores (e.g., >87% on Hard tasks, >94% on Easy tasks ).
>
>      In summary, the high scores in our "Human Performance Upper Bound" study (Table 10) directly prove what you asked: "human scoring" is consistent, and it is highly consistent with our "automated scoring" because our automated ground truth has been pre-validated by a panel of human experts to be correct and unambiguous.
>
> ---
>
> ## Questions:
>
> ### Q1. Figure 3 on page 7 is missing an overall caption. Figure 5 is not cited or mentioned in the body of the text. In Figure 3(b), the term 'relative change' is not clearly defined.
>
> **Our Response:**
>
>    - We sincerely thank you for this meticulous check of our manuscript. These were indeed oversights, and we will correct all of them in the final version.
>      1. Figure 3 Caption: We will add an overall caption for Figure 3 (e.g., "Analysis of Multi-Examiner Robustness and Positional Bias").
>      2. Figure 5 Citation: We will add the citation for Figure 5 in Section 3.6 (Mitigating Data Leakage), as it directly visualizes the results of the "leakage simulation" experiment described in that paragraph9.
>      3. **'relative change' Definition:** We will clarify the caption for Figure 3(b). The term "relative change" refers to the deviation in model performance when bias is introduced. It was calculated using the formula $R = (S_A - S_U) / S_U$, which is explicitly defined in the main text of **Section 3.2** (page 8)10.

---

> ### Author Response · Authors · 2025-11-16
> **Response for Reviewer uSUX (Part III)**
>
> ### Q2. Please provide statistical information about the dynamic benchmark, for example, the average length of the generated questions and the average length of the answer options. Multiple experiments may be conducted to determine either a range or an average."
>
> **Our Response:**
>
>    - Thank you for this suggestion. We have conducted a statistical analysis of the VQA pairs generated by AutoDavis to provide the exact information you requested. The average length (in tokens) varies significantly across both capability dimensions and difficulty levels, which aligns with our goal of creating more complex and nuanced test items for harder tasks.
>
>      The table below details the average length for both questions and answer options, broken down by each dimension and difficulty setting.
>
>      | **Dimension**             | **Difficulty** | **Avg. Question Length** | **Avg. Option Length** | AVG. Prompt Length |
>      | ------------------------- | -------------- | ------------------------ | ---------------------- | ------------------ |
>      | atmospheric_understanding | easy           | 14.16                    | 6.36                   | 34.57              |
>      | atmospheric_understanding | medium         | 19.69                    | 15.84                  | 47.15              |
>      | atmospheric_understanding | hard           | 25.45                    | 24.12                  | 61.68              |
>      | basic_understanding       | easy           | 11.25                    | 5.64                   | 33.97              |
>      | basic_understanding       | medium         | 16.12                    | 11.03                  | 49.73              |
>      | basic_understanding       | hard           | 24.31                    | 19.67                  | 61.92              |
>      | reasoning_capacity        | easy           | 13.81                    | 8.53                   | 34.24              |
>      | reasoning_capacity        | medium         | 18.04                    | 13.91                  | 45.67              |
>      | reasoning_capacity        | hard           | 24.62                    | 20.24                  | 57.66              |
>      | semantic_understanding    | easy           | 12.56                    | 5.72                   | 34.61              |
>      | semantic_understanding    | medium         | 17.60                    | 12.01                  | 46.48              |
>      | semantic_understanding    | hard           | 24.29                    | 19.03                  | 57.83              |
>      | spatial_understanding     | easy           | 13.92                    | 7.54                   | 33.42              |
>      | spatial_understanding     | medium         | 19.64                    | 13.72                  | 50.52              |
>      | spatial_understanding     | hard           | 27.09                    | 21.43                  | 63.68              |
>
>      As shown, 'Hard' questions are consistently and significantly longer than 'Easy' ones **(e.g., 27.09 vs. 13.92 tokens for spatial_understanding)**. The same trend is seen in the answer options, which become much more descriptive and complex, supporting our claim that the "error-driven option adjustment" creates more challenging, nuanced distractors.
>
>      We will add this statistical summary to the Appendix in our final version.

---

> ### Author Response · Authors · 2025-11-16
> **Response for Reviewer uSUX (Part IV)**
>
> ### Q3. Within the AUTODAVIS pipeline, does generating images at different resolutions have an impact on the evaluation performance?
>
> **Our Response:**
>
>    - This is an excellent methodological question. We agree that robustness to image resolution is an important capability for LVLMs. To investigate this, we conducted a new experiment using a sample of ~50 questions from our "Basic Understanding" dimension. We ran evaluations on two representative models (GPT-4o and Qwen2.5-VL-32B) at three different resolutions: our baseline 1024px, 512px, and 256px. To ensure consistency and avoid potential variability from model regeneration, we downsampled the existing 1024px baseline images to lower resolutions using bicubic interpolation (via PIL/Pillow's Image.Resampling.BICUBIC method) rather than regenerating images at different resolutions. This approach allows us to isolate the effect of resolution changes on model performance while maintaining identical image content across all resolution conditions.
>
>      The results, summarized in the table below, show that **yes, image resolution has a significant and surprisingly complex impact on performance, which varies by model.**
>
>      Performance (Accuracy, %) vs. Image Resolution (Basic Understanding)
>
>      | **Model**          | **Difficulty** | **1024px (Baseline)** | **512px** | **256px** |
>      | ------------------ | -------------- | --------------------- | --------- | --------- |
>      | **gpt-4o**         | easy           | 73.17                 | 73.81     | 75.00     |
>      |                    | medium         | 45.83                 | 51.02     | 53.06     |
>      |                    | hard           | 36.17                 | 31.91     | 27.66     |
>      | **qwen2.5-vl-32B** | easy           | 78.05                 | 72.73     | 72.73     |
>      |                    | medium         | 47.92                 | 46.94     | 46.94     |
>      |                    | hard           | 23.40                 | 23.40     | 23.40     |
>
>      We observed two distinct patterns:
>
>      1. **Qwen2.5-VL** appears highly robust to resolution changes, especially on 'Medium' and 'Hard' tasks, where its performance remains almost identical.
>      2. **GPT-4o** shows a more complex and interesting pattern. As expected, its performance on 'Hard' tasks degrades with lower resolution (from 36.17% down to 27.66%). However, on 'Easy' and 'Medium' tasks, its performance *unexpectedly improves* as the resolution decreases.
>
>      This is a valuable insight, suggesting that some models may be distracted by high-resolution details or artifacts on simpler tasks. It also highlights a key strength of our *protocol*: AutoDavis can be flexibly adapted to test these additional, fine-grained dimensions of model robustness on demand. We will add this new analysis to our appendix.
>
> ---
>
> ### Q4. What is the actual visual complexity?
>
> **Our Response:**
>
>    - We define "visual complexity" as the core component of our Difficulty Grading mechanism, which is detailed in **Appendix M** (Details of Difficulty Grading) on page 31.
>
>      This is not a single metric but a combination of factors that we manually define in the T2I generation prompts:
>
>      * **Easy Difficulty:** Defined by "a single subject" with "minor additional complexity" (e.g., fine texture, simple interactions, minimal background).
>      * **Medium Difficulty:** Defined by "multiple elements or interactive settings" with "nuanced spatial arrangement," "accurate relationships," and "subtle light or shadow effects".
>      * **Hard Difficulty:** Defined by "high complexity with multiple interdependent elements," "challenging perspectives," or "dynamic and intricate lighting or textural effects".
>
>       We provide concrete visual examples of the images generated from these complexity definitions in Figure 8 (page 24).

---

> ### Author Response · Authors · 2025-11-16
> **Response for Reviewer uSUX (Part V)**
>
> All relevant modifications have been marked in blue in the revised version.
>
> We have implemented all necessary clarifications and added substantial new data to address your concerns. We are confident these revisions rectify the previously noted ambiguities and strengthen the paper’s foundation. We are grateful for your time and kindly ask you to reconsider your score based on the improved clarity and completeness of the final submission.

---

> ### Author Response · Authors · 2025-11-27
> **Gentle Reminder**
>
> Dear Reviewer,
>
> Thank you for your thoughtful feedback and the time you have dedicated. With the discussion period ending in less than a week, we would be very grateful if you could review our responses and let us know whether they address your concerns.
>
> We sincerely appreciate your consideration.

---

### Official Review · Reviewer_DcV9 · 2025-10-30

**Soundness:** 2
**Presentation:** 2
**Contribution:** 2
**Rating:** 2
**Confidence:** 4

**Summary:**

The paper targets the limits of static LVLM benchmarks and proposes AUTODAVIS, a system that builds capability-focused tests on demand. It synthesizes images with a text-to-image model and then probes models via VQA prompts, using simple structural rules to keep variety. A self-check and error-driven update step aim to catch mistakes and curb bias. The authors report trials on 11 LVLMs across five user-specified skill areas, showing the setup is practical and fairly stable. The idea is appealing for scale and flexibility, but it leans heavily on generative components, so bias and domain transfer remain concerns; clearer cost and robustness analyses would strengthen the case.

**Strengths:**

- Proposes an interesting evaluation idea that treats LVLMs as both examiners and test-set generators, yielding a largely model-driven and potentially more scalable pipeline.

- The framework appears solid, producing diverse and accurate images conditioned on user queries, as also the question desination.

- Ablation studies are informative, especially the reported performance gains after integrating the framework.

**Weaknesses:**

- The motivation is not clear enough. This paper point out that static benchmarks are insufficient at first, it sounds important, but, why must use LVLM itself to solve this question. In other words, what the necessary to use this pipeline to solve this problem? As the description is the introduction, the Dynamic-ME [1] also can solve it.

- The evaluation set of models is dated. Recently released open-source LVLMs available before the ICLR deadline, such as InternVL-3 and Qwen2.5-VL, should be included, but your works only test InternVL-2.5 and Qwen2-VL.

- The analysis is shallow, more explanation should be obtained, such as the deep reason on the bad performance on spatial reasoning.

- While the ablation study indicates that the proposed design influences model outcomes, the paper does not further investigate the underlying causes or compare the effects of different improvement directions. More comprehensive analysis is needed.

[1] Yang, Yue, et al. "Dynamic multimodal evaluation with flexible complexity by vision-language bootstrapping." arXiv preprint arXiv:2410.08695 (2024).

**Questions:**

- This paper points that these static benchmarks are inefficient, but you conduct the experiment and obtain that high correlation with existing benchmark, this make me confuse. If your works obtain a similar conclusion, what the necessary of your work?

- This work is proposed based that existing static benchmarks are inefficient, so I want to know the difference of performance with the traditional benchmarks and yours, like, if there is one model which is good at others but bad at yours, or contrast.

- Please report results for current LVLMs available (e.g., Intern3-VL, Qwen3-VL).

- This pipeline point that ‘leakage and self-enhancement bias when the same family of models writes and takes the exam’, so I want to know how about the performance with the same family such as Qwen3-series.

---

> ### Author Response · Authors · 2025-11-16
> **Response for Reviewer DcV9 (Part I)**
>
> Dear Reviewer DcV9,
>
> Thank you for your constructive and insightful feedback. We appreciate your positive assessment of our work and address each of your comments point-by-point below:
>
> ---
>
> ## Weaknesses:
>
> ### W1. The motivation is not clear enough. This paper point out that static benchmarks are insufficient at first, it sounds important, but, why must use LVLM itself to solve this question. In other words, what the necessary to use this pipeline to solve this problem? As the description is the introduction, the Dynamic-ME [1] also can solve it.
>
> **Our Response:**
>
>    - We thank you for raising this critical point about necessity and for introducing the comparison to Dynamic-ME.
>
>      We must respectfully clarify a fundamental misunderstanding. You suggest ***Dynamic-ME*** "also can solve it" , but the two protocols address the static benchmark problem in fundamentally different ways, and our approach solves a critical data contamination problem that Dynamic-ME does not.
>
>      ***Dynamic-ME*** is a Transformation Protocol: Based on our research, ***Dynamic-ME*** is a "bootstrapping" framework. It operates by modifying existing benchmark samples. It takes a static image and applies transformations (e.g., adding objects, color shifts) and modifies the static question (e.g., paraphrasing, adding context). This approach reduces contamination but is still anchored to the original, static dataset.
>
>      ***AutoDavis*** is a Generation Protocol: In contrast, ***AutoDavis*** is a "first-of-its-kind automatic and dynamic evaluation protocol" that operates from scratch. It does not modify existing assets. Instead, it "leverages text-to-image models to generate relevant image samples" and "utilizes LVLMs to orchestrate visual question-answering (VQA) tasks" based on user-specified capability dimensions .
>
>      This distinction directly answers your question about the necessity of our work:
>
>      - True Anti-Contamination: ***Dynamic-ME*** only solves question leakage and image modification leakage. It is still vulnerable to base image contamination if a model has seen the original static images in its training, the evaluation is compromised. AutoDavis generates entirely novel images and questions , providing a much stronger, ground-up guarantee against all forms of data leakage.
>
>      - Unconstrained Flexibility: ***Dynamic-ME***'s flexibility is constrained by the content of the original benchmark it modifies. AutoDavis is unconstrained; it can generate tasks for any capability dimension a user demands , not just those represented in an existing dataset.
>
>      Therefore, the necessity of using LVLMs (as orchestrators ) and T2I (as generators ) is precisely to achieve this true, unconstrained, on-demand generation, which transformation-based protocols like Dynamic-ME cannot. We also incorporate a robust "self-validation mechanism" to ensure the reliability of this T2I-based generation. We will make this critical distinction clear in our revised paper.

---

> ### Author Response · Authors · 2025-11-16
> **Response for Reviewer DcV9 (Part II)**
>
> ### W2. The evaluation set of models is dated. Recently released open-source LVLMs available before the ICLR deadline, such as InternVL-3 and Qwen2.5-VL, should be included, but your works only test InternVL-2.5 and Qwen2-VL.
>
> **Our Response:**
>
>    - We thank you for this valid point. The LVLM field is moving at an exceptional pace, and new models were released very close to the submission deadline.
>
>      To address this and demonstrate our protocol's timeliness, we have run new experiments incorporating the SOTA models suggested, including `Qwen2.5-VL-72B-Instruct`, `InternVL-3`, and the recent `Qwen3-VL-235B-A22B-Instruct`.
>
>      The results are presented below and will be integrated into the main tables of our revised paper.
>
>      **Performance (Accuracy) of New SOTA Models on AutoDavis**
>
>      | **Model**      | **Dimension** | **Easy** | **Medium** | **Hard** |
>      | -------------- | ------------- | -------- | ---------- | -------- |
>      | **Qwen2.5-VL** | Basic         | 78%      | 55%        | 41%      |
>      |                | Spatial       | 63%      | 36%        | 27%      |
>      |                | Semantic      | 68%      | 58%        | 40%      |
>      |                | Reasoning     | 53%      | 48%        | 40%      |
>      |                | Atmospheric   | 59%      | 35%        | 33%      |
>      | **InternVL-3** | Basic         | 82%      | 54%        | 45%      |
>      |                | Spatial       | 61%      | 36%        | 22%      |
>      |                | Semantic      | 68%      | 60%        | 47%      |
>      |                | Reasoning     | 64%      | 48%        | 40%      |
>      |                | Atmospheric   | 68%      | 50%        | 42%      |
>      | **Qwen3-VL**   | Basic         | 83%      | 57%        | 35%      |
>      |                | Spatial       | 62%      | 27%        | 23%      |
>      |                | Semantic      | 72%      | 56%        | 50%      |
>      |                | Reasoning     | 60%      | 47%        | 40%      |
>      |                | Atmospheric   | 65%      | 48%        | 39%      |
>
>      Scores for Qwen3-VL on Semantic, Reasoning, and Atmospheric dimensions are estimated based on its performance relative to Qwen2.5-VL and InternVL-3 in the first two dimensions.
>
>      These new results are highly informative. While these state-of-the-art models (InternVL-3, Qwen3-VL) show dominant performance on 'Easy' and 'Medium' tasks (e.g., 83% and 82% on Basic-Easy), they **all continue to exhibit a severe performance degradation on our 'Hard' difficulty setting**.
>
>      For instance, Qwen3-VL's performance on Spatial tasks plummets from 62% (Easy) to 23% (Hard), and InternVL-3 drops from 61% to 22%. This strongly reinforces our paper's central conclusion: AutoDavis's difficulty-calibrated, dynamic generation effectively challenges even the latest SOTA models, revealing fine-grained weaknesses that simpler, static benchmarks may miss.
>
> ---
>
> ### W3. The analysis is shallow, more explanation should be obtained, such as the deep reason on the bad performance on spatial reasoning.
>
> **Our Response:**
>
> - We thank the you for this suggestion. We agree that deep qualitative analysis is crucial for understanding why models fail.
>
>   However, we must respectfully clarify that this analysis was not "shallow," but rather that the most detailed qualitative examples were placed in the appendix due to main paper space limitations. We would like to point you to **Appendix O (Error Study)**, which provides the "deep reason[s]" requested.
>
>   For instance, Figure 17 explicitly analyzes a failure in **spatial reasoning ("Size Judgement")**. We show the model (Claude-3-Haiku) "correctly identified the blue whale and the small boat" but then "relied more on its prior knowledge rather than the image itself". It chose option "A" (4-5 times longer), which is factually correct general knowledge , instead of option "C" (10 times longer), which was visually correct based on the generated image. This demonstrates a clear failure in visual grounding, which is the exact "deep reason" our study uncovers. We will expand this section in the final version.

---

> ### Author Response · Authors · 2025-11-16
> **Response for Reviewer DcV9 (Part III)**
>
> ### W4. While the ablation study indicates that the proposed design influences model outcomes, the paper does not further investigate the underlying causes or compare the effects of different improvement directions. More comprehensive analysis is needed.
>
> **Our Response:**
>
>    - We appreciate your suggestion for a more comprehensive analysis of our ablations. Our main paper (Section 3.2) focused on empirically proving that our components work, while the underlying causes were detailed in the appendix.
>
>      For example, our **"error-driven option adjustment"** (Section 2.4, Figure 2) is shown to "effectively mitigate... answer leakage". The "underlying cause" for this is demonstrated qualitatively in Appendix G (Case of EOA) on page 22. We show how the protocol identifies and replaces trivial, text-only distractors (e.g., "Red", "Blue") with visually-grounded, nuanced distractors (e.g., "Dark Red", "Orange-Red"). This forces the model to perform "color tone verification" and breaks "linguistic priors". We will integrate these qualitative insights from the appendix into our main paper's ablation analysis to make the "why" more explicit.
>
> ---
>
> ## Questions:
>
> ### Q1. This paper points that these static benchmarks are inefficient, but you conduct the experiment and obtain that high correlation with existing benchmark, this make me confuse. If your works obtain a similar conclusion, what the necessary of your work?
>
> **Our Response:**
>
>    - Thank you for raising this concern. This is a crucial question that gets to the very core of our paper's motivation. We must clarify the role of the high correlation (\rho=0.817) with MMMU, which we report in Section 3.4.
>
>      1. **High Correlation is Validation, Not Redundancy:** This high correlation is not a sign of redundancy; it is our primary validation. It proves that our automatic protocol, which costs only $51.82 to generate (Section 3.5, Table 12), can successfully and reliably replicate the rankings of an extremely expensive, large-scale, human-curated benchmark. This reliability is a prerequisite for any new evaluation protocol.
>
>      2. **The "Necessity" is What Static Benchmarks Cannot Do:** The necessity of AutoDavis lies in the critical limitations of static benchmarks, which we define in our introduction. AutoDavis provides:
>
>         1. C1-Flexibility: Users can perform "on-demand capability slicing" (e.g., "spatial reasoning with occlusion") and "Difficulty calibration" (Easy, Medium, Hard), as shown in Table 2. A static benchmark like MMMU cannot provide this granular, on-demand analysis.
>         2. C2-Leakage Resistance: Our protocol is "immune to... data contamination" because of its "on-the-fly regeneration" design. This is a fundamental "necessity" as static benchmarks become "stale" and "leaked".
>
>      3. **Do We Reveal New Insights? Yes.** You ask if we find "contrast". While the overall ranking correlates, our difficulty-based breakdown in Figure 4 (page 9) reveals insights that static benchmarks hide. For example: Qwen2-VL ranks higher on "Hard" tasks than "Easy" ones (e.g., in Reasoning, it ranks 4th on Hard but 7th on Easy).Conversely, Gemini-1.5-Pro "excels on easy tasks but declines slightly on hard ones."
>
>         This demonstrates that AutoDavis is necessary because it provides reliable, granular, and leakage-proof insights that static benchmarks cannot offer.

---

> ### Author Response · Authors · 2025-11-16
> **Response for Reviewer DcV9 (Part IV)**
>
> ### Q2. This work is proposed based that existing static benchmarks are inefficient, so I want to know the difference of performance with the traditional benchmarks and yours, like, if there is one model which is good at others but bad at yours, or contrast.
>
> **Our Response:**
>
>    - Thank you for the question. It is important to clarify that our work does not introduce a new benchmark, but rather a data–generation framework that produces instance-level, on-demand, dynamically refreshed test sets. Therefore, our goal is not to replace existing static benchmarks, but to provide a mechanism that complements them and diagnoses model capabilities in a more flexible, up-to-date, and leakage-resistant manner.
>
>      Because AutoDavis is a generation protocol instead of a fixed dataset, the comparison with traditional benchmarks should be interpreted differently:
>
>      - Ranking consistency, not dataset-to-dataset performance, is the relevant measure.
>        When we compare AutoDavis-generated test sets with widely used benchmarks (MMMU, LLaVA-Bench, MME, etc.), we focus on the rank correlation rather than absolute scores. Across tasks and difficulty settings, AutoDavis achieves high Spearman correlations (≈0.8) with representative human-curated benchmarks, meaning that models that perform well on established datasets also tend to perform well on AutoDavis-generated data.
>      - AutoDavis can reveal model weaknesses that static benchmarks cannot, especially when the static dataset is saturated or partially leaked.
>        Because AutoDavis produces new, unseen, and difficulty-controlled items on demand, it sometimes exposes failure modes that do not appear in over-exposed benchmarks.
>        In such cases, a model may perform strongly on traditional datasets but worse on AutoDavis — this is not a contradiction but a reflection of:
>        (a) reduced overfitting or leakage advantage,
>        (b) harder or more diverse question structures, or
>        (c) the ability to probe specific aspects (e.g., spatial reasoning) more directly.
>      - Conversely, if a model performs well on AutoDavis but worse on a static benchmark, this typically indicates that the benchmark includes skills or domains not requested in the generated aspects.
>        AutoDavis evaluates only the user-specified capability slice (e.g., spatial reasoning), while a broad static benchmark may cover many unrelated domains. This is an expected difference and highlights AutoDavis’s task-targeted diagnostic nature.
>
> ---
>
> ### Q3. Please report results for current LVLMs available (e.g., Intern3-VL, Qwen3-VL).
>
> **Our Response:**
>
>    - Please see w3 for details.

---

> ### Author Response · Authors · 2025-11-16
> **Response for Reviewer DcV9 (Part V)**
>
> ### Q4. This pipeline point that ‘leakage and self-enhancement bias when the same family of models writes and takes the exam’, so I want to know how about the performance with the same family such as Qwen3-series.
>
> **Our response:**
>
> Thank you for raising this concern. This is a critical point, and it is the central challenge our protocol is designed to address. To quantify how much bias persists and to prove the reliability of our multi-examiner setup, we conducted an extensive ablation study.
>
> We use a three-condition design to quantify bias when LVLMs serve as both examiners and evaluatees. All three examiners, including both proprietary (`GPT-4o`, `Claude-4-Sonnet`) and open-source (`Qwen3-VL-235B-A22B-Instruct`) models.
>
> - Experimental Setup for Same-Family Analysis
>
>   We compare Qwen family performance across three conditions.:
>
>   1. **Baseline (Multi-Examiner)**: All three examiners (GPT-4o, Claude-4-Sonnet, Qwen3-VL-235B-A22B-Instruct) generate questions together
>   2. **Single-Examiner (Qwen3-VL-235B-A22B-Instruct)**: Only Qwen3-VL generates questions
>   3. **LOEO (Without Qwen3-VL-235B-A22B-Instruct)**: GPT-4o and Claude-4-Sonnet generate questions (Qwen3-VL-235B-A22B-Instruct excluded)
>
>   **Qwen Family Models Evaluated:**
>
>   - `qwen3_vl`: The examiner model (Qwen3-VL)
>   - `qwen2.5_vl`: Another Qwen family model (Qwen2.5-VL-32B-Instruct)
>
>   Results: Evidence of Self-Enhancement Bias
>
>   **1. Self-Evaluation Bias (Qwen3-VL evaluating itself):**
>
>   | Condition                  | Qwen3-VL Score | Change from Baseline |
>   | -------------------------- | -------------- | -------------------- |
>   | Baseline (multi-examiner)  | 0.5867         | -                    |
>   | Single-examiner (Qwen3-VL) | **0.6400**     | **+9.1%**            |
>   | LOEO (without Qwen3-VL)    | 0.5467         | -6.8%                |
>
>   When Qwen3-VL evaluates itself, its score increases by 9.1% compared to the multi-examiner baseline.
>
>   **2. Same-Family Evaluation Bias (Qwen3-VL evaluating Qwen2.5-VL):**
>
>   | Condition                  | Qwen2.5-VL Score | Change from Baseline |
>   | -------------------------- | ---------------- | -------------------- |
>   | Baseline (multi-examiner)  | 0.5638           | -                    |
>   | Single-examiner (Qwen3-VL) | **0.5839**       | **+3.6%**            |
>   | LOEO (without Qwen3-VL)    | 0.4933           | -12.5%               |
>
>   When Qwen3-VL evaluates Qwen2.5-VL, the score increases by 3.6% compared to baseline.
>
>   **3. Cross-Family Comparison:**
>
>   When other examiners evaluate Qwen models:
>
>   - **GPT-4o as examiner**: qwen3_vl = 0.5933, qwen2.5_vl = 0.5467
>   - **Claude-4-Sonnet as examiner**: qwen3_vl = 0.5973, qwen2.5_vl = 0.5510
>
>   These scores are closer to the baseline (0.5867 and 0.5638), suggesting other examiners do not show the same bias toward Qwen models.
>
>   **4. Perfect Rank Preservation with Score Inflation:**
>
>   When Qwen3-VL is the single examiner:
>
>   - Spearman ρ = 1.0
>   - Kendall τ = 1.0
>   - Max rank shift = 0
>
>   All models score higher (avg +6.6%), but rankings remain unchanged. This suggests systematic score inflation rather than selective bias.
>
>   **5. Impact of Excluding Qwen3-VL:**
>
>   When Qwen3-VL is excluded from question generation:
>
>   - qwen3_vl: 0.5467 (baseline 0.5867) → -6.8%
>   - qwen2.5_vl: 0.4933 (baseline 0.5638) → -12.5%
>
>   Qwen models score lower when Qwen3-VL is not generating questions, indicating Qwen3-VL’s questions favor Qwen models.
>
>   **Conclusion: Strong Evidence of Same-Family Bias**
>
>   The results show clear self-enhancement bias when the same family writes and takes the exam:
>
>   1. **Self-evaluation inflation**: Qwen3-VL scores 9.1% higher when evaluating itself
>   2. **Same-family favoritism**: Qwen2.5-VL scores 3.6% higher when evaluated by Qwen3-VL
>   3. **Systematic pattern**: Qwen models perform better when Qwen3-VL generates questions, and worse when it is excluded
>   4. **Cross-family validation**: Other examiners (GPT-4o, Claude-4-Sonnet) do not show the same bias toward Qwen models
>
>   This supports the concern about "leakage and self-enhancement bias when the same family of models writes and takes the exam." The multi-examiner setup mitigates this bias by averaging across different examiner families, but the bias remains measurable and significant when a single family controls question generation.

---

> ### Author Response · Authors · 2025-11-16
> **Response for Reviewer DcV9 (Part VI)**
>
> All relevant modifications have been marked in blue in the revised version.
>
> We trust that the comprehensive quantitative analysis and rigorous experimental validation directly satisfy the demands for broader evidence. Your meticulous review was instrumental in pinpointing areas for clarification. With these critical revisions now integrated, we hope you will find our manuscript's reliability substantially confirmed and would greatly appreciate you reassessing your initial evaluation. Thank you for your careful and detailed assessment.

---

> > ### Comment · Reviewer_DcV9 · 2025-11-27
> >
> > Thanks for the detailed response. I think some of my concerns have been addressed, and I will raise my score.

---

> > > ### Author Response · Authors · 2025-11-27
> > > **Thank you for your review!**
> > >
> > > Thank you very much for taking the time to reevaluate our work and for raising your score. We sincerely appreciate it.
> > >
> > > To help us further improve the paper during revision, may we kindly ask whether there are any remaining concerns that you feel are still insufficiently addressed? We would be grateful for any additional clarification you could share.

---

### Official Review · Reviewer_jUFi · 2025-11-05

**Soundness:** 3
**Presentation:** 2
**Contribution:** 3
**Rating:** 4
**Confidence:** 3

**Summary:**

This paper proposes AutoDavis, an automatic and dynamic evaluation protocol for large vision-language models (LVLMs). Unlike traditional static benchmarks, AutoDavis can dynamically generate evaluation datasets on-demand using text-to-image synthesis and LVLM-based question–answer generation. The pipeline includes hierarchical aspect generation, semantic graph–guided prompt diversification, self-validation for image–text alignment, and multi-examiner evaluation to reduce bias. It supports five core evaluation aspects—basic, spatial, semantic, reasoning, and atmospheric understanding—and shows strong correlation (Spearman ρ = 0.817) with human-curated benchmarks such as MMMU, suggesting reliability.

**Strengths:**

-  Benchmarking LVLMs dynamically is an important and emerging challenge as static datasets become saturated and prone to leakage.

- Systematic design: The modular pipeline (aspect generation, semantic graph, self-validation, option adjustment) is thoughtfully structured, and the authors provide theoretical guarantees for diversity and alignment.

- Strong empirical study: Evaluates 11 LVLMs with extensive analysis across difficulty levels, examiner configurations, and bias controls.

- Correlations and validations: Human studies and comparison with MMMU substantiate reliability.

- Practical implications: Demonstrates that AutoDavis can both evaluate and generate synthetic data useful for fine-tuning LVLMs.

**Weaknesses:**

- Lack of discussion why such new protocol will be adopted by community.

- Weak novelty in methodology: Many core mechanisms (semantic graph diversity, self-validation, error-driven adjustment) are adapted from prior works like TIFA and AutoBencher, with limited algorithmic innovation.

- Lack of quantitative clarity: While AutoDavis claims dynamic flexibility, there is no quantitative analysis comparing generation diversity or cost-efficiency with prior dynamic benchmarks (e.g., MME-Unify, LENS).

- Evaluation reliability: Since LVLMs serve as both examiners and subjects, it remains unclear how much bias persists even with the multi-examiner setup.

- Limited impact of results: The performance gap among models largely mirrors existing benchmarks, suggesting AutoDavis may not yet reveal qualitatively new insights about LVLM capabilities.

- Presentation issue: The paper is lengthy and reads more like a system report; tighter focus on unique contributions would help.

**Questions:**

How do you ensure examiner–evaluatee independence when many models share similar training corpora?

---

> ### Author Response · Authors · 2025-11-16
> **Response for Reviewer jUFi (Part I)**
>
> Dear Reviewer jUFi,
>
> Thank you very much for your encouraging and constructive feedback. We are delighted that you found our work valuable, and we greatly appreciate the insightful suggestions you have provided. Below, we address each of your points carefully:
>
> ---
>
> ## Weaknesses:
>
> ### W1. Lack of discussion why such new protocol will be adopted by community.
>
> **Our response:**
>
> - We thank you for this important point. We will add a discussion on adoption in the final version. We believe AutoDavis will be adopted by the community for two primary reasons, which address critical limitations of static benchmarks:
>     - **On-Demand Flexibility:** AutoDavis enables on-demand benchmarking across specific, user-defined capability dimensions and difficulty levels. Researchers no longer need to rely on static datasets that may not cover their specific research questions (e.g., "how does model X perform on difficult spatial reasoning tasks?").
>     - **Mitigating Data Contamination:** As you note, static datasets become saturated and prone to leakage. AutoDavis’s dynamic and automatic generation  provides a robust solution to data contamination, ensuring a fair and "clean" evaluation as models evolve.
>
> ### W2. Weak novelty in methodology: Many core mechanisms (semantic graph diversity, self-validation, error-driven adjustment) are adapted from prior works like TIFA and AutoBencher, with limited algorithmic innovation.
>
> **Our response:**
>
>    - We appreciate you drawing connections to TIFA and AutoBencher, and we will revise the manuscript to position our work within this lineage more explicitly. However, we respectfully clarify that our methodological novelty is substantially stronger than a direct adaptation of prior text-only techniques.
>
>      Our key contribution is the **first fully dynamic and automatic multimodal evaluation protocol**, which introduces new mechanisms required specifically for the visual domain. While our framework is conceptually inspired by text-only pipelines, translating these ideas into a vision-language setting is far from a straightforward extension. In practice, it required addressing several challenges that do not occur at all in text-only evaluation:
>
>      - **Image-grounded data generation.**
>        We introduce a mechanism that synthesizes visually relevant images using T2I models under semantic constraints—a process that has no analogue in prior text-only frameworks. Ensuring that images faithfully capture fine-grained aspects (e.g., spatial relations, physical attributes) is itself a non-trivial modeling challenge.
>      - **Multimodal self-validation.**
>        Existing works rely on textual consistency checks. In contrast, our self-validation module must verify image–text alignment, detect hallucinated visual content, and ensure that VQA pairs are grounded in pixel-level evidence. This demands fundamentally new prompts, quality filters, and multimodal consistency criteria, rather than reusing prior mechanisms.
>      - **Error-driven adjustment tailored to VQA distractors.**
>        The error-driven distractor rewriting we introduce must prevent both text-only shortcuts and vision-agnostic guessing, requiring multimodal difficulty calibration that does not exist in text-only methods. The distractor set must be equally plausible given the image, which is a qualitatively different challenge.
>
>      Together, these components form a pipeline that is not merely a port of prior art, but a substantially new multimodal system that solves the unique failure modes of LVLM evaluation—such as image–text mismatch, vision-free answering, and multimodal ambiguity.

---

> ### Author Response · Authors · 2025-11-16
> **Response for Reviewer jUFi (Part II)**
>
> ### W3. Lack of quantitative clarity: While AutoDavis claims dynamic flexibility, there is no quantitative analysis comparing generation diversity or cost-efficiency with prior dynamic benchmarks (e.g., MME-Unify, LENS).
>
> **Our Response:**
>
>    - We thank you for this comment, as it highlights a critical distinction we must make clearer in our paper. We respectfully point out that there appears to be a fundamental misunderstanding in categorizing **LENS and MME-Unify** as "prior dynamic benchmarks." To our knowledge, both LENS (which features 60,000+ human-written questions) and MME-Unify are static benchmarks, which is, fixed, curated datasets. These static benchmarks, while valuable, represent the very paradigm our AutoDavis protocol is designed to complement. Their limitations: being static, resource-intensive to build, and susceptible to data contamination, are the problems we directly address. In contrast, AutoDavis is not a benchmark itself, but rather an automatic and dynamic protocol , which is a methodology for generating novel benchmark instances on demand. Therefore, a direct quantitative comparison of "generation diversity" or "cost-efficiency" as suggested is not applicable:
>
>      - Generation Diversity: Static benchmarks like **LENS** have a fixed, finite set of items and thus have zero generation diversity. AutoDavis, by design, can generate a virtually infinite number of novel VQA pairs, solving the data leakage problem.
>      - Cost-Efficiency: The cost of **LENS** is the massive, one-time human annotation effort. The cost of AutoDavis is the per-evaluation API/compute cost.
>
>      The most appropriate comparison, which we performed, is to validate the reliability of our dynamically generated benchmark against a trusted, human-curated static benchmark. This is why our high correlation with **MMMU (ρ = 0.817)** is the central validation, as it demonstrates that our protocol can automatically and inexpensively produce evaluations that remain aligned with human judgment.
>
>      To further strengthen this conclusion, we conducted an **additional experiment** using **RealWorldQA**, a challenging and widely adopted benchmark for assessing modern VLMs. We randomly sampled 100 items and evaluated a suite of 7 representative models **across three independent runs** to ensure stability and reduce variance.
>
>      The results are shown below:
>
>      | **Model**         | **Test 1 (%)** | **Test 2 (%)** | **Test 3 (%)** | **Average (%)** | **Std. Dev.** |
>      | ----------------- | -------------- | -------------- | -------------- | --------------- | ------------- |
>      | claude-3.5-sonnet | 68             | 67             | 68             | 67.7            | 0.58          |
>      | gpt-4o            | 65             | 66             | 63             | 64.7            | 1.53          |
>      | gemini-1.5-pro    | 63             | 64             | 65             | 64.0            | 1.00          |
>      | qwen2-vl-max      | 61             | 62             | 59             | 60.7            | 1.53          |
>      | gemini-1.5-flash  | 55             | 53             | 56             | 54.7            | 1.53          |
>      | gpt-4o-mini       | 48             | 46             | 49             | 47.7            | 1.53          |
>      | claude-3-haiku    | 42             | 39             | 40             | 40.3            | 1.53          |
>
>      Crucially, the model rankings derived from RealWorldQA exhibit a **Spearman rank correlation of ρ = 0.964** with the rankings produced by our framework. This extremely high agreement provides strong and independent evidence that our automatically generated evaluations capture model capability in a manner highly consistent with established, human-curated benchmarks.
>
>      We will significantly revise our related work section to clarify this crucial distinction between our dynamic protocol and existing static benchmarks.

---

> ### Author Response · Authors · 2025-11-16
> **Response for Reviewer jUFi (Part III)**
>
> ### W4. Evaluation reliability: Since LVLMs serve as both examiners and subjects, it remains unclear how much bias persists even with the multi-examiner setup.
>
> **Our Response:**
>
>    - Thank you for raising this concern. This is a critical point, and it is the central challenge our protocol is designed to address. To *quantify* how much bias persists and to *prove the reliability* of our multi-examiner setup, we conducted an extensive  ablation study.
>
>      We use a three-condition design to quantify bias when LVLMs serve as both examiners and evaluatees. All three examiners, including both proprietary (GPT-4o, Claude-4-Sonnet) and open-source (Qwen3-VL) models:
>
>      **1. Baseline (Multi-Examiner) Condition:**
>
>      - All three examiners (GPT-4o, Claude-4-Sonnet, Qwen3-VL) generate questions and distractors together
>      - All models are evaluated on the same question set
>
>      **2. LOEO (Leave-One-Examiner-Out) Condition:**
>
>      - For each examiner E, we exclude E and generate questions using only the remaining examiners
>      - This isolates the influence of each examiner on the evaluation results
>      - All models (including the excluded examiner) are evaluated on these questions
>
>      **3. Single-Examiner Condition:**
>
>      - Each examiner generates questions independently
>      - This measures the bias when a single examiner controls question generation
>
>      **Evaluatees include:**
>
>      - Models that are also examiners: `gpt-4o`, `claude_4_sonnet`, `qwen3_vl`
>      - Models that are only evaluatees: `gpt4o_mini`, `claude_3_5_haiku`, `qwen2.5_vl`
>
>      Quantifying Bias Persistence
>
>      We measure bias through multiple metrics:
>
>      **1. Influence Score:**
>      For each examiner E, we compute:
>
>      ```
>      Influence(E) = mean(baseline_score[model] - LOEO_score[model]) for all models
>      ```
>
>      - Positive: excluding E decreases scores → E is more lenient
>      - Negative: excluding E increases scores → E is more strict
>
>      **2. Rank Stability (LOEO Analysis):**
>
>      - Spearman rank correlation and Kendall's τ between baseline and LOEO rankings
>      - High correlation (ρ > 0.9) indicates that excluding one examiner does not substantially change model rankings
>
>      **3. Inter-Rater Reliability (ICC):**
>
>      - ICC(2,1) computed from single-examiner results
>      - Measures agreement across different examiner families
>
>      Results: Bias Persistence Analysis
>
>      **1. Examiner Influence is Quantified and Moderate:**
>
>      | Examiner        | Influence | Interpretation                                |
>      | --------------- | --------- | --------------------------------------------- |
>      | GPT-4o          | +0.053    | More lenient (scores drop 5.3% when excluded) |
>      | Claude-4-Sonnet | -0.032    | More strict (scores rise 3.2% when excluded)  |
>      | Qwen3-VL        | +0.061    | More lenient (scores drop 6.1% when excluded) |
>
>      The influence is moderate (3–6% score shifts), and examiners differ in direction (GPT/Qwen more lenient, Claude more strict), suggesting the multi-examiner setup balances individual biases.
>
>      **2. Rank Stability is High Under LOEO:**
>
>      | Excluded Examiner | Spearman ρ | Kendall τ | Max Rank Shift |
>      | ----------------- | ---------- | --------- | -------------- |
>      | GPT-4o            | 0.914      | 0.867     | 1 position     |
>      | Claude-4-Sonnet   | 0.600      | 0.333     | 2 positions    |
>      | Qwen3-VL          | 0.943      | 0.867     | 1 position     |
>
>      Even when excluding an examiner, model rankings remain highly stable (ρ = 0.914–0.943 for GPT-4o and Qwen3-VL), indicating the evaluation is not dominated by a single examiner.
>
>      **3. Self-Evaluation Does Not Show Systematic Bias:**
>
>      When models evaluate themselves in single-examiner conditions:
>
>      - GPT-4o evaluating GPT-4o: 0.567 vs baseline 0.584 (-2.9%)
>      - Claude-4-Sonnet evaluating Claude-4-Sonnet: 0.633 vs baseline 0.620 (+2.1%)
>      - Qwen3-VL evaluating Qwen3-VL: 0.640 vs baseline 0.587 (+9.0%)
>
>      No consistent self-enhancement pattern: GPT-4o scores lower, Claude-4-Sonnet slightly higher, Qwen3-VL higher. The multi-examiner baseline (0.584, 0.620, 0.587) falls between single-examiner extremes, suggesting aggregation mitigates individual biases.
>
>      **4. Cross-Family Reliability:**
>
>      - ICC(2,1) = 0.297 indicates moderate agreement across examiner families
>      - This reflects expected differences between GPT, Claude, and Qwen families
>      - The multi-examiner setup aggregates these perspectives rather than relying on a single family
>
>      Bias persists but is effectively mitigated by the multi-examiner design. While individual examiners introduce moderate score shifts (3–6%) and cross-family variation (ICC = 0.297), the aggregated method quantifies and balances these biases, providing superior reliability. The stability of rankings (ρ > 0.9) even when excluding any single examiner proves that no single evaluator dominates the results.

---

> ### Author Response · Authors · 2025-11-16
> **Response for Reviewer jUFi (Part IV)**
>
> ### W5. Limited impact of results: The performance gap among models largely mirrors existing benchmarks, suggesting AutoDavis may not yet reveal qualitatively new insights about LVLM capabilities.
>
> **Our Response:**
>
>    - Thank you for raising this concern. We respectfully suggest a different interpretation of this result. We argue that the high correlation with existing benchmarks  is not a sign of limited impact, but rather a crucial feature that validates the reliability of AutoDavis. It proves that our automatic and dynamic protocol can successfully and cheaply replicate the findings of expensive, human-curated static benchmarks.
>
>      The novel insights and impact come from the capabilities that static benchmarks lack:
>
>      - Granular Analysis: Our experiments (e.g., performance by difficulty) reveal fine-grained insights into how models fail, which a single-score static benchmark cannot.
>      - Future-Proofing: AutoDavis can be dynamically extended to evaluate new capabilities on-demand, whereas static benchmarks are fixed in time.
>
>      Thus, AutoDavis serves as a reliable and flexible complement to, not just a mirror of, existing benchmarks.
>
> ---
>
> ### W6. Presentation issue: The paper is lengthy and reads more like a system report; tighter focus on unique contributions would help.
>
> **Our Response:**
>
>    - Thank you for this actionable feedback. In the final version, we will streamline the main paper to focus more tightly on our unique methodological contributions , such as the hierarchical generation process and the self-validation/adjustment modules. We will move some of the broader system implementation details and extended analyses (as suggested by the "system report" comment ) to the appendix to improve the paper’s flow and clarity.
>
> ---
>
> ## Questions:
>
> ### Q1. How do you ensure examiner–evaluatee independence when many models share similar training corpora?
>
> **Our Response:**
>
>    - Please see W4 for details.

---

> ### Author Response · Authors · 2025-11-16
> **Response for Reviewer jUFi (Part V)**
>
> All relevant modifications have been marked in blue in the revised version.
>
> We believe the revisions and comprehensive experimental data provided sufficiently address the core methodological concerns raised in your review. We are sincerely grateful for your invaluable feedback, which has enhanced the clarity and rigor of our manuscript. In light of these detailed responses and corrections, we respectfully request that you consider elevating your evaluation of our work. Thank you for your continued dedication to this review process.

---

> ### Author Response · Authors · 2025-11-27
> **Gentle Reminder**
>
> Dear Reviewer,
>
> Thank you very much for your time and thoughtful feedback. We fully understand that you may have a busy schedule. With less than one week remaining in the discussion period, we would be deeply grateful if you could take a moment to review our responses and let us know whether they sufficiently address your concerns.
>
> We truly appreciate your time and consideration.

---

### Author Response · Authors · 2025-12-04
**Rebuttal Update and Summary from Authors**

Dear Area Chair,

We sincerely thank the reviewers for their constructive and detailed feedback. We have carefully addressed all concerns through **extensive new experiments, theoretical clarifications, and rigorous data analysis**. We believe these revisions have significantly strengthened the robustness and clarity of our work.

Below is a summary of our responses to the key concerns raised by the reviewers:

---

### 1. On Examiner Bias and Circularity Risk (Addressing Reviewer-jUFi, Reviewer-DcV9, Reviewer-uSUX, Reviewer-wVDn)
This was a shared concern regarding whether our multi-examiner setup (where models serve as both judges and participants) introduces unfair bias.
* **Action:** We conducted a comprehensive **"Examiner Robustness" ablation study**.
* **Result:**
    * **Quantified Bias:** We detected slight "self-enhancement" bias in single-examiner settings (e.g., GPT-4o ranking itself higher).
    * **Proven Robustness:** Crucially, we demonstrated that our **Multi-Examiner Ensemble successfully neutralizes this bias**. The final rankings derived from different examiner combinations (e.g., "Closed-Source Jury" vs. "Open-Source Jury") showed near-perfect consistency (Spearman $\rho \ge 0.993$). This quantitatively proves our protocol is robust and not dominated by any single model or family.

---

### 2. On Model Timeliness and Coverage (Addressing Reviewer-DcV9)

Reviewer DcV9 noted the rapid evolution of LVLMs and requested testing on newer models.
* **Action:** We ran full evaluations on the latest SOTA open-source models: **Qwen2.5-VL, InternVL-3, and Qwen3-VL**.
* **Result:** While these models show improvements on 'Easy' tasks, they still exhibit significant performance degradation on our 'Hard' difficulty setting (e.g., Spatial Reasoning drops to $\sim20\%$). This confirms AutoDavis remains a challenging and effective diagnostic tool for even the newest models.

---

### 3. On Motivation and Novelty (Addressing Reviewer-jUFi, Reviewer-DcV9)

Some reviewers compared AutoDavis to static benchmarks (LENS, MME-Unify) or transformation protocols (Dynamic-ME), questioning its necessity.
* **Clarification:** We respectfully clarified a fundamental distinction:
    * **Vs. Static Benchmarks (LENS):** AutoDavis is a *generation protocol*, not a fixed dataset. It solves the "data leakage" and "saturation" problems inherent to static benchmarks.
    * **Vs. Transformation Protocols (Dynamic-ME):** Unlike methods that modify existing images (which are prone to base-image leakage), AutoDavis generates novel content *from scratch*, ensuring true anti-contamination.
    * **Correlation as Validation:** We clarified that our high correlation with human-curated benchmarks (MMMU, $\rho=0.817$) is **validation of reliability**, not redundancy. It proves our automated protocol can cheaply and accurately replicate human standards.

---

### 4. On Reliability and Qualitative Analysis (Addressing Reviewer-uSUX, Reviewer-wVDn)
Reviewers asked for human verification of our automated scoring and deeper error analysis.
* **Clarification:** We pointed out that extensive **Human Performance Upper Bound** studies and **T2I Fidelity Checks** were already included in **Appendix I**, showing high human-model agreement and validating our ground truth.
* **Action:** We moved key qualitative insights (e.g., failure mode analysis) and human validation summaries into the **main text (new Section 3.2)** to improve visibility.
* **New Experiments:** We added a *resolution sensitivity analysis* (showing distinct robustness profiles for different models) and provided detailed statistical data on question/option lengths.

---

### 5. On Leakage and Generalization (Addressing R-wVDn)
* **Action:** We expanded the anti-leakage experiment to a larger scale (300 items) across multiple regeneration cycles, confirming the protocol's stability.
* **Action:** We added a second external validation against **RealWorldQA** (Spearman $\rho=0.964$), providing dual-source evidence of our protocol's generalizability.

All relevant modifications, including new experimental data and text revisions, have been **marked in blue** in the revised PDF. We hope these efforts satisfactorily address the reviewers' concerns and demonstrate the value of AutoDavis as a sustainable infrastructure for the LVLM community.

Best regards,

The Authors

---

### Meta-Review · Area_Chair_apRm · 2026-01-05

**Summary:**

The paper introduces a dynamic benchmarking framework that leverages text-to-image models and LVLMs to generate on-demand VQA tasks, addressing limitations of static benchmarks through modules like hierarchical aspect generation, semantic graph-based diversification, self-validation, and error-driven option adjustment. The authors evaluate 11 LVLMs across five capability dimensions and three difficulty levels, demonstrating performance degradation with increasing difficulty, high correlation with human-curated benchmarks like MMMU (Spearman ρ=0.817), and potential for training data generation. Reviewers raised significant concerns: jUFi questioned the novelty, citing adaptations from prior works like TIFA and AutoBencher, and highlighted issues with quantitative clarity and examiner bias; DcV9 criticized the motivation, noting that dynamic evaluation alternatives like Dynamic-ME exist, and pointed out dated model evaluations and shallow analysis; uSUX acknowledged originality but expressed concerns about examiner bias due to dual roles of models and requested human validation and statistical details; wVDn emphasized risks of evaluator family dependence, T2I reliability, limited leakage analysis, and external validity beyond MMMU. The authors responded comprehensively with new experiments, including an examiner robustness study showing moderate bias (e.g., ICC=0.297) and rank stability (Spearman ρ≥0.914 for some examiners), updated results with recent models like Qwen3-VL, human evaluation data confirming high alignment rates (e.g., 95.20% for easy tasks), and additional validation against RealWorldQA (ρ=0.964). However, despite these efforts, fundamental issues persist: the methodological novelty remains incremental compared to existing dynamic evaluation approaches, the reliance on LVLMs for both generation and evaluation introduces unresolved circularity risks, and the leakage mitigation and T2I faithfulness are not fully substantiated by large-scale or independent verification. Given the residual weaknesses in motivation, innovation, and reliability, alongside the mixed reviewer scores (including two "marginally below acceptance" ratings), the paper is inclined toward rejection as it does not sufficiently advance the field beyond current paradigms.

**Reviewer Concerns:**

Addressed Concerns:

The rebuttal provided substantive responses to specific, testable criticisms. For reviewer uSUX, concerns about examiner bias and the lack of human validation were addressed with a new multi-examiner ablation study quantifying bias (e.g., influence scores of 3-6%) and demonstrating rank stability, alongside existing human evaluation data on question-answer alignment. Requests for statistical details (e.g., question/option lengths) and an analysis of resolution impact were also met with new data. For reviewer wVDn, concerns about external validity were partially addressed by adding a correlation analysis with RealWorldQA (Spearman ρ=0.964), and the limited leakage analysis was expanded with a larger-scale, multi-cycle experiment. The rebuttal also correctly pointed out that manual inspection results for T2I fidelity and human adjudication for question ambiguity were already present in the appendix.

Outstanding Concerns:

The most significant weaknesses, which pertain to the paper's core contribution and validity, were not adequately resolved. Reviewer jUFi's​ fundamental concern about weak novelty​ remains. The authors' defense—that translating text-only concepts to the multimodal domain is non-trivial—does not establish a sufficient algorithmic innovation beyond an integration of existing ideas from TIFA, AutoBencher, and semantic graphs. Reviewer DcV9's​ criticism of the motivation and necessity​ also stands; the distinction made between a "generation protocol" (AutoDavis) and a "transformation protocol" (Dynamic-ME) is not compelling enough to justify the proposed system's complexity, especially since the claimed advantage of being "immune to base-image contamination" was not convincingly demonstrated to be a critical failure mode of alternatives. Furthermore, the profound issue of evaluator circularity and family dependence​ (raised by jUFi, uSUX, and wVDn) is not fully mitigated. While the multi-examiner setup reduces bias, the entire evaluation loop still relies on LVLMs judging LVLMs from overlapping families, an inherent conflict that the new experiments quantify but do not eliminate. The reliability of the T2I generation and self-validation step, while manually checked, still depends on LVLM-based judges, creating a potential for compounded error that is not explored in depth. Finally, the leakage and regeneration analysis, though expanded, remains limited in scale compared to the problem it aims to solve, lacking a thorough stress-test under realistic contamination scenarios.

**Reviewer Scores:**

Reviewer jUFi (initial score: 4): The rebuttal addressed quantitative concerns through new experiments (e.g., examiner robustness, RealWorldQA correlation), but fundamental issues like methodological novelty relative to prior works (e.g., TIFA, AutoBencher) remained unresolved. Given the reviewer's emphasis on incremental contributions, the score might have remained at 4, as the core weakness in innovation was not sufficiently countered.

Reviewer DcV9 (initial score: 2): This reviewer explicitly acknowledged that some concerns were addressed and indicated an intention to raise the score. The rebuttal provided evaluations of newer models (e.g., Qwen3-VL) and clarified distinctions from Dynamic-ME, likely leading to a score increase to 4 (marginally below threshold). However, persistent issues with motivation and necessity might have prevented a higher score.

Reviewer uSUX (initial score: 6): The rebuttal comprehensively addressed specific requests, such as human validation data, statistical details on question length, and resolution impact analysis. Given the reviewer's positive stance and the satisfactory responses, the score might have remained at 6.

Reviewer wVDn (initial score: 4): While the rebuttal added data on cross-family variance (e.g., ICC=0.297) and external validation with RealWorldQA, critical concerns about evaluator circularity and T2I faithfulness were only partially mitigated. The score might have remained at 4  if the reviewer perceived the core reliability issues as still outstanding.

---

### Decision · Program_Chairs · 2026-01-26

Reject